# Emergence of Quantised Representations Isolated to Anisotropic Functions

## Abstract

Presented is a novel methodology for determining representational structure, which builds upon the existing Spotlight Resonance method. This new tool is used to gain insight into how discrete representations can emerge and organise in autoencoder models, through a controlled ablation study that alters only the activation function. Using this technique, the validity of whether function-driven symmetries can act as implicit inductive biases on representations is determined. Representations are found to tend to discretise when the activation functions are defined through a discrete algebraic permutation-equivariant symmetry. In contrast, they remain continuous under a continuous algebraic orthogonal-equivariant definition. This confirms the hypothesis that the symmetries of network primitives can carry unintended inductive biases, leading to task-independent artefactual structures in representations. The discrete symmetry of contemporary forms is shown to be a strong predictor for the production of symmetry-organised discrete representations emerging from otherwise continuous distributions — a quantisation effect. This motivates further reassessment of functional forms in common usage due to such unintended consequences. Moreover, this supports a general causal model for a mode in which discrete representations may form, and could constitute a prerequisite for downstream interpretability phenomena, including grandmother neurons, discrete coding schemes, general linear features and a type of Superposition. Hence, this tool and proposed mechanism for the influence of functional form on representations may provide insights into interpretability research. Finally, preliminary results indicate that quantisation of representations correlates with a measurable increase in reconstruction error, reinforcing previous conjectures that this collapse can be detrimental.

## 1 Introduction

This introduction will begin by outlining how many of deep learning's current functions are expected to influence the representations. This is hypothesised to constitute a function-driven emergent bias, motivating a comparative analysis against analogous functions that adhere to differing symmetries. Finally, an approach to isolate such influences through a controlled ablation study is discussed. This is centred around empirically testing a prediction made for such functional forms (Bird, 2025a).

### 1.1 Overview for How Functional Form Choices May Result in Biases

Many of the foundational functions in contemporary deep learning produce maps that differ as the direction changes; a property termed '*anisotropy*'. This angular variation of the functions is hypothesised to make specific directions distinct, which the network may then alter its representations about through optimisation. Hence, an angularly varying map may be expected to induce shaped angular representations. It is this connection between functional forms and representations which shall be explored in this work.

Directions singled out by a functional form's structure may be termed a 'privileged basis' (Elhage et al., 2022) or a set of 'distinguished directions'. Such distinguished directions are commonplace through activation functions, normalisers, initialisers, regularisers, optimisers, architectures, operations, and more (Bird, 2025a). Hence, there are several avenues hypothesised through which representational influence may occur: directly

through functional forms, indirectly through optimiser forms, and inherently through neuron connectivities. This proposed triad of influences suggests that the representational alignment phenomenon may be systematic to deep learning's anisotropic primitives and general practice. As a starting point, this work will isolate and determine the direct influence of activation functions as one of these causal influences on representations.

In particular, contemporary functional forms tend to make distinct the standard basis' vectors, $\hat{e}_i$, and in such a way that each basis vector's distinction is equal. This particular form of anisotropy can also be denoted through a permutation symmetry (Godfrey et al., 2023), represented standardly. This can be promoted to a *defining* characteristic relation underlying contemporary functional forms and categorised accordingly (Bird, 2025a). This permutation symmetry is often due to their elementwise application, but can appear more generally. In effect, anisotropic primitives provide an '*absolute frame*', for each representation space, about which the representations may organise and begin discretising into more distinct symmetry-led clusters.

The specific relation depends on the primitive in question; for most activation functions, this is an algebraic permutation equivariance, displayed in *Eqn.* 1[1]. Where $S_n$ is the permutation group, $\mathbf{P}$ is an element of that group represented standardly as a matrix, and $\mathbf{f} : \mathbb{R}^n \to \mathbb{R}^n$ is the function of interest. The hypothesis is that the *maximal* symmetries to which functional forms belong are a strong predictor of the emergent, representational structure organised by symmetry.

$$\forall \mathbf{P} \in S_n, \forall \vec{x} \in \mathbb{R}^n \quad \mathbf{f}\left(\mathbf{P}\vec{x}\right) = \mathbf{P}\mathbf{f}\left(\vec{x}\right) \tag{1}$$

The Spotlight Resonance Method (Bird, 2025b) has already empirically demonstrated that such activation functions can produce a *tendency* toward representational alignments about the distinguished directions that transform with this basis, thereby inferring an absolute frame. However, this study did not demonstrate that these functions were sufficient to produce these phenomena.

This was achieved through transformations to the standard basis and also varying the number of distinguished directions through the functional form. The former amounts to different matrix representations of the permutation group influencing representations, and the latter often corresponds to differing discrete rotation groups which vary in rotation angle (although some notably do not form group-defined forms).

It was found that corresponding changes in the representations occurred, which followed the transformed structure of the distinguished directions. The conclusion was that these functional forms stimulate the tendency towards representational alignment, producing emergent task-agnostic alignment structure in representations.

This established a causal relationship between the two; both transformed in step, and differing discrete symmetries produced similarly organised clusters. Hence, the observed structure showed a tendency for representations to cluster more densely around the functional form's distinguished directions. This work also indicated the presence of 'grandmother neurons', demonstrating how many clusters represent a distinct semantic.

However, this SRM study (Bird, 2025b) did not establish whether discretisation itself was a phenomenon of this choice of symmetry-defined functional form, as it did not compare continuous symmetry-defined forms against discrete symmetry-defined forms. This sufficiency is determined in the present work.

### 1.2 Motivating The Approach for this Study

Isotropic Deep Learning (Bird, 2025a) takes the consequences of such observations a step further. The Spotlight Resonance Method determined that task-agnostic structure in representations is associated with a functional form's distinguished directions. The Isotropic Deep Learning approach proposes that, since this structure observably transforms with the group's representation, which defines the functional form, then the choice of an anisotropic form may be the fundamental causal mode that induces the array of interpretable structures to begin with.

---

[1] This is assuming a map $\rho : \mathcal{G} \to \mathrm{GL}_n\left(\mathbb{R}\right)$ representation of the group, in this case one which permutes the standard basis. This form of representation will be assumed and implicit in all groups represented as matrices moving forward.

This work directly explores this prediction: testing whether downstream alignment is predicated on anisotropic forms. If validated, it reveals a systematic, unappreciated foundational bias embedded in modern deep learning. Additionally, it enables a mode of control over this bias and the resulting generation of function-induced structure. If this prediction is validated, it is singularly pivotal to assess whether this unintended bias is beneficial or detrimental — preliminary results of *App.* D support the latter. It would also have significant ramifications for the interpretability of networks, where the appearance and organisation of discrete-like features can be controlled.

Hence, it raises the question of whether the ubiquity of anisotropy in contemporary deep learning constitutes the underlying origin of many such interpretable and emergent structural phenomena commonly observed. If these are symmetry-led, then the organisation will be determined by the symmetry.

This position motivates the comparative study of this work. Determining whether one can control and demonstrate the emergence and elimination of *organised* structure through broadening deep learning's primitives to other forms, particularly those defined through differing maximal symmetries.

The hypothesis is that each may provide its own inductive bias. One such proposed choice is by redefining primitives through an orthogonal algebraic-equivariance (Bird, 2025a), displayed in *Eqn.* 2 for the orthogonal group $\mathrm{O}(n)$ and its elements represented standardly as matrices $\mathbf{R}$. This is termed an 'isotropic' definition for activation function forms, and is a supergroup-symmetry of the original permutation symmetry, $S_n \subset \mathrm{O}(n)$.

$$\forall \mathbf{R} \in \mathrm{O}(n), \forall \vec{x} \in \mathbb{R}^n \quad \mathbf{f}(\mathbf{R}\vec{x}) = \mathbf{R}\mathbf{f}(\vec{x}) \tag{2}$$

Choosing this definition for primitives is interesting, since it can eliminate all form-distinguished directions in a network. Therefore, functions developed to adhere to this modified symmetry definition can serve as a comparative mode to determine whether functional forms can induce these interpretable structures and to indicate whether downstream alignment phenomena are attributable to anisotropic choices.

If this is the case, it would demonstrate that these differing forms do carry unintended inductive biases, which would warrant major reevaluation of the forms used in standard practice.

The investigation of this mechanism improves upon the SRM (Bird, 2025b) by explaining *why* axis-aligned structures are produced, not just how they behave under basis changes.

Incidental alignment with directions chosen to be defined and observed as 'individual neurons' would otherwise have a vanishingly small likelihood of occurring in high-dimensional spaces. This is because there are vastly more orientations that do not align with the standard basis than those that do in high-dimensional spaces. However, if the analytic properties of permutation-based functions produce maps which differ along these directions, 'distinguished directions', then the symmetry breaking of the space would single out such directions and make alignment less improbable.

Therefore, it indicates an unintended bias of the model. A comparison between anisotropic and isotropic forms can directly probe such questions by adding or removing any distinction between directions.

Consequently, this study can be considered to determine a more foundational prerequisite phenomenon than the Spotlight Resonance Method observed and causally explained. This prior method elucidated how functional forms can causally influence *alignment* structure. In contrast, this work investigates whether forms can lead to the emergence of any induced task-agnostic structure. Determining whether current forms induce representational quantisation may indicate a causal mechanism for these other downstream phenomena. This then indicates whether these phenomena are truly fundamental to deep learning or are stimulated by, and hence contingent on, the specific *choices* in functional form currently in use. This could have significant implications, demonstrating that such choices do carry inherent, unappreciated biases in their implementation and affect deductions made in the approach to network interpretability.

### 1.3 Discussion of Methodological Approach

The Spotlight Resonance Method (SRM) hypothesised a hierarchy of anisotropic influences, particularly through the triad discussed previously. These interacting distinguished bases may be highly non-trivial to

predict any symmetry-led organisation of representations. Therefore, the causal mechanism was established in a clean, isolated, and minimalistic way through ablation studies of only the activation functions, to minimise confounding or obscuring influences.

Establishing this foundational case of a causal mechanism, linking symmetry-defined primitive functional forms to a form of representation influence, may then be expected to carry implications downstream through more general networks built on these primitive influences, where its more complicated interactions may be further predicted and studied. A discussion of generalising the predictions to other architectures is discussed in *App.* C.1. Hence, the SRM approach of establishing such a causal mechanism in a minimalistic setting will be repeated in this methodology.

As stated, instead of rotating the privileged basis or altering the number of distinguished directions to determine the alignment of representations, this study will compare representations in the presence and absence of any distinguished directions to determine whether this is a causal mechanism for the production of approximately discrete representations.

Due to the expected systematic nature of this inductive bias, great care is necessitated to prevent the unintentional reintroduction of anisotropy. To achieve this, distinguished directions will be removed network-wide to avoid these confounding and obscuring interactions. This is achieved by utilising isotropic primitives and an autoencoder model for reconstruction[2], except for an activation function which can selectively reintroduce anisotropy through changes to its form. Hence, this isolates the phenomenon's causality to the activation functions.

This requires two activation functions that are highly similar in action but differ only in whether anisotropy is present.

The Isotropic paper (Bird, 2025a) provides these preliminary isotropic activation functions, which enable this comparative analysis.

An analogue of standard elementwise (anisotropic) tanh, displayed in *Eqn.* 3, can be compared with its orthogonal analogue Isotropic-Tanh, displayed in *Eqn.* 4 — each is displayed in multivariate form for comparison $\mathbf{f} : \mathbb{R}^n \to \mathbb{R}^n$. Standard-Tanh is algebraically equivariant to the hyperoctahedral group (standard-basis permutation with sign-flips), $B_n$, whilst isotropic-tanh is algebraically equivariant to the orthogonal group, $\mathrm{O}(n)$, with $B_n \subset \mathrm{O}(n)$. The term $\|\vec{x}\| = \|\vec{x}\|_2$ is the standard two-norm of the vector $\vec{x} \in \mathbb{R}^n$.

$$\mathbf{f}_{\mathrm{aniso}}(\vec{x}; \{\hat{e}_i | \forall i\}) = \sum_{i=1}^{N} \tanh(\vec{x} \cdot \hat{e}_i)\, \hat{e}_i \quad (3) \qquad\qquad \mathbf{f}_{\mathrm{iso}}(\vec{x}) = \tanh(\|\vec{x}\|)\, \hat{x} \qquad (4)$$

These functions produce *exactly equivalent non-linear maps along the standard basis directions*: $f_{\mathrm{iso}}(\alpha \hat{e}_i) = f_{\mathrm{aniso}}(\alpha \hat{e}_i)$ forall $\hat{e}_i$ and $\alpha$. They only diverge away from the standard basis, but remain highly analogous in their non-linear map.

This enables a comparative ablation study of the influence of functional-form symmetries on representations and is sufficient to determine a causal mechanism for the emergence of a discrete-like structure.

Similar to this implementation, experiments are later extended to a Leaky-ReLU (Maas et al., 2013) family, which are modified to adhere to the following symmetries $S_n \subset B_n \subset \mathrm{O}(n)$. This is discussed further in *App.* B.4 and indicates that algebraic symmetries are the primary predictor, since other analytical properties of Leaky-ReLU substantially differ from the tanh-like family of *Eqns.* 3 and 4.

It is essential to note that isotropic networks still enable such clustering. They retain the ability through their bias terms, but they are suggested not to disproportionately encourage it through task-agnostic biases.

---

[2]Additionally, the use of an autoencoder model is essential to prevent anisotropy reintroduced through choices in the output layer representation. For example, one-hot encoding in classification outputs is also a choice in form defined through permutation symmetry; therefore, it could induce further spurious structure. A reconstruction task using autoencoders may therefore mitigate this and enable more data-representative and 'natural' embeddings. Moreover, it has been speculated (Bird, 2025b;a) whether the neural-collapse classification phenomenon (Papyan et al., 2020) is related to the proposed mechanisms through this permutation-defined classification layer. So, autoencoder reconstruction provided a suitable minimalistic setting, free from these influences, to study form-induced representational quantisation directly and without confounders.

Instead, it would likely reflect a task-necessitated collapse. Hence, there is no reason to expect perfect isotropy in representational distributions, as the underlying data or latent layer may include discretised aspects that arise independently of model effects. This makes the symmetry-induced structure the pivotal signature of these function-induced effects.

Therefore, this is *not* a test of whether distributions are anisotropic or isotropic; instead, it is the investigation of whether the emergent structure, specifically about distinguished directions, can be turned on or off through functional form choices. Overall, isotropic networks may be expected to produce emergent structures that are more representative of the underlying data, rather than being dependent on functional forms. This frame independence may be generally preferable and a natural choice, without task-agnostic distortions induced by distinguished directions. Hence, determining the validity of this prediction may provide crucial insight into the potential of Isotropic deep learning as an approach.

This work also advances the prior Spotlight Resonance Method (SRM). This modified and novel tool is called the "Privileged-Plane Projective Method" (PPP method) and operates on a similar principle.

However, unlike SRM, this new and adapted tool does not project out the vector length in the process. Thus, information regarding both angular and magnitude distributions is retained. This provides a more nuanced understanding of the network's complicated representational structures, which is critical for better interpreting the organised structure and the phenomena of study.

Additionally, the PPP method does not produce any angular-falloff smearing effect related to the cone width of SRM. Therefore, this modified method enables higher-fidelity angular maps of representations, which may be beneficial in many circumstances and essential for determining the exact nature of representational structures[3]. Together, this high-resolution geometric and intuitive insight, which importantly preserves magnitude-dependent, is the methodological intent behind developing and introducing the PPP visualisation tool used in this study.

This also positions the two tools — the existing SRM and the newly introduced PPP — for this study as similar, complementary, yet distinct approaches. The former remains better suited to statistical measures of angular representations, whilst the latter is preferable where complex geometry and high-resolution discernment are critical — such as in this study. The loss of angular fidelity in angular clustering metrics, such as SRM, can lead to a loss of discernment between the discrete linear features and the superposed arrangements of interest in this study. Hence, the novel PPP was developed and prioritised in this study to determine such structures, revealing a clearer geometric structure of representations. The following *Sec.* 3 provides a detailed overview of this new tool.

Using a comparative analysis between standard-Tanh and isotropic-Tanh, and similarly for the Leaky-ReLU family of functions, this PPP method can determine whether functional-form symmetry constitutes a causal mechanism for a *tendency* for discrete representation cluster structure to form due to task-agnostic functional-form symmetries.

It is used to demonstrate that algebraic permutation equivariance tends to produce a *symmetry-organised* quantising inductive bias on otherwise, underlying, near-continuous data structure, rather than task-necessitated representational collapses. This is contrasted with continuous-symmetry-defined functional forms, which are expected to tend to preserve a more continuous representation by not introducing spurious structure. Validation would directly demonstrate that such *choices* can carry representational inductive biases and require significant exploration.

Consequently, it suggests that downstream phenomena, such as interpretable neurons, discrete codes, (representational) superposition (Elhage et al., 2022), and possibly Neural Collapse (Papyan et al., 2020), are not fundamental but rather a direct consequence of anisotropic functional forms. This may have benefits for AI safety and interpretable networks; however, preliminary evidence also suggests that this function-driven, task-agnostic collapse and structure can be pathological, leading to increased reconstruction error.

---

[3]Together, the magnitude-preservation and high-fidelity angular discernment, uniquely position the Privileged-Plane Projective method as a tool for assessing whether neural refraction (Bird, 2025a) occurs for out-of-training-distribution outlier representations.

These pathological qualities of anisotropy are a secondary, preliminary outcome of this work. It may be associated with reduced representational capacity and expressiveness, as suggested by results in *App.* B, and with other hypothesised causes, such as robustness, discussed in *App.* C.3. However, this study is not intended to elucidate the exact mode of the pathology, so the results remain a preliminary indication. As a future direction, this may be investigated, resulting in a performance-interpretability tradeoff, if findings continue to be supported.

## 2 Theoretical Framework and Considerations

In this section, the theoretical aspects of the predictions are significantly developed. Particularly articulating how and why detectable signatures may form, their predicted appearance, alongside their potential implications.

A further discussion of how the theory may relate to other architectures and of the potential pathological implications of functional-form biases is presented in *App.* C, which provides further theoretical details.

### 2.1 Mechanistic Discussion of Function-Induced Structure

In this section, a more detailed discussion of how the proposed phenomenon is hypothesised to occur in networks is provided. This discussion makes clear the identifiable and detectable signatures that may present as a consequence of the theory. These explain why the prediction of quantised representations was expected for anisotropy, and lack thereof for isotropy, deriving from previous predictions (Bird, 2025a).

A further discussion pertaining to gradient-flow and first-order representational corrections is provided in *App.* C.2, which examines the mathematical emergence of the phenomenon.

This section's discussion is felt to deepen and ultimately supersede the more simplistic 'distinguished direction' heuristic. This is because, multivariately, the notion of distinguished directions is rather absolute, spatially discrete, and arbitrary in how they are defined. Instead, one can consider the multivariate maps to be more spatially continuous and locally varying in the regions they favour and to what degree they are preferred. Hence, this generalises the finite distinguished directions to a more spatially-continuous 'priviledging-field'-like approach. Where the priviledging-field is stronger, representations accumulate; where lower, they disperse — although the staticness of such a field is not implied. Notably, distinguished directions do remain important in characterising the particular representation of the (discrete) symmetries, so are retained to define the planes of the PPP method.

At a high level, the inducement of discrete structure is argued to stem from the function's anisotropy with its uneven computational map. Differing localities of this map may have differing practical usabilities. The network may act to leverage these by moving representations towards favourable areas and away from mappings of less utility — or, in the most general sense, attractors and repulsers may naturally form that influence representations' trajectories during optimisation. Hence, one would generally expect anisotropy to have more and less probabilistically favourable regions for representations to be embedded by the parameterised maps — where absolute uniformity is possible but probably unlikely to form.

Consequently, one would expect the anisotropy to shift from the algebraic definition, in the functional form, into the representations to some extent. These would yield predictable disturbances in density as the network learns to leverage the more useful local transforms. The implication is that these representations may gather more densely into certain regions whilst rarifying others during optimisation. These would be function-induced structures that form predictable over- and under-represented areas. These become a baseline signature of the phenomenon for that function.

Note that this does not determine the number of such clusters, which depends on the particular function that instantiates the form, nor, at this stage, its organised structures or its generalisability beyond the specific function.

This motivates the second essential aspect: the symmetry of this functional form. This determines the nature of that anisotropy, particularly its organisation, through its computational map's *degeneracies*. Additionally,

symmetry is also important not only in the inducement of such organised structure and implication for representational degrees-of-freedom, but this organisation also relates to the functional form as a whole. It may allow collation of generalisable behaviours across forms in a taxonomised symmetry approach (Bird, 2025a). The determination of such generalised findings would be important for model design along a symmetry-primitive design axis. Generalised findings supporting this taxonomy are demonstrated within this paper. A novel avenue could also explore general functions that can then be decomposed into a unique selection of symmetry-defined functional forms with known biases, each of which might contribute to the combined function.

Beginning with the symmetry organisation. Depending on the symmetry, these favourable computational maps may exhibit discrete degeneracy — locally repeating at numerous instances throughout the representation space. This may be beneficial to the network whilst providing a clear and detectable signature.

If a particular map is highly advantageous, then the network may be expected to crowd many representations across its localised non-linearity to achieve its desired computation. If there is only one occurrence of such a map, then the network may balance this desirable computation whilst retaining sufficient representational degrees of freedom. If it becomes too concentrated, information encoded by the distances between representations may be forfeited or suppressed as clusters become increasingly concentrated.

For example, this may take the form of many directions being projected onto a single axis, reducing the distribution's dimensionality and leading to conflation of associated semantics. Another possibility is that specific directions tangential to the usable map are scaled down to bring them closer to the usable locality, suppressing the associated semantics. These examples are non-exhaustive heuristics. This may be similarly applicable symmetry-or-not[4].

Hence, the network may redistribute representations to balance global representational degrees of freedom that express varied semantic information as a tradeoff with the locality of usable maps. Some nuance remains: some fraction of representations may benefit from one form of map, whilst another set may benefit from a different locality, potentially forming multiple clusters.

However, with the symmetry of such functional forms, there is this symmetry-induced organised degeneracy to these usable maps. The balance between representational degrees of freedom and densifying over these desirable maps is a markedly different tradeoff. The network needn't sacrifice inter-activation structure to the same degree; instead, important representational degrees of freedom can be retained by exploiting these same/similar maps in differing localities, such that the distribution remains more distinctly separable and more semantics are preserved.

The consequence of this hypothesis is that the denser patches of activations develop a task-agnostic structure, as clusters recur over similar maps, which are themselves organised directly by this symmetry of the functional form. The result is recurrent, symmetry-organised densification of representations onto multiple equivalent patches, without sacrificing degrees of freedom as much as concentrating in just one place. This organised densification is the key signature arising from such symmetry, which the PPP method then amplifies into observables.

The construction of the ablation trial can validate this empirically. One would expect to see a symmetry-defined angular structure in anisotropic networks, since the practicality of local maps fluctuates; however, in isotropic networks, one would not expect to see such *task-agnostic*, form-driven structure, as all angular localities are equivalent — there is no need for the network to overrepresent certain areas. It is the continuous angular limit in map degeneracy; there is no need to suppress angular degrees of freedom when seeking a localised map.

---

[4]'Not' still relates to the identity symmetry, you just lose the crucial signature of symmetry-dictated organisation. Similarly, in the SRM study (Bird, 2025b), some proposed maps don't clearly arise from symmetry for some (approximate) Thomson bases, yet they are likely to still have some discrete rotational symmetry, even if approximate; hence, representation organisation may still occur about them. Potentially different discrete symmetries arise at different approximation tolerances. This may be explored through weighted combinations of functional forms.

Hence, the tradeoff between representational degrees-of-freedom and leveraging localised computations disappears in isotropic networks, as there are no angularly localised maps[5] — it can utilise practical mappings without a detrimental tradeoff[6]. Hence, suppression of particular semantics needn't happen unless task-necessitated.

Consequently, one would predict that the general signature of discrete-symmetry networks is a task-agnostic representational structure organised about the defining symmetry. A distinct absence of such task-agnostic structure would then be expected for isotropy. Observing a difference in task-agnostic structure between these would validate this general hypothesis. This general approach will be used to study how various function-driven symmetries interact and act on the network and its representations, determining the various biases induced.

For example, considering permutation-like symmetries about a particular basis would result in an orthant partitioning of the space (Bird, 2025a). Depending on the specific permutation symmetry, various sets of these orthants may be analytically degenerate, producing transformed copies of equivalent maps — a manifestation of the underlying group definition. The network's optimisation may then shape representations about these orthants, producing analogous clusters under these equal mappings and yielding an approximate symmetry structure resembling the orthant decomposition and the initial, defining permutation symmetry. Hence, a response of optimisation to the symmetry.

Generalising these anisotropic structures may be an important foundational mode for the formation of classical axis-alignment, discrete linear features, (representational) Superposition, grandmother neurons and more.

It is also singularly essential to note that this is a form-driven 'task-agnostic' organisational structure, something seen repeated through different networks and hence infers a foundational function-driven bias as opposed to a data-dependent fluctuation. Data may naturally contain such over- and underdensities, so the detection of which is not sufficient to deem it a foundational bias. Hence, conflation between the algebraic an/isotropy of the function and a direct expectation of a distributional an/isotropy in representations is not encouraged.

Instead, the *tendency* towards *repeated*, and *symmetry-organised*, structure is the hallmark signature of this potentially significant and unappreciated bias. This motivates the careful setup of this study to demonstrate clearly that it is function-driven, not data-driven, by detecting repeated instances of the same structure within the same data and task, then comparing them against differing functional forms, which constitutes the comparative ablation approach.

The PPP method is particularly effective in amplifying this symmetry-organised structure by conglomerating many single planes, themselves defined by the symmetry, into a composite plot. Hence, if such over- and under-densities are organised by symmetry, then their discrepancies will compound, making the signature more evident.

This does not eliminate the possibility of a single, very strong, dense patch of representations, rather than many smaller symmetry-organised patches. This latter consideration constitutes an additional reason why a comparative ablation trial against isotropy is essential, as either form of task-agnostic structure can be attributed to the functional form presence of anisotropy if it does not similarly occur within the purely isotropic networks. Despite this caveat, in this paper, symmetry remains the central defining factor shown to induce structure. It will be shown that PPP frequently demonstrates several clearly visible patches organised about the symmetry. Hence, the results advance the symmetry-based functional form theory on how the wide variety of interpretable structures fundamentally form in a causal manner. This makes clear

---

[5]This makes clear the goal of isotropic function search: to recreate a highly usable map across all angles and hence such a tradeoff is never necessitated. Anisotropic maps necessarily vary because the maximal symmetry is discrete, so this is not possible; the network must strike a balance. This positions isotropy as potentially preferable if suitable functions can be found. If multiple maps are needed for differing representations, one can stack isotropic functions and potentially explore function-routing masks per representation/locality.

[6]A radial tradeoff may still occur, due to the non-linear map along any radial direction. However, some non-linearity is required for operation. This associates with this paper's proposed coding scheme: a magnitude-direction hypothesis for stimulus-degree and semanticity, respectively.

these are consequences of symmetry-defined functional form *choices* than any fundamental and spontaneous emergent/mysterious occurrences of such representational phenomena.

Finally, a note on the sharpness of such clusters. In general, these may vary between diffuse and sharp clusters, depending on the map, but should remain symmetry-organised fluctuations in any case due to the shared functional form.

In the extreme of a highly usable locality near much less usable localities, one would expect to observe much sharper representation distributions, producing near-discrete-like quantised clusters. The shape and narrowness of these may indicate the shape and extent of the useful computations. This may produce narrow beams where inter-activation degrees of freedom are suppressed in favour of the improved map, alongside the semantics associated with these directions, as separability is lost.

The number of such beams is still undetermined by the symmetry-defined form, as is specific to the function, but may in future work be explored in terms of a multiplicity due to the symmetry (particularly in the upcoming orthant picture). In the results, it is observed that multiple narrow, discrete-like beams shift and converge into a single stronger beam, showing that optimisation forms highly dense patches over particularly useful maps, which may shift to continue 'finding' better transforms.

It also so happens, that the observations provided in this work, do tend to have these very sharp nearly-discrete quantised representational clusters for anisotropy, and often approximatly smooth continua for isotropy, making the analysis incidently more idealistic and hence, more certainty in conclusions — although, both do show some variation from plot-to-plot. However, despite an association between the two, it may not always be expected that anisotropic and isotropic definitions for functions and representation distributions will coincide.

Overall, it is instead the symmetry-led *organisation* of generally more concentrated and rarefied representation patches which is the general signature. These structured unevennesses in the representations would then reflect the function-form-induced symmetry. Where multiple functions and symmetries are involved, this structure may become more complicated; hence the necessity of a minimalistic, otherwise isotropically controlled autoencoder on reconstruction to minimise these complicating confounders.

This section offers a mechanistic account predicting that functional forms are not benign in their effects on representations, but instead directly induce their structure into predictable, task-agnostic fluctuations in the density of representations. The organisation of these directly stems from the symmetry defining the particular functional form. Drawn from this is the primary hypothesis that since isotropic functional forms are devoid of distinguished directions, they may consequently eliminate any tendency towards task-agnostic representational collapse organised by a discrete symmetry's uneven map.

Validating this unequivocally shows that foundational biases arise directly from our *choices* of functional form for primitives, and other primitives must be similarly investigated. Since these choices are then shown to carry consequences, this necessitates a highly significant interrogative analysis of the foundational functions of the deep learning approach and supports a taxonomisation of functions and their consequences. This generalised investigation is encouraged to span across initialisation, normalisers, optimisers, operations and more. If its findings generalise, they change the conceptual foundation for how we understand the organisation and appearance of learned representations. This study supports the hypothesis that such choices may be significant representational inductive biases and worth considerable exploration.

## 2.2 Broader Impact Concerns

Although indirect, there exists potential for the consequences of isotropy to be leveraged for undesirable means. As it will be shown, isotropic networks tend to produce smoother continua of representations, opposed to anisotropic's inducement of clustered representational structure. With isotropic development comes the risk of intentionally using such functions to create influential networks that are functionally black boxes. This is a critical issue where networks may need to be transparent in decision-making, such as in the medical or justice systems.

Many current interpretability tools rely on this cluster separability to intuit model decisions, predicted behaviours, and broadly understanding the network — isotropic functions induce distributions opposed to

such tooling. They could be used to obfuscate such understanding, maliciously making the models harder to audit, and in turn requiring the development of new tooling to match existing interpretability.

It is felt that practitioners should be mindful of these non-negligible risks when implementing functions and, in parallel, strive to develop tooling that brings the interpretability of isotropic networks in line with current anisotropic tooling. This may also be circularly advantageous for creating flexible, generalised tools that may benefit the broader field beyond isotropy. This ensures that developments in isotropic model capability do not come at the expense of transparency.

Both the SRM (Bird, 2025b) and the PPP method already provide examples of isotropic-considerate interpretability tools. Incorporating built-in semantic interpretation into these tools may give further insight for safety-critical applications.

Overall, the development of isotropic functions is encouraged as a practical and advantageous route to exploring and leveraging foundational biases; yet, it is vital to be cautious of the stated downsides and proactively mitigate this potential eventuality.

## 3    Privileged-Plane Projective Method Implementation

The Privileged-Plane Projective Method (PPP method) operates similarly to the Spotlight Resonance Method (SRM). It appears to work on any architecture, at any layer and for any task. It only requires a set of basis vectors to operate on, typically the distinguished vectors, producing 'privileged (oriented) planes'. It provides a tool to interpret the structure of internal embeddings in high-dimensional spaces by projecting them down to privileged two-dimensional planes, where representations have been shown to align about. This differs from the SRM method, which sweeps out these planes with rotating probe cones to map the density fluctuations in representations. Instead, this method produces modified slices of the space to provide crisper details to the structure of representations through over- and under-densities — hot spots produced by the technique indicate these. This provides greater information to intuit distributions in a visual manner, including magnitude information, and is therefore proposed as a successor to the prior method. The methodology is as follows:

Given a set of distinguished unit-normalised vectors, $\left\{\hat{b}_i | \forall i\right\}$, create all non-parallel pairwise combinations or permutations of the set: $\left\{\left(\hat{b}_1, \hat{b}_2\right), \cdots\right\}$. These two choices are referred to as Combination-PPP and Permutation-PPP, respectively.

From these two vectors, oriented $\mathbb{R}^2$ planes can be formed as a subspace of the $\mathbb{R}^n$ space. These are the 'privileged planes' onto which representations can be projected. There are several ways to choose a threshold for projection, such as a thresholded closest distance to plane; however, an angular threshold was instead chosen[7].

This thresholding can be achieved through the decomposition of representation vectors, $\vec{v}$. These can be decomposed into an in-plane component and a perpendicular component $\vec{v} = \vec{v}_{\parallel} + \vec{v}_{\perp}$. This is achieved through determining the coefficients within the plane: $\vec{v}_{\perp} = \vec{v} - \vec{v}_{\parallel}$.

The pair of vectors defining a plane may not be orthogonal in non-standard sets. When displaying the two coefficients, this can produce artefactual distortion in the method. These can be removed by forming an orthonormal basis for the plane: $a\hat{b}_1 + b\hat{b}_2 = c\hat{e}_1 + d\hat{e}_2$, where $\hat{e}_i \cdot \hat{e}_j = \delta_{ij}$. This can be achieved in steps, driven by the need to preserve the plane's orientation. The unit-vector $\hat{b}_1$ is assigned to $\hat{e}_1$ and $\hat{e}_2$ is chosen such that $\hat{e}_2 \cdot \hat{b}_2 \geq 0$ — preserving the orientation. Therefore, the new basis is defined as $\hat{e}_1 = \hat{b}_1$ and through *Eqn.* 5, which is a Gram-Schmidt procedure for orthonormal basis construction.

---

[7]This latter one gives a better interpretation for alignment, all within a small angle of the plane, and practically was found to yield better results. However, vectors close to the origin are often outside of the angular threshold due to the vanishingly small volume. Therefore, frequently, an artefactual circular zone of absent representations is observed. Hence, other choices or a hybrid of these two approaches may also be considered, but were not included for simplicity.

$$\hat{e}_2 = \frac{\hat{b}_2 - \left(\hat{b}_2 \cdot \hat{b}_1\right)\hat{b}_1}{\left\|\hat{b}_2 - \left(\hat{b}_2 \cdot \hat{b}_1\right)\hat{b}_1\right\|} \tag{5}$$

These two $\mathbb{R}^n$ orthonormal vectors can then be stacked into a matrix of shape $\mathbf{E} \in \mathbb{R}^{2 \times n}$, and pseudo-inverse can be used to determine the coefficients for the in-plane vector with respect to this basis: $\vec{w} = \left(\mathbf{E}^T\mathbf{E}\right)^{-1}\mathbf{E}^T\vec{v}$, with $\vec{v}_\parallel = \vec{w}^T\mathbf{E}$. This constitutes the desired projection into the privileged plane.

Finally, using the vector magnitudes one can produce an angular threshold: $\left\|\vec{v}_\parallel\right\|/\left\|\vec{v}\right\| > \epsilon$, where $\theta_{\text{threshold}} = \arccos\epsilon$. Any representations which meet this angular threshold have an associated $\vec{w}$ displayed in the PPP method's plot. This is repeated for all pairwise vectors chosen and all representations chosen.

The coefficients $\vec{w}$ can be displayed in many ways in a $\mathbb{R}^2$ plot. It would seem beneficial to generally have this centred around the origin with equally scaled axes. This could be achieved through a scatter plot or a density plot. The latter was chosen only for visual appeal. This required forming a high-resolution grid over a region of interest and recording the value of a narrow Gaussian centred with a mean of $\vec{w}$ and small variance $\alpha 1_2$. A Gaussian for each $\vec{w}$ was produced, and the grid represents their sum. This could then be displayed as a series of images with colour gradients scaled by the maximum value across all images. This provides an impression of representation density, which remains comparable across time steps. All the following plots are produced in such a manner. Alternatively, colour coding could be used to indicate the angle subtended between each representation and the privileged plane, within a scatter plot. This latter method would provide further angular information perpendicular to the plane, but was not used in the results of this paper.

In this paper, $\epsilon = 0.75$ was found to work well and was therefore used in the production of all results. Similar results were found for larger $\epsilon$, but the plots had fewer representations due to the smaller volume, so structures were less definitive in their appearance. Unless otherwise stated, all measurements were taken using the combination-PPP method, using the standard basis. This was found to be beneficial compared to permutation-PPP, which could produce statistical artefacts discussed further in *App.* A.

## 4    Experiments and Interpretation

In this section, comparisons between isotropic and anisotropic activation functions are displayed and discussed. The examples shown are an excerpt from a larger variety of results available in *App.* B. The exact method for training these networks is discussed in *App.* E. A code repository for the PPP method is available at [URL REDACTED FOR ANONYMITY].

Additionally, a small disclaimer: in the extended results of *App.* B, a normalisation technique is used to mitigate dataset-specific statistical artefacts arising from the $[0, 1]$ bounding of data elementwise. This normalisation step consists of a preprocessing step in which each sample has the dataset pixel-wise mean subtracted and the pixel-wise variance unit-standardised, excluding pixels that are zero. This reduced the impact of data-format-related artefacts on the findings. This process is discussed more thoroughly in *App.* A.

However, this preprocessing standardisation was considered somewhat artificial and was introduced for pragmatic reasons. Therefore, it is not included in the results of this section. However, the observations were present with and without normalisations. Thus, conclusions reached are not influenced by such choices, as quantisation induced by anisotropic functions was seen in both cases, with only minor qualitative differences.

The first result provides a particularly clear demonstration of the quantisation phenomenon resulting from anisotropic functional forms, as opposed to the more continuous representations produced by isotropic functional forms.

In all cases from *Fig.* 1, one can observe that the anisotropic row representations form distinct, highly dense, approximately discrete structures, which through training align (horizontal and vertical) or anti-align (diagonal) with the distinguished directions. In contrast, the bottom-five isotropic rows produce more circularly symmetric, continuous representations with no specific alignment tendencies. Furthermore, this example alone is considered sufficiently indicative of a representational inductive bias deriving from functional

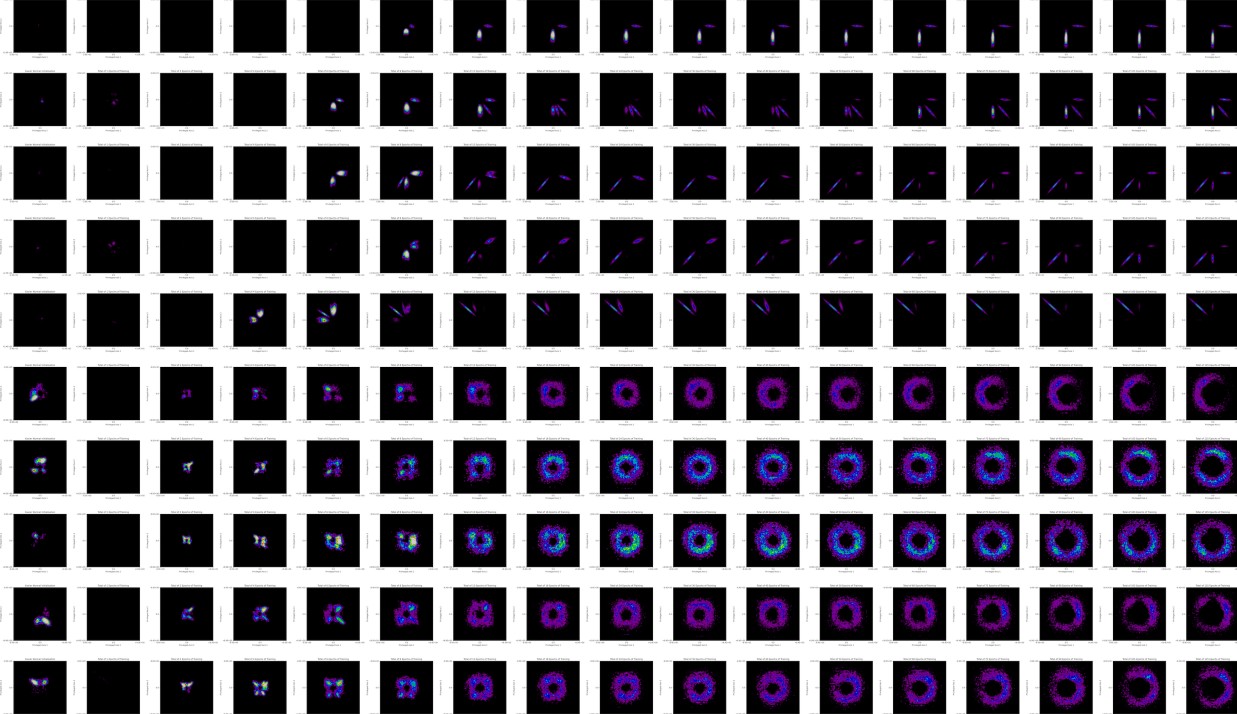

Figure 1: Displays ten rows of PPP-method's results, divided into columns. Each row represents an independent autoencoder network trained on the reconstruction of the CIFAR dataset (Alex, 2009). Each column represents the results of the PPP method at various stages of training. The leftmost column shows a freshly initialised model before training, and moving rightward, the network is progressively trained for up to 125 epochs, as shown in the rightmost column. Hot-spots indicate where the internal latent layer is particularly dense with representations — collated over all samples from the CIFAR training set. The top five rows depict networks which utilise the anisotropic activation function, standard-tanh, whilst the bottom five rows utilise the isotropic activation function, isotropic-tanh. Every other detail is identical otherwise. Figure titles indicate the exact number of epochs trained for. This specific network consists of a latent layer of 18 neurons, with standard (unnormalised) input-output pairs drawn from CIFAR. The dark centres about the origin are attributed to a vanishing volume due to an angular threshold, rather than the absence of representations.

form choices, thereby validating the hypothesis. This suggests that anisotropic forms are sufficient to induce a discrete representational tendency with task-agnostic clustering along distinguished directions.

There are several interesting further aspects of these results.

Firstly, the construction of this specific autoencoder ensured no activation function preceded the latent layer of study, similarly to the spotlight resonance method. This eliminated the possibility that these alignment phenomena are trivially due to the bounding geometry of an activation function, including limit points. The activation functions are only applied after the latent layer of study. Hence, this indicates how the functional forms in later layers can exert a representational bias generally through optimisation throughout the network. Thus, the functional form can bias the parameter trajectories during learning, which then transforms the representations into these discrete structures. Therefore, this represents a non-trivial representational bias due to functional forms, confirming the prior prediction (Bird, 2025a), and in the general case preceding any trivial bounding or neural refraction. Further results also substantiate this.

Additionally, the results suggest that discrete-representational Superposition (Elhage et al., 2022) arises uniquely in the anisotropic functional form. Several rows, namely two through five, display discrete directions

that are either aligned with or anti-aligned with the standard basis, but are sometimes also arranged at more general angles. This is further demonstrated in *Fig.* 2, which is the same autoencoder but with a greater number of hidden latent layers.

This is believed to be indicative of a *discrete* representational structure consistent with descriptions of Superposed features[8] (Elhage et al., 2022). Additionally, it also shows that activation functions can exert influence backwards through the model, affecting representations in latent layers that precede any activation function in the forward pass. This reframes several Superposition experiments (Elhage et al., 2022), which identify the phenomenon in such cases.

This indicates that discrete representational Superposition phenomena arise due to anisotropic forms, but whether these forms provide a complete description of both suggested forms of Superposition remains debatable — especially for the phenomenon's dual through parameters, which could still be applicable under continuous representations. Further analysis of individual planes from the PPP method may be necessary to assess the nuances of the Superposition in more detail, rather than relying on this conglomerate plot across all planes. However, this would require many more samples to populate individual planes sufficiently. Additionally, several columns on isotropic networks for early-stage training also indicate directions of high density, but these progressively distribute throughout training. These are believed to be statistical in nature, as optimisation has not had the opportunity to transfer asymmetry in forms to representations yet. This is discussed further in *App.* A. Overall, it is suggestive that representational Superposition may be a network-adaptive emergent phenomenon predicated upon the imposed task-agnostic anisotropic biases.

Generally, anisotropic functional forms appear to produce an increasingly discrete structure in representations as training progresses, starting with angularly arranged but more diffuse representations and gradually refining into narrow, angular beams. This appears consistent with a linear feature hypothesis (Elhage et al., 2022). This structure can be seen to be contingent on the anisotropic functional form, suggesting it carries a quantising inductive bias. Additionally, the anisotropic network's linear features align about both positive and negative distinguished directions, appearing to respect the sign-flip and permutation (hyperoctahedral symmetry) of the standard-tanh functional form. This may be indicative of the analytically degenerate orthants produced by such forms. This is contrasted with the isotropic form examples, which retain a general continuous representational structure which becomes decreasingly discrete through training.

Furthermore, hot spots in isotropic representations are more generally angularly distributed and broader, without forming distinct geometries as in the anisotropic cases, such as the high densities along orthogonal directions. This may suggest that the distribution produced in isotropic networks more closely represents the natural structure of the dataset, as it is likely to have more varied angular clusters. Yet, overall, this demonstrates that the representations appear to be influenced by the symmetry structure of the functional forms.

Overall, these observed structures are consistent with the sign-flip and permutation (hyperoctahedral) symmetry of the standard basis, which defines the construction of these anisotropic forms. In contrast, the continuous orthogonal functional form yields a smoother, more continuous distribution. This indicates that the maximal symmetry family is a useful predictor for some representational inductive biases. Comparisons between additional functions derived from each form can substantiate this connection, and this is undertaken in *App.* B.4. Furthermore, it suggests that symmetry taxonomies are potentially crucial to investigate in terms of interpretability, representational capacity, and resultant performance.

Finally, regarding the development through training, one may notice the absence of PPP method signatures for early training and initialisation of anisotropic networks only. This is not an absence of representations, but a relative underdensity compared to late-stage training plots.

This is due to the colour map chosen for the PPP method, which is calculated row-wise, such that all plots on the same row have identical colour map thresholds, enabling better comparison for structure development through training. However, in later layers, the clustering towards these discrete directions is so high that the

---

[8]These are thought to be representational examples of the phenomenon, which differs from the original measurement, which was often concerning the autoencoding network's parameters. Thus, this potentially provides a sort of dual observation of the phenomenon. Additionally, row two demonstrates exclusively anti-aligned representations to the standard basis, which appears to diverge from the classical alignment-first features.

relative densities in early plots are comparatively negligible and are therefore coloured black by the colour map. This was observed in the number of samples provided by the PPP method. For these plots, the typical number of samples that met the threshold was approximately 2000 for both early-layer anisotropic networks and all isotropic networks. However, for late-stage training in anisotropic networks, this typically increases by two orders of magnitude to around 200,000 samples, indicating that a great majority of representations align with the observed structure. In contrast, isotropic representations remain more evenly distributed throughout the space throughout training.

This shows that through training, anisotropic networks vastly increase in the number of representations aligned with these structures, offering an indication that functional forms can provide a *strong* inductive bias. This is corroborated by the isotropic networks, where the maximal density is significantly lower and consistent throughout training; therefore, early-training representations appear on the colour map. Furthermore, isotropy's more even utilisation of available space suggests representation capacity may grow exponentially with network width for isotropic models, whereas anisotropic networks often *tend* to have an aligned structure, which indicates a linear growth associated with a local coding. However, more complicated Superposed arrangements can improve this. This preliminary indication of exponential representational capacity of isotropy is discussed further in *App.* B.1.

With all these results, there remains the possibility that the anisotropic representations are connected outside of the thresholded region for the projection; however, this appears highly unlikely in two regards. First, it would seem rather improbable that such an elusive connection would form in the exact perpendicular directions that the PPP method projects over. Moreover, the exceptionally high density of representations within these discrete clusters in anisotropic rows would suggest that there are insufficient remaining samples to connect clusters outside of the projected volume. Overall, it seems prudent to conclude that the representations follow from the symmetry-defined form of these activation functions.

A final observation is that spontaneous symmetry breaking appears to set distinct directions early during training, and later training only subtly refines these directions. This indicates a strong dependence on initialisation in networks and supports considering the initialiser's probabilistic symmetry presented in the symmetry-based taxonomy (Bird, 2025a). These may also constitute inductive biases through interactions. Additionally, in the future, this could be explored in terms of whether isotropic networks have a slightly reduced dependence on initialisation, due to momentum enabling a traversal of identically flat connected basins along the symmetry group's orbit (Bird, 2025a).

All these observations remain pertinent to *Fig.* 2, which presents measurements from a similar, yet deeper, autoencoder model.

This model notably produces more complicated discrete structures in anisotropic functional forms, indicating that more nuanced representational Superposition may be occurring due to the increased model depth. Additionally, the bottom five rows corresponding to isotropic functional forms have an increased amount of clustering compared to the previous, more uniform distribution of *Fig.* 1. This clustering is unaligned with the standard basis, particularly evident in rows 6, 8, and 9. Close inspection of rows 7 and 10 also reveals subtler misalignments. Together with slight clusterings in *Fig.* 1, this suggests that it is not due to an imposed structure from functional forms and is indicative of a desirable task-driven clustering.

This observation evidences that an isotropic network can still enable the phenomenon when beneficial. However, these clusters form a more general non-standard arrangement, suggesting that externally imposing discretisation along a standard basis, using contemporary anisotropic functions, may not be optimal. Additionally, the same networks were found to return to a largely smooth continuous distribution in subsequent latent layers, discussed further in *App.* B.3.

In the appendix results, there are instances where isotropic experiments display slight hotspots, which sometimes align or anti-align with the standard basis for isotropic networks. This would be surprisingly unlikely given the uniform nature of isotropy and is instead demonstrated to be a statistical artefact following further controls on experimentation in *App.* A. Namely, this is believed to be a statistical result due to averages over independent variables for anti-aligned positions, whereas averages over correlated variables for axis-aligned positions. The volumes about the distinguished vectors become overrepresented in the plot,

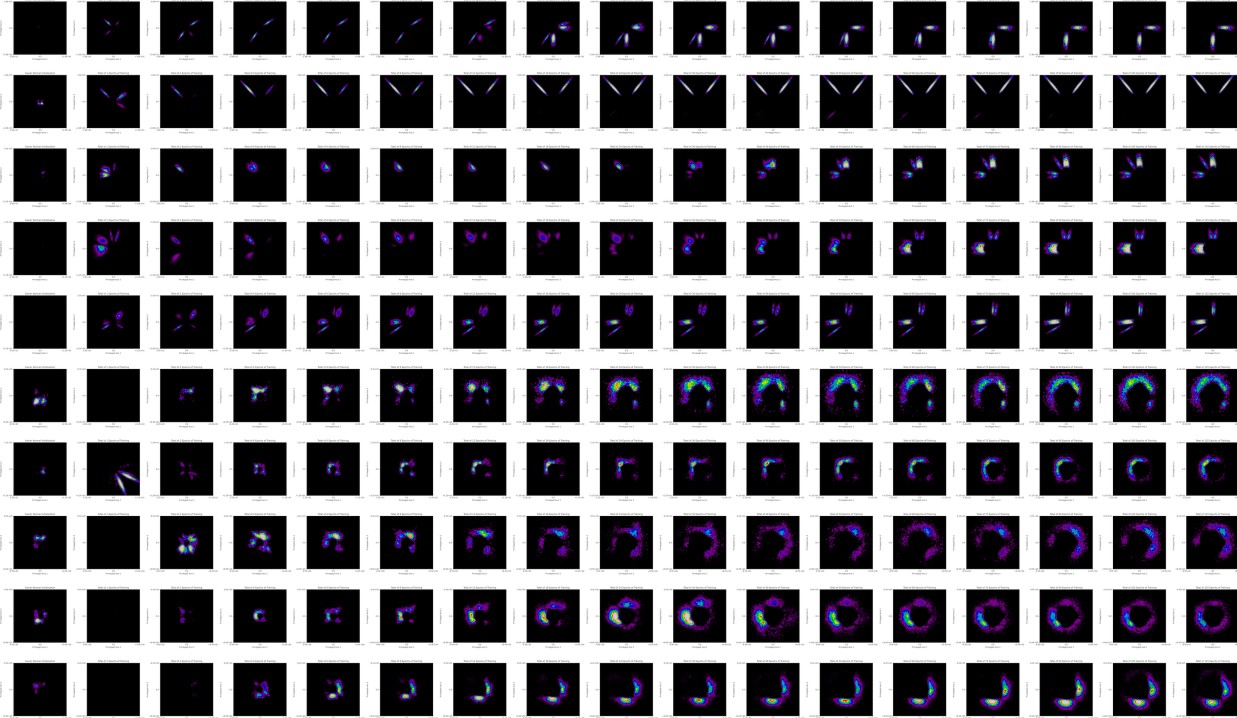

Figure 2: Displays an identical plot to *Fig.* 1, except for the results being drawn from a network with three latent layers, each with 18 neurons per layer. The latent layer studied is the first latent layer, which precedes any activation function. Moreover, one can see in rows 1, 3, 4, and 5 that narrow individual beams slowly shift in position during optimisation, converging on strong axis-aligned distributions. This may suggest these axis-aligned positions offer more favourable non-linear maps for computation than elsewhere.

due to the repeated intersection of privileged planes derived from the standard basis. Therefore, averages on these intersections display higher variability, due to the inapplicability of the regression-to-the-mean principle in dependent variables. Hence, under- or over-densities may be statistically more prevalent along standard basis directions, but this effect is typically subtle, although it can be observable. This appendix presents demonstrations of rotated measurement bases for identical isotropic models, which continue to display the phenomenon; therefore, it is deemed highly unlikely that persistent unintended anisotropies exist in implementations or true clusters. Instead, they are attributed to the statistical noise inherent to such conglomerated projection methods. This statistical noise may be skewing row 10 of *Fig.* 2 to appear slightly axis-aligned; alternatively, this could also be entirely coincidental to the experiment run. Nevertheless, the strong trend holds that function-derived symmetries act as inductive biases.

Finally, *Fig.* 3, demonstrates a similar phenomenon in a comparison between standard Leaky-ReLU and isotropic-Leaky-ReLU functions — details of these functions and further tests can be found in *App.* B.4. Such results are significant, not only because the ReLU family has a higher prevalence than tanh in contemporary models, but also because they analytically differ from tanh in almost every regard, except for the underlying symmetry constructions. Therefore, the demonstration of quantising effects of these activation functions is highly indicative that it is the underlying algebraic symmetries which correspond to such an outcome. This isolates algebraic symmetry as a significant predictor of such phenomena, which is considered impactful for interpretability and explainable AI approaches. Furthermore, each symmetry may correspond to a particular inductive bias, supporting the grouping of functions into taxonomic categories, which may aid in generalising the predicted emergent phenomena.

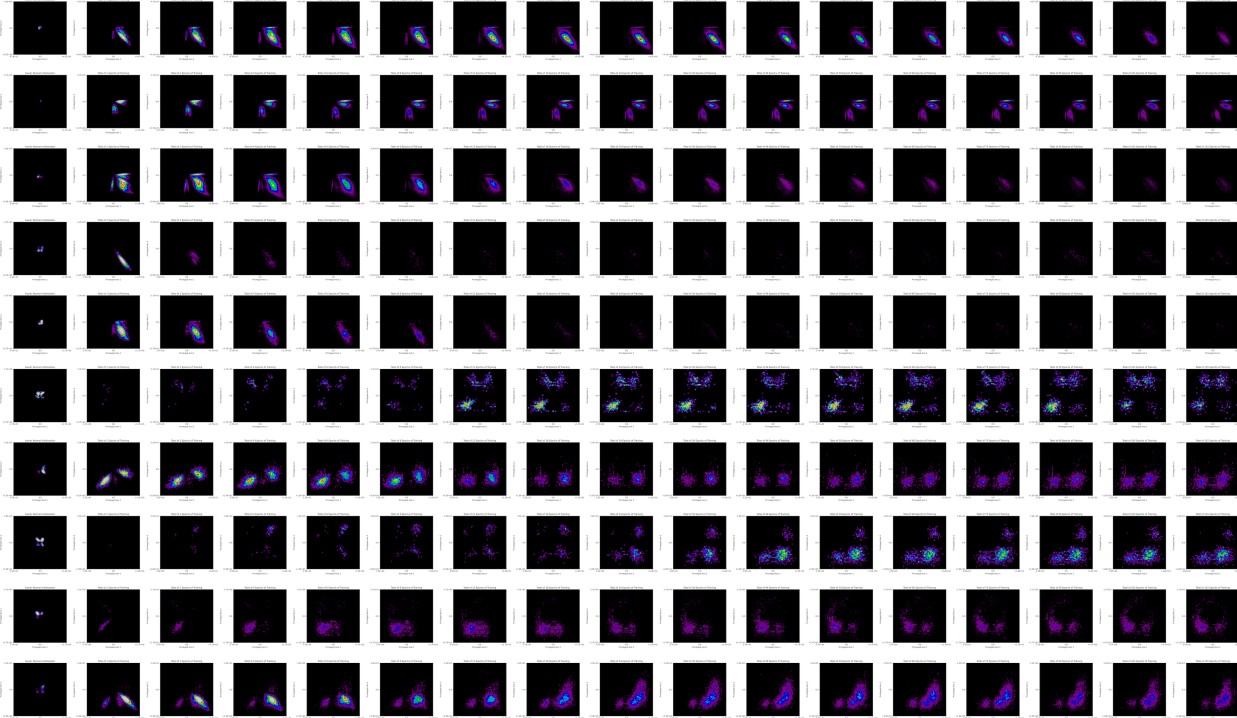

Figure 3: This plot displays identical networks, in every way, to *Fig.* 2, except from differing in the activation function applied. The top five rows utilise the standard Leaky-ReLU function, whilst the bottom five rows utilise an analogous isotropic Leaky-ReLU function, defined in *App.* B.4. Isotropic-Leaky-ReLU results appear more clustered, but don't present as discretised like the aligned representational rays in standard Leaky-ReLU. The slight alignments in isotropic examples are thought to be a statistical result, discussed in *App.* A.2, and this is corroborated by further results in *App.* B.4, which mitigate the statistical phenomenon. Hence, one can conclude that standard Leaky-ReLU also induces a quantisation bias, whereas isotropic Leaky-ReLU does not.

The alteration to Leaky-ReLU ensured that these effects aren't trivially due to bounding on the backwards pass either, since Tanh-like functions may produce geometric vanishing gradients in various shapes corresponding to forward-pass saturation. This is not the case for Leaky-ReLU, which is non-saturating, like Tanh, and non-dying, like ReLU. Hence, Leaky-ReLU differs in almost every property from tanh-like functions, besides the shared symmetry construction. This indicates that the symmetry group is responsible for such quantising effects on representations, and each carries an associated inductive bias. This is considered a significant development in representational geometry, elevating the importance of algebraic symmetries as a subject of study.

The appendices present a multitude of additional results for autoencoders of varying widths, depths, datasets, input-output normalisations, latent layer and activation-function dependencies. All of these indicate a similar general tendency towards task-agnostic structure in representations arising from the choice of functional form, particularly the algebraic symmetry involved. Moreover, they also suggest that geometries in representational superposition may be more likely a result of symmetry than the orthogonal to antipodal to Thomson geometries, which are suggested to balance representation capacity and interference (Elhage et al., 2022). This is because hyperoctahedral tanh forms continue to demonstrate orthogonal arrangements first, yet Leaky-ReLU permutation-only forms move immediately to antipodal arrangements *for the same task*. This would encourage a reconsideration of function-derived symmetries as the primary causal mode for the particular geometries produced, when balancing interference and representational capacity.

It is also established that discretisation can be contingent on the functional form's maximal algebraic symmetries. Across all results, these maximal symmetries are shown to be good predictors of the emergent task-agnostic structure in representations. The establishment of these links suggests these functional *forms* are not benign in their effect and can induce significant, unintended biases into networks indicative of their respective maximal symmetries. This initiates and further encourages the identification and documentation of these influences and their extent.

These results show that considering maximal symmetries at the primitive scale may be important, and motivate a reevaluation of the contemporary functional form of primitives in general. These result observations collectively reframe the foundational origin of discrete structure in representations and the assumptions therein. Leveraging such properties could be highly advantageous.

## 5    Conclusion

In this series of experiments, evidence is provided that functional form choices, namely anisotropic activation functions, can induce a representational collapse in latent layers. This establishes a causal mechanism for how task-agnostic structures can emerge in representations due to model choices, rather than emerging from deep learning more fundamentally.

In particular, this structure appears indicative of the maximal symmetry group to which the function algebraically belongs. This is achieved through comparisons of various activation functions with differing maximal symmetries, indicating that symmetry is a good primary predictor rather than solely the specifics of the chosen function. This empirically motivates a thorough investigation of all primitives in terms of their resultant inductive biases when interacting with general networks, previously encouraged in Bird (2025a).

Tanh-like activation functions are chosen in two ways. The first is when their form is defined through maximally satisfying a permutation and sign-flip algebraic equivariance of the standard basis, termed the hyperoctahedral group. This function definition results in representational clusters about standard distinguished directions, which accumulate through training, progressively approximating a discrete direction. Whereas, functional forms which maximally abide by an orthogonal algebraic equivariance produce apparent smooth continua of representations. This indicates that the hyperoctahedral *discrete* group is inducing an approximately-discrete structure to emerge in representation space, implying the functional form is acting as a quantising bias on the representations. This also demonstrates that the prediction that isotropic functional forms eliminate task-agnostic discretising structure is validated. A similar result is found for algebraic permutation symmetry standard Leaky-ReLU, two differing hyperoctahedral and an orthogonal variation of Leaky-ReLU. These also produce a quantising bias characteristic of the discrete groups, whereas isotropic Leaky-ReLU tends to produce smooth continua. This reinforces that the algebraic symmetry is a good predictor of the inductive bias besides solely function specifics, since Tanh-like and Leaky-ReLU-like functions differ substantially in analytical qualities. This empirically validates an earlier hypothesis (Bird, 2025a). Additionally, these inductive biases seem broadly consistent across specific instances of groups, such as $O(18)$, $O(24)$, etc., indicating that the overall group family, e.g. orthogonal, can be a good categorical label for these biases.

Moreover, these phenomena are demonstrated in autoencoders, where one might not typically expect a representational collapse to be beneficial due to information loss under such quantisation, which would further reduce information available for accurate reconstruction. This provides secondary heuristic evidence of the task-agnostic nature of this structure. This collapse persists in circumstances where the activation function does not precede the measured latent layer, so it is non-trivial in cause, such as from forward-pass geometrical boundaries.

Isotropic primitives are suggested to remove structural biases; this does *not* then imply that the representations must be isotropic, they merely do not tend to produce form-induced task-agnostic aligned structure to the distinguished directions. However, in the datasets studied, the isotropic networks did *tend* to form representations which smoothly filled the representation space; consequently, enabling a stronger conviction for the conclusions reached. Effectively, this was an incidentally ideal scenario for determining the validity of the hypotheses. Additionally, it is crucial to consider only the *tendency* of models to produce task-agnostic

induced structure. Instances where other arrangements emerge do naturally arise, and this motivated the ensemble plots of repetitions to clarify.

Consequently, the empirical results strongly support the prediction that *choices in activation function form are not benign, but impart unintended inductive biases corresponding to their symmetry properties, which are sufficient in many cases to strongly influence representations*, confirming a prior prediction. The phenomenon is expected to be systematic for general anisotropic primitive forms, and follow-up studies can investigate this.

The empirical evidence thus far strongly suggests the link between algebraic symmetries and inductive biases on representations. If such a finding holds over further analyses and is determined to be generally valid over a broad scope of primitives and models, then one might conjecture that *Algebraic Symmetries of Functions Utilised in Models Tend to Propagate Their Geometry Into Emergent Representational Structures Through Training* as a potential principle for representational geometry. Despite the evidence presented in this paper, this general statement remains a tentative, speculative, conjecture at the current time, requiring substantial further investigation. Refinement of this statement and detailed mathematical modelling can continue to be evaluated. For example, this may not be expected to hold where the symmetry cannot 'act' on the network, such as in the case of algebraic end-to-end symmetries (Bronstein et al., 2021) which use it to transform representations with the data structure — unless this is reframed as a side-effect of such an inductive bias.

Additionally, combinations of network primitives are expected to exert influence on the representations of larger models, potentially in an interacting, non-trivial hierarchical manner. Understanding this is a pivotal goal of Taxonomic Deep Learning's approach (Bird, 2025a). This is partially demonstrated in the deeper models discussed in the appendices. Yet, overall, these potentially conflating and obscuring influences motivated the first-principles, minimalistic, and ablation-study approach undertaken in this paper, which isolates the cause to individual primitives in otherwise discretising bias-free autoencoders, as opposed to larger models.

This work also corroborates the findings from the Spotlight Resonance Method study and provides a more fundamental causal framework for its observed alignment phenomenon. Coupled with the findings of this prior study, this work suggests a function-driven, symmetry-based causal mechanism for the emergence of several interpretability phenomena, suggesting that they are predicated upon the contemporary anisotropic functional forms. This representational response to anisotropic functional forms appears to explain the downstream emergence of grandmother neurons, generally discrete linear features, axis-alignment, and the *representational* superposition of discrete features, amongst other phenomena. Hence, axis-alignment and grandmother neurons are shown to be phenomena stimulated by design choices, rather than being spontaneous and fundamental to deep learning as a paradigm. Whilst the other phenomena appears to be strongly dependent upon such choices. Overall, this strongly supports the emergence of axis-aligned findings that are frequently observed (Bau et al., 2017; Olah et al., 2019) and suggests that structural asymmetry in distributions can emerge, particularly from certain functional forms. This is considered as opposing evidence for representational uniformity (Szegedy et al., 2014) or spontaneous production of axis-alignment. Together, these results may offer explanatory mechanisms to generalise to further representational geometry and interpretability phenomena.

The isotropic deep learning paper also predicts that this phenomenon extends to semantics. These results demonstrate the formation of approximately discrete representations, which, when combined with the spotlight resonance method's attribution of semantics to these representations, are considered to validate the semantic prediction as well. An end-to-end analysis could also be undertaken in future work. This direct mode for the formation of grandmother neurons is currently limited to deep learning and does not establish an empirical link to biological systems. Nevertheless, it could be used to formulate an interdisciplinary hypothesis.

In addition, the Privileged-Plane Projective method offers an alternative tool for determining alignment phenomena in representational spaces, while retaining both angular and radial information. This is beneficial where a visualisation of the complicated geometry of representations is desirable. The prior SRM may still be better suited to statistical analysis techniques, although it comes at the cost of reduced angular fidelity and no radial information. If a suitably discerning quantitative statistical metrics can be developed, they may be an advantageous future direction for supporting the conclusions drawn. One could attempt to construct a hybrid methodology that incorporates the benefits and mitigates the shortcomings of each method in a joint, visualisable, and statistical tool. Additionally, once the induced geometry is better

understood, one could derive suitable distribution estimates for anisotropic and isotropic representations, enabling statistical measures for further insight into the geometry. At this time, predicting such distributions is not clear, as the specific positioning of clusters depends on the function in use. However, their overall organisation remains functional-form induced. This would also require a quantity to determine the angular 'usefulness' of the function. This may reinforce conclusions, but will need follow-up work to assess the appropriate implementation of such a joint approach. The visualisations are still considered sufficient for the conclusions drawn, given the clear, distinctive, and repeated occurrences of the characteristic symmetry-organised signatures across a range of setups. However, metrics may further strengthen the certainty of the interpretations if an appropriately rigorous methodology can be developed to discern these signatures numerically.

The presence of discrete-like structure is also suggested to be often circumstantially pathological (Bird, 2025a). Preliminary results support this statement for autoencoding networks. Isotropic formulations are shown to enable the systematic elimination of these biases from a network, as the autoencoders tested featured entirely isotropic primitives, except for the activation functions. This establishes isotropies' inductive bias over a wider variety of primitives; follow-up can extend this for anisotropic networks by selectively reintroducing permutation-derived functions in other primitives.

Regarding informative future directions arising from this study: Establishing whether this phenomenon is systematic across other primitives and the hierarchical influence may enable a predictive approach to function-driven induced biases in deeper, general models. Gradient flow analysis will likely elucidate more details on how forms produce representational inductive biases, and their link to taxonomic-symmetry group structures, since the influence appears to persist non-trivially. Moreover, determining the emergent inductive biases of alternative symmetry-defined forms is speculated to be an informative avenue of research, which, when suitably developed, could be advantageously leveraged. Additionally, this will indicate whether the maximal symmetry continues to be a good predictor of its emergent inductive biases and whether representational structure always follows from the symmetry group's structure. Other speculated pathological modes still require validation, such as whether the quantisation of anisotropic representations may produce untrained and unpredictable embedding-free regions, resulting in a particular susceptibility to adversarial attacks.

Additionally, this may suggest that interpretable models and AI safety are especially tractable due to anisotropic inductive biases. Thus, design choices between anisotropy, isotropy, quasi-isotropy (Bird, 2025a), or other definitions may allow tuning of models to balance these behaviours with representation capacity and performance — especially if the pathological nature is further validated. This provides a framework and implementable tunable mode, which earlier work encourages for greater interpretability and safety (Elhage et al., 2022). However, it would also suggest that representations generated from isotropic implementations currently act in opposition to the assumptions of many interpretability approaches. Heuristically, this suggests that isotropic networks yield more unconstrained representations, which may be more natural or inherent to deep learning structures, as they are free from the human-imposed task-agnostic discretising structure imposed by anisotropy. Yet, it is also often this discretising structure which is made interpretable when understanding AI mechanisms. Hence, bridging such an interpretability tool gap may be beneficial, and the PPP method offers a starting point.

The appendices present further results regarding the dependence on depth, width, dataset, activation functions and latent layer of study, all of which are found to be consistent with the validation of the hypothesis. It also includes a novel permutation and even-sign-flipped Leaky-ReLU alternative, which may be beneficial. Additionally, a GitHub link is provided for code implementations of the Privileged-Plane Projective Method.

Overall, this implicates discrete-permutation defined forms of activation functions, as stimulating a task-agnostic alignment about the standard basis. This contrasts with continuous-orthogonal defined forms, which do not. This indicates that forms carry an inductive bias for representations, which appears to be indicative of the maximal symmetry group structure of the form. Activation functions are demonstrated as a direct causal mechanism for quantised representations, and several downstream emergent phenomena are established through connection to prior work (Bird, 2025b). Hence, this approach has established a causal origin for this interpretable structure and several interpretability phenomena.

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

# A Statistical-Geometrical Artefacts

The Privileged-Plane Projective method can produce some minor statistical artefacts inherent in its construction. Notably, these artefacts can arise in the plots, which may be misleading without suitable controls. These occur through two modes, which tend to act in opposing ways. The first is the absence of a regression-to-the-mean phenomenon over privileged directions, which tends to under- or overrepresent projections in these directions due to greater variability. The second concerns how spontaneous symmetry breaking produces a slight tendency for anti-aligned representations to form under the projections of this method. The first phenomenon is particularly subtle in its effect and inherent to the methodology, yet it can noticeably persist throughout all results. It applies equally to anisotropic and isotropic networks, which, alongside its almost negligible subtlety, does not counteract any conclusions reached through the study — though it causally explains slight artefactual structure.

The second modality is more significant, but can be mitigated through appropriate normalisation, which is employed through all appendix results to remove the effect. Normalisation was not included in the main body, due to its slight contrivance; however, any stochastic pseudo-structure is inconsistent with the distinct opposing structures observed to emerge *through* training, which remain solely attributable to the functional forms exchanged. This latter artefactual structure also applies equally to both anisotropic and isotropic networks, further corroborating its lack of implication on the comparative conclusions reached. In addition, this is further mitigated by the breadth of results and numerous independent, repeated observations per plot, where true structure was repeatedly observed in distinct ways.

The following controls will demonstrate that these are pseudo-structures emerging from statistical artefacts, as opposed to true representational alignments.

## A.1   Overrepresentation Artefacts

Fundamentally, the proposed method produces two-dimensional slices, projected from thresholded volumes, over a much higher-dimensional representational space. These high-dimensional spaces have a very large hypervolume; therefore, the projected planes typically do not intersect.

However, particularly for a standard basis, these planes do *repeatedly* intersect along standard basis vector directions. Due to the orthogonality of the standard basis, these overlaps also systematically accumulate in the vertical and horizontal regions of the plot. This is a geometrical and statistical consequence of such standard projections and the conglomeration of data over multiple planes. This means that the same representations within certain angles about distinguished directions get repeatedly projected into the PPP slice in these regions. If distinguished directions have a slight under- or over-density stochastically in these regions, then their effect can be amplified. In comparison, other areas in the conglomerate plot are measured over many differing hypervolumes of the larger space. This tends to suppress fluctuations through a regression-to-the-mean phenomenon, due to differing effective sample sizes.

This is believed to be an inescapable and inherent statistical phenomenon when conglomerating multiple planes derived from standard distinguished directions. This can produce misleading results stochastically, but the effects were mitigated in the results by demonstrating a breadth of various networks and repeats, allowing a more holistic interpretation and conclusion.

In this section, this geometrical phenomenon will be demonstrated. This can be achieved by performing the PPP method over random samples drawn from a standard multivariate normal distribution, whilst also choosing random orthogonal bases from which to compute PPP uniformly distributed over the $\mathfrak{so}\,(n)$ manifold. The repeated occurrence of slight clusters over repeatedly chosen random bases is indicative of a statistical artefact, as the underlying distribution is known to be probabilistically invariant over the orthogonal group, and similarly for the studied basis. A statistical model of such clusters may be an advantageous pursuit in future, particularly for the similar SRM method.

In *Fig.* 4, one can see that clusters about the distinguished directions are frequent despite the known approximate isotropic uniformity of the data. This is indicative of a statistical artefact which may occur, particularly when the PPP threshold provides fewer samples. This discrepancy between anti-aligned and

aligned over- and under-representations is not resolved by averaging over multiple plots or increasing the sample size; however, the disparity between regions will decrease as they converge to the mean at differing rates.

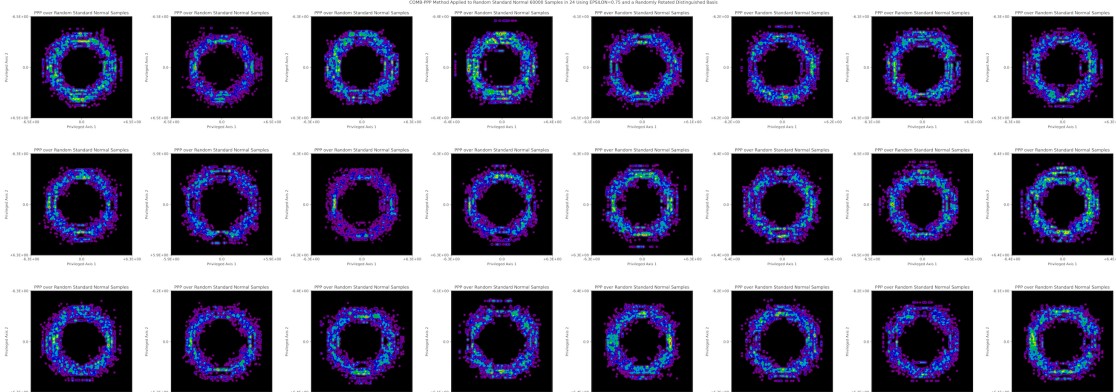

Figure 4: Displays the combination PPP-method applied to 60000 samples (identical to CIFAR training set size) drawn from a multivariate normal and applied using a random orthogonal basis. A value of $\epsilon = 0.75$ and 24 neurons was chosen to make them comparable to all other results. Each individual plot shows a different random initialisation of both samples and basis. Zero-indexed by (row, column), one can observe axis-aligned underrepresentations, particularly in plots $(1, 1)$, $(2, 6)$, and overrepresentations, particularly in $(1, 4)$, $(2, 4)$, $(2, 7)$, and $(1, 0)$. Yet, it is known that higher-dimensional samples are approximately angularly uniform; therefore, this is a geometrical and statistical artefact resulting from the projection method collated over multiple planes.

Finally, one might note that straight lines occur in the plot, especially clear at larger absolute values of the projection. These may suggest a rotational asymmetry, but actually represent a single representation which aligns closely to a distinguished vector. Due to this proximity, it is reprojected along multiple privileged planes, accounting for the occurrence of the same absolute value along one axis, producing the observed straight lines.

Overall, these phenomena are slight, and the weak density peaks are wholly insufficient to produce the more extreme alignments repeatedly measured in anisotropic networks. Hence, they are deemed negligible in effect and do not change conclusions drawn regarding quantisation due to functional form inductive biases. Moreover, the same phenomenon will occur in an equal stochastic manner between both results, so it does not account for the stark differences observed between anisotropic and isotropic networks.

However, awareness of them is essential to avoid misinterpreting the presence of true structure in more rotationally uniform distributions. This phenomenon is believed to wholly account for slight alignments (overrepresentations) or slight anti-alignments (underrepresentations) that are otherwise unexpected, such as in isotropic plots. This phenomenon becomes increasingly apparent in higher-dimensional spaces due to the typically sparser density of isotropic distributions, resulting in fewer samples provided by the PPP method. Additionally, permutation-PPP has a higher susceptibility due to increased overlap over these directions, which constitutes the preference for combination-PPP in these results. Additionally, combination-PPP can indicate remaining asymmetries in distribution, which permutation-PPP symmetrises out.

Finally, statistical analysis of this phenomenon is complicated due to the dependence on the basis, $\epsilon$-threshold, and closed-form volumes for hyperspherical segments and general hypervolumes of such shapes. Although it would be a prudent future direction if found to be statistically tractable, this is currently a challenge.

## A.2 Spontaneous Symmetry Breaking Artefacts

An additional statistical artefact can emerge in the PPP plot, dependent upon the dataset's subspace. This was initially observed in freshly initialised plots, which prompted further study.

Often, one might expect representations to be isotropically distributed on a fresh initialisation of the model, especially if a left- and right-flavour probabilistic invariant initialiser (Bird, 2025a) is employed. However, spontaneous symmetry breaking upon choosing a particular initialisation can produce artefactual geometry due to a dataset's subspace.

Since biases are initialised to zero, these initialisations produce linear, not affine, maps[9]. If a dataset is strongly anisotropic, such as being situated in a $[0,1]^{28\times28}$ subspace, then when it is randomly embedded, it has a larger chance that specific directions are measured to have an anisotropy. For example, where the $\vec{1}$ becomes embedded will be angularly denser in representations than where any $\hat{e}_i$ is embedded. These embedded representational anisotropies often occur anti-aligned due to the small chance of spontaneously aligning precisely with the distinguished directions for the plot.

This is displayed in *Fig.* 5. This plot uses data which is drawn uniformly from the $[0,1]^{32\times32\times3}$ space, which is then embedded in a $\mathbb{R}^{24}$, intended to mimic similar conditions to the CIFAR plots, using a linear transform initialised using a random standard multivariate normal. One can see a clear tendency towards anti-aligned artefactual anisotropies resulting from the spontaneous symmetry-broken embedding and the dataset's bounds. This may give the false impression of model-imposed anisotropy, which requires mitigation.

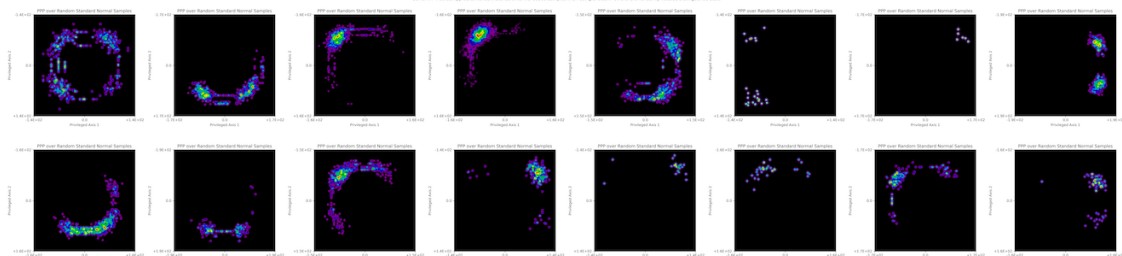

Figure 5: Each individual plot displays an independent instance of a combination-PPP $\epsilon = 0.75$ plot, of 60000 uniformly drawn samples from $[0,1]^{32\times32\times3}$, which are embedded in $\mathbb{R}^24$ using a linear map randomly drawn from a standard normal. Additionally, a random orthogonal distinguished basis was used for PPP, which was freshly sampled for each plot. One can observe a tendency towards strongly anti-aligned clusters due to the geometry of the stochastic dataset, map and PPP method. These artefactual clustered structures can be a more significant issue for interpreting PPP plots; therefore, the underlying dataset warrants normalisation efforts for better analysis and stronger conviction in conclusions. Additionally, these clustering artefacts remain inconsistent with the emergence of the highly concentrated narrow beams, which emerge through training in prior results.

The expectation is that if the dataset distribution is rescaled, it will more closely approximate a standard normal distribution while retaining its dataset's relative structure. Such distributions may reduce this artefactual structure in plots. Particularly from the overrepresented $\vec{1}$ direction, in $[0,1]^n$ bounded samples relative to the origin.

This was achieved by element-wise normalisation of samples by statistics obtained across the entire dataset. For a dataset of size $\mathbb{R}^{\text{samples}\times\text{dimension}}$, a mean value was calculated $\mu \in \mathbb{R}^{\text{dimension}}$, likewise for the standard deviation. Each sample was then consistently normalised using these tensors, elementwise subtraction of the mean and division by the standard deviation. Where the standard deviation was zero, that element was instead divided by 1 without consequence. Overall, the representational effect on manifold structure were considered negligible $f_{\text{norm}} : \mathbb{R}^{\text{samples}\times\text{dimension}} \to \mathcal{S}^{\text{samples}-2} \times \mathbb{R}^{\text{dimension}} \hookrightarrow \mathbb{R}^{\text{samples}\times\text{dimension}}$. Similar may be achieved by monotonically remapping the hypercube to an origin-centred hypersphere. Normalising by covariance was also considered but found to be infeasible due to the production of singular matrices, which prevented the calculation of inverses; therefore, only standard variances were normalised.

The plot of *Fig.* 6 is constructed identically to *Fig.* 5, yet the samples are normalised first.

---

[9]The success in this form of bias initialisation could be reconsidered and freshly explored under the lack of symmetry breaking perspective developed under the wider taxonomy (Bird, 2025a).

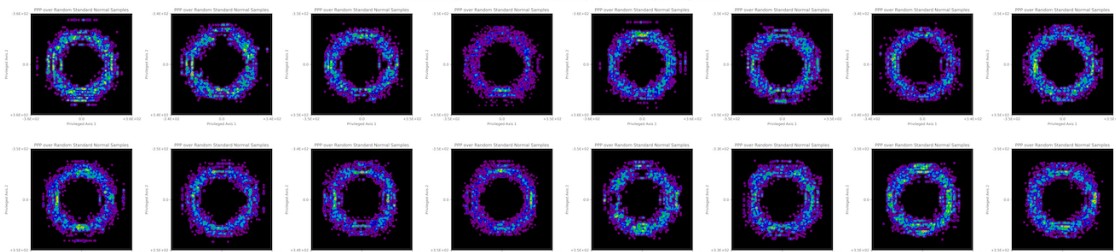

Figure 6: Each individual plot displays an independent instance of a combination-PPP $\epsilon = 0.75$ plot, of 60000 uniformly drawn samples from $[0, 1]^{32 \times 32 \times 3}$, *which are then normalised as described*, following this the samples are embedded in $\mathbb{R}^{24}$ using a linear map randomly drawn from a standard normal. Additionally, a random orthogonal distinguished basis was used for PPP, which was freshly sampled for each plot. This plot demonstrates that samples are much more uniformly distributed rotationally when using the PPP method, strongly mitigating the discussed statistical artefacts. Despite the normalisation, infrequently, due to the embedded hypercuboidal distribution, slight artefactual structures persist. Additionally, one can observe the reemergence of slight clusterings resulting from the phenomenon discussed in *App.* A.1.

Despite their remaining hypercuboidal bounding, these results are much more comparable to a known embedded isotropic distribution, displayed in *Fig.* 7, which is similarly constructed, but the original samples are drawn as a standard multivariate normal over a $\mathbb{R}^{32 \times 32 \times 3}$ space and then embedded in $\mathbb{R}^{24}$ using a similarly drawn linear map. Hence, this justifies the normalisation employed in many of the appendix results.

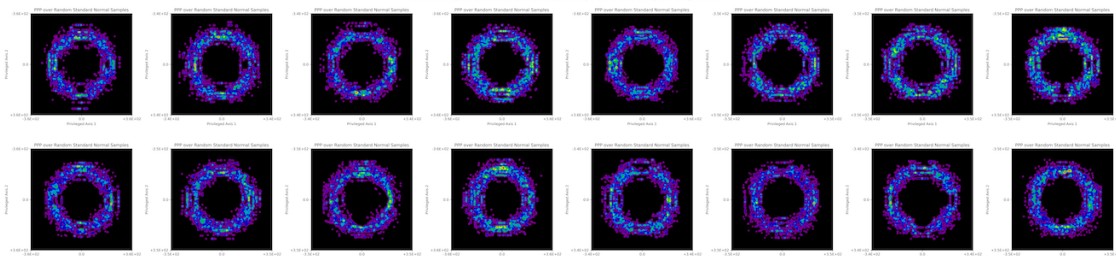

Figure 7: Each individual plot displays an independent instance of a combination-PPP $\epsilon = 0.75$ plot, of 60000 uniformly drawn samples from $[0, 1]^{32 \times 32 \times 3}$, *which are then normalised as described*, following this the samples are embedded in $\mathbb{R}^{24}$ using a linear map randomly drawn from a standard normal. Additionally, a random orthogonal distinguished basis was used for PPP, which was freshly sampled for each plot. One can continue to observe the slight clusterings resulting from the phenomenon discussed in *App.* A.1.

Overall, this statistical phenomenon can be more significant than the one discussed in *App.* A.1, and hence warrants normalisation efforts to mitigate such artefacts and establish better certainty in conclusions.

This approach constitutes the majority of the results in the appendices. However, such normalisation was also perceived as an unusual imposition on the data, so the results of the main body do not display this normalisation to ensure they do not appear as contrived. Moreover, the phenomena which are concluded to occur from functional forms persist whether normalisation is used or not, though normalisation aids in a higher certainty for the conclusions drawn.

An observation of a slight impact was noted: networks trained on samples without normalisation were found to tend to produce slightly more concentrated clusters, likely due to the somewhat different training dynamics and optimisation trajectories resulting from the absence of normalisation. Such a difference may be insightful in determining the exact mechanisms through which algebraic bias induces representational structure through optimisation. Despite imparting some effect, the highly localised structures observed were unique to anisotropy; moreover, their concentration and diffusion through training for anisotropy and isotropy,

respectively, are not consistent with any statistical artefacts, but instead remain highly consistent with the conclusions drawn, supported by all appendix results. Since this pseudo-structure effect persists across both anisotropic and isotropic plots in an equal stochastic manner, it cannot account for the distinct difference observed between the ablations, which is therefore solely attributable to the functional form change. Hence, it does not alter the validity of the conclusions drawn, but instead offers further insight into altered training dynamics due to normalisation.

Overall, in all cases, statistical pseudo-structure is an insufficient explanatory mode for the comparative observations across the ablation structure. Hence, the induced bias by functional form remains the only consistent explanatory mode for the results.

# B    Extra Tests

The prior results exemplified how functional form choices influence representations. In this section, comparisons are made between autoencoders of varying depths, widths, dataset reconstructions, and activation functions, as well as how representations evolve through sequential latent layers. This highlights nuances of their structure, including indications of representational capacity, such as how isotropic networks appear to embed across the entire representational volume. All results undertaken were normalised as described.

Due to space restrictions, these image sizes are noticeably compressed, as otherwise it would have made the addition of many further results infeasible to include within this document. However, headings and axes titles remain consistent with prior figures, and all notable representational details across all plots remain visible. The examples shown are representative of this broader array of results.

## B.1    Dependence on Network Width

Varying the autoencoder's layer width across all hidden layers tends to produce distinct changes in the representations.

Typically, one might expect the representations to become sparser as network width increases. This may be expected as hypervolume grows exponentially with width. Hence, for the fixed sample size of 60000 for CIFAR's training set, the density of representations might be expected to decrease; therefore, fewer representations may appear within PPP's projective hypervolume.

However, crucially, this was observed only in isotropic networks; whereas, anisotropic networks tended to produce denser distributions concentrated about the distinguished directions. This can be seen through plots *Fig.* 8 (18 neurons), *Fig.* 9 (24 neurons), and *Fig.* 10 (32 neurons). This was a frequently observed behaviour over many networks.

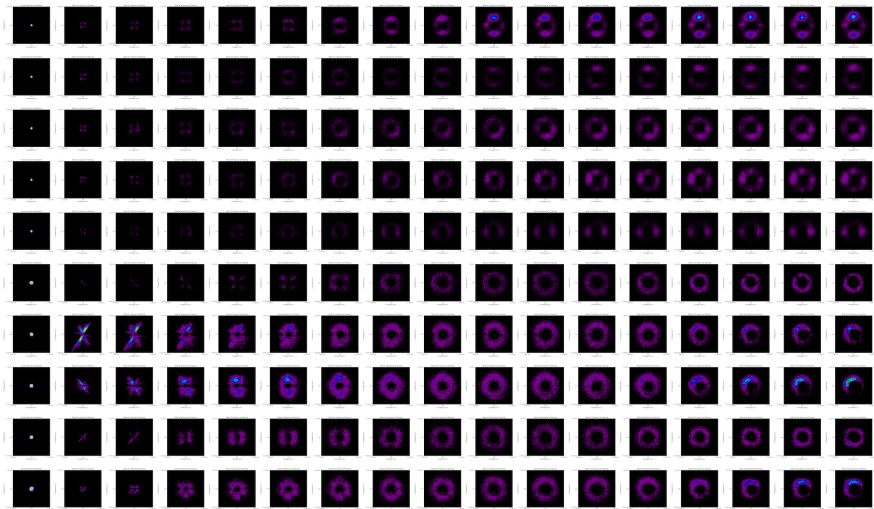

Figure 8: This network shows the PPP method applied over networks of 0 hidden layer blocks, 18 neurons and using input-output normalisation as described in *Apps.* A and E. The top five rows consist of the standard-tanh autoencoders, whereas the bottom five rows utilise isotropic-tanh. Everything else is equal between plots. Training proceeds rightwards over columns, in an identical fashion to *Fig.* 1. Overall, this plot shows that anisotropy tends to result in the emergence of axis-aligned representation clusters, including several instances of apparent antipodal alignments. Whilst isotropic examples do not produce such a structure and tend to be more evenly distributed. It can be seen that under normalisation, there is a tendency for more diffuse clusters in anisotropic results, particularly evident in narrower networks.

This suggests that isotropic networks tend to occupy the entire representation space following a bottleneck, whereas anisotropic networks tend to cluster primarily along the axes. This is preliminary evidence that

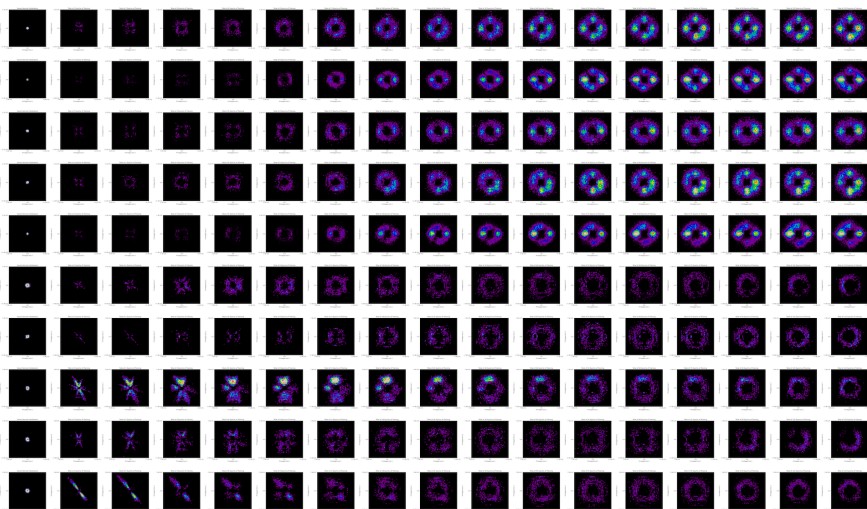

Figure 9: Similarly this network demonstrates identical setup to *Fig.* 8; however, it depicts all phenomena for networks with a width of 24 neurons. The production of axis-aligned structures in anisotropic networks is considerably more distinct, indicating that representations produce denser clusters along the distinguished directions as the width increases. Isotropic networks are noticeably sparser in representations, indicating that they are more smoothly distributed over the considerably larger angular surface in 24 dimensions as opposed to 18 dimensions. Due to this sparsity, a higher prevalence of *App.* A.1 occurs, likely resulting in the difference in row 8. Normalisation appears to again produce a more diffuse fall-off in the alignments compared to the sharper clusters in its absence.

isotropic networks tend towards a continuous distributive code, whereas anisotropic networks tend to approximate a discrete local code. Correspondingly, this would suggest that anisotropic networks are producing dense clusters in approximately $2n$ directions for $n$-neuron width, whereas isotropic networks are distributing representations over $\propto e^n$ hypervolume.

This might be limited to autoencoding bottlenecks, yet it can be hypothesised that a subspace would be more evenly occupied for isotropy, whenever a representational space is increased in dimension, too. One might investigate whether functions which include non-monotonic foldings may then gradually fill such a volume, approximating space-filling curve behaviour. In either case, this is indicative of a significantly increased representation volume and a very different approach by isotropic networks to balancing the interference-capacity tradeoffs; finding ways to leverage such encoding may be highly beneficial — perhaps particularly in generative modelling. Additionally, for very narrow networks, such as 12 neurons, some results showed anomalously few representations produced by the PPP method in anisotropic networks, even sparser than isotropy. This is indicative of a repulsion from the distinguished directions rather than attraction to them. The latter is typically observed in wider networks. **This is speculated to be likely due to the production of a dense-coding-like representation, which the network may produce to inflate representational capacity in particularly narrow architectures — a network adaptation to strong bottlenecks when also constrained by a form's symmetry.**

Additionally, since a similar trend in representation basis was observed independent of width, it suggests that categorising representational biases by families of groups, such as orthogonal, as opposed to individual instances $O(8)$, $O(12)$, etc. Indicates that documentation of various inductive biases may be more generalisable over group families and therefore more tractable to leverage.

Overall, the indication of isotropy producing distributed representation has potentially very significant ramifications for representational capacity and requires further study, especially for architectures which may leverage it. These networks may be able to represent underlying continuous data much more effectively without task-agnostic imposed quantisation. This may be beneficial in preserving data structures in models and indicates that increasing width may compensate for anisotropic's quantised representations. These

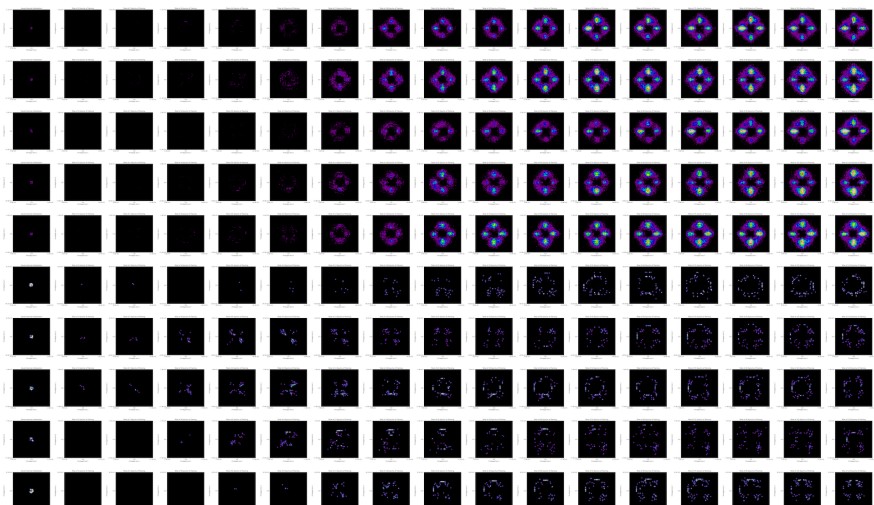

Figure 10: Finally, this network demonstrates an identical setup to *Figs.* 8 and 9; however, it depicts all phenomena for networks with a width of 32 neurons. The anisotropic results appear more sharply peaked about the distinguished directions than *Figs.* 8 and 9, especially if plots are overlaid. The aforementioned fall-off about the strong alignments appears to be shrinking slightly with width. This indicates an even greater density of representations moving towards these directions through training. All plots are the $0^{\text{th}}$ latent-layer, without preceding activation functions, so this is non-trivial in effect. Furthermore, the contrast in sparsity between anisotropic and isotropic networks is very distinct, indicating the isotropic network's tendency to distribute evenly across the full representation space. Additionally, all results demonstrate that the scale of representations increases as training progresses.

continuous representations by isotropic networks may also aid in tasks that require modelling of a distribution, such as for generative applications.

## B.2 Dependence on Autoencoder Depth

In many results, the dependence on depth is less conclusive. There appears to frequently be a trend where deeper networks produce more complicated structures in representations, particularly in isotropic networks, but these observations are less clear in trend. Additionally, sometimes in deeper networks, the $0^{th}$ layer results demonstrated less anisotropic clustering than shallower networks; yet, clustering still emerged deeper into latent layers, making disentanglement of trivial and non-trivial causes difficult in these specific cases.

Results indicating the emergence of more complicated embedding structures are demonstrated in *Figs.* 11, 12 and 13. These results correspond to identical networks that differ only in the number of hidden layer blocks, specifically 1, 2, and 3 blocks, respectively. In all cases, the $1^{st}$ latent layer is measured.

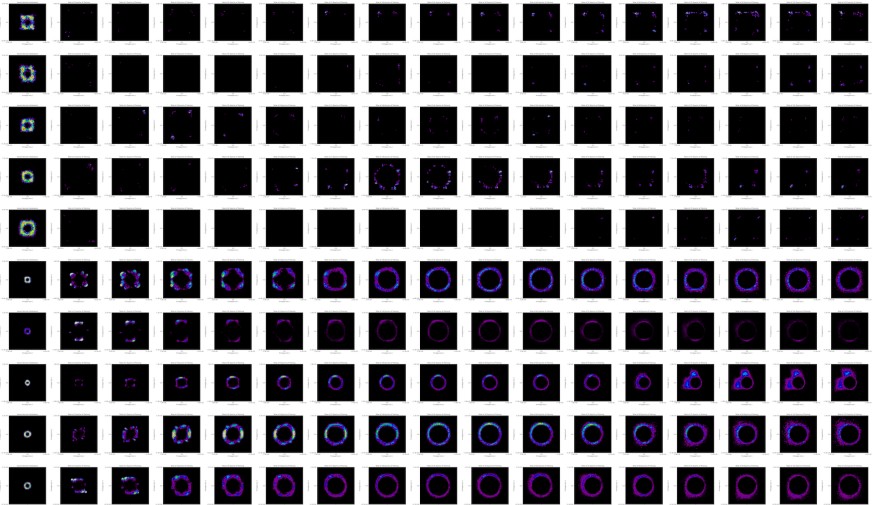

Figure 11: This network shows the PPP method applied over networks of 1 hidden layer block, 18 neurons and using input-output normalisation as described in *Apps.* A and E. The top five rows consist of the standard-tanh autoencoders, whereas the bottom five rows utilise isotropic-tanh. Everything else is equal between plots. Training proceeds rightwards over columns, in identical fashion to *Fig.* 1. It is evident from this plot that the bottom-five isotropic networks tend to produce a smooth, continuous-like distribution of representations. The top-five anisotropic plots indicate the emergence of structure tending to be anti-aligned to the standard basis. This is more sparse, even than the uniform isotropy, likely since many representations are producing narrow anti-aligned beams in the anisotropic case. The PPP method is likely only capturing the more diffuse tail of such a cluster due to the $\epsilon = 0.75$ thresholding, which is significantly more stringent than $2/\sqrt{18 \times 2}$ required to include anti-aligned diagonals. Overall, this suggests the anisotropic networks are producing an anti-aligned structure consistent with a dense-like code.

Overall, these results suggest, as may be expected, that depth produces more nuanced representational clusters indicative of the greater expressibility of deeper networks. Yet, in all cases, comparing anisotropic to isotropic activations indicates a difference in the presence of structure.

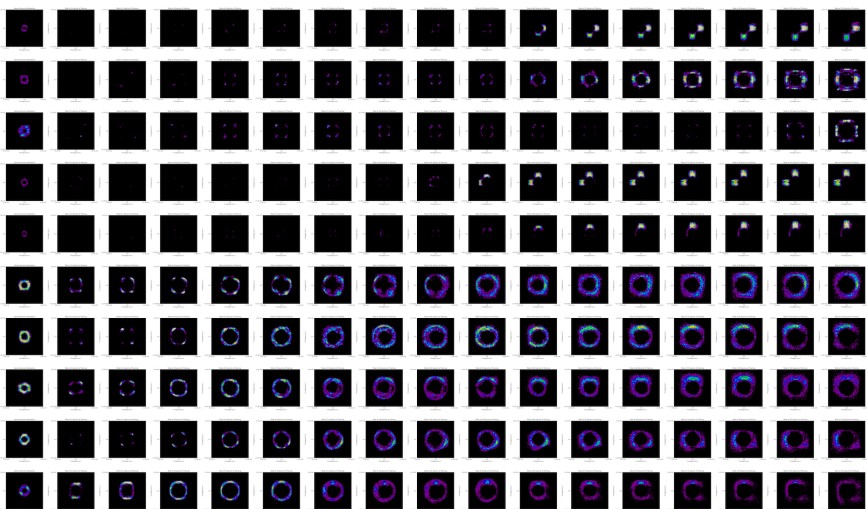

Figure 12: This plot is identical in every way to *Fig.* 11, except for the network comprising of 2 hidden-layer blocks. Anisotropic networks are shown to exhibit a more pronounced axis-aligned structure, frequently comprising of orthogonal directions across all conglomerated planes. This differs from the isotropy plots, which demonstrate a more continuous distribution. In these plots, slightly clearer divergences in dense representations and double-rings are observed compared to the simpler ring distributions in *Fig.* 11. This indicates depth corresponds to more nuanced representations, as may be expected. Rows 2 and 3, for anisotropy, indicate a more complicated structure, which is more diffuse than the typical discrete-like representations, but still demonstrates a clear axis-aligned structure.

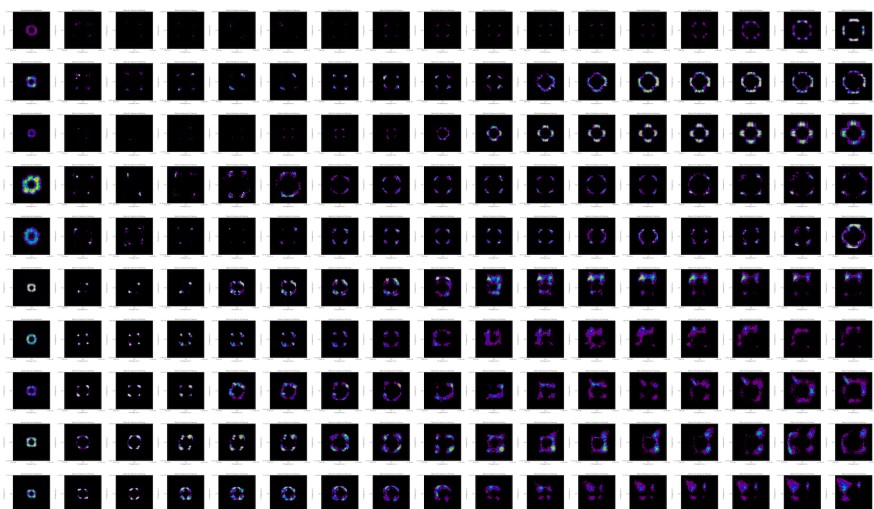

Figure 13: This plot is identical in every way to *Figs.* 11 and 12, except for the network comprising of 3 hidden-layer blocks. The anisotropic structure in the top five rows is more pronounced, with denser regions of representations in anti-aligned and aligned directions. Most interestingly, the isotropic plots show significantly nuanced structures compared to the previous, more ring-like representations. Several denser clusters are shown to form, but in a non-axis-aligned manner. The slightly anti-aligned structure is likely statistical in nature, as indicated by its presence in initialisations.

### B.3 Dependence on Latent Layer of Study

Analysis of different latent layers indicates a tendency of isotropic networks to produce representations that become progressively more continuous in later latent layers, whilst the opposite was found for anisotropy. This is demonstrated in *Figs.* 14, 15, 16, and 17, which indicate the exact same networks, with PPP performed at various latent layers.

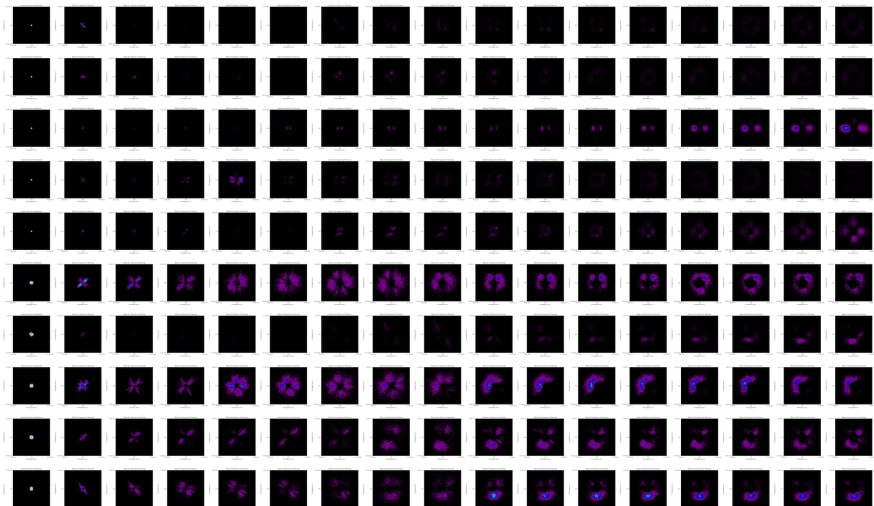

Figure 14: This network shows the PPP method applied over networks of 3 hidden layer block, 18 neurons and using input-output normalisation as described in *Apps.* A and E. PPP was applied to the $0^{th}$ hidden layer. The top five rows consist of the standard-tanh autoencoders, whereas the bottom five rows utilise isotropic-tanh. Everything else is equal between plots. Training proceeds rightwards over columns, in identical fashion to *Fig.* 1. One can observe that representations are absent in anisotropic rows 1, 2, and 4, likely a result of anti-alignment, whereas some axis-aligned structure is present in rows 3 and 5. Isotropic rows demonstrate larger, more continuous representational structures, but also demonstrate clustering such that they are not distributed smoothly over the representation space.

Overall, in many networks tested, anisotropic structure appeared progressively clearer and denser in later latent layers, whilst isotropic networks tended to produce progressively more continuous distributions in later latent layers. This trend was observed across many of the networks studied, but sporadic differences between networks also occurred. This may be due to the increased number of an/isotropies in the forward pass or indirectly through the backwards pass, yet further study may elucidate the exact reason why progressively clearer structure may emerge in later layers.

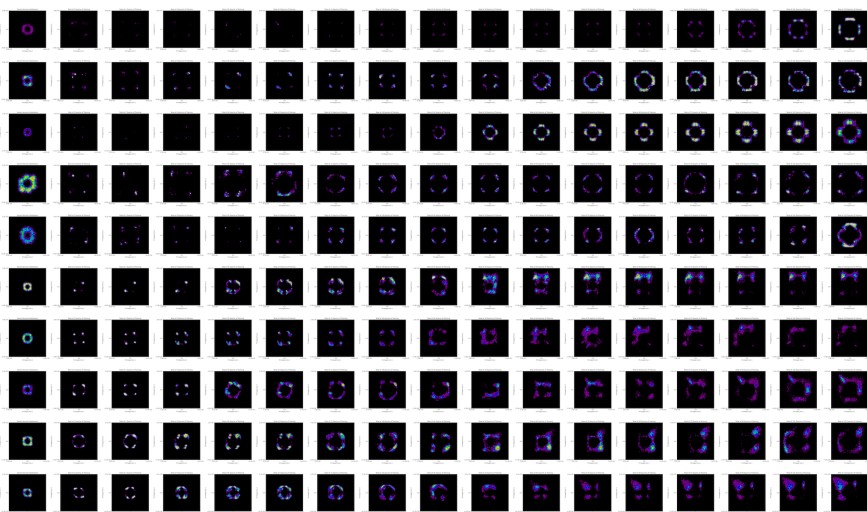

Figure 15: The results are identical to *Fig.* 14, yet demonstrate the PPP method on the 1st latent layer of the three blocks. Each row corresponds to the exact same network as *Fig.* 14. One can observe more distinct alignment structures emerging in all anisotropic rows through training; these structures are slightly more complicated than previous examples, likely due to the deeper network employed. Isotropic networks also demonstrate slightly more diffuse representations than the earlier latent layer, and some divergence consistent with the greater depth of the network.

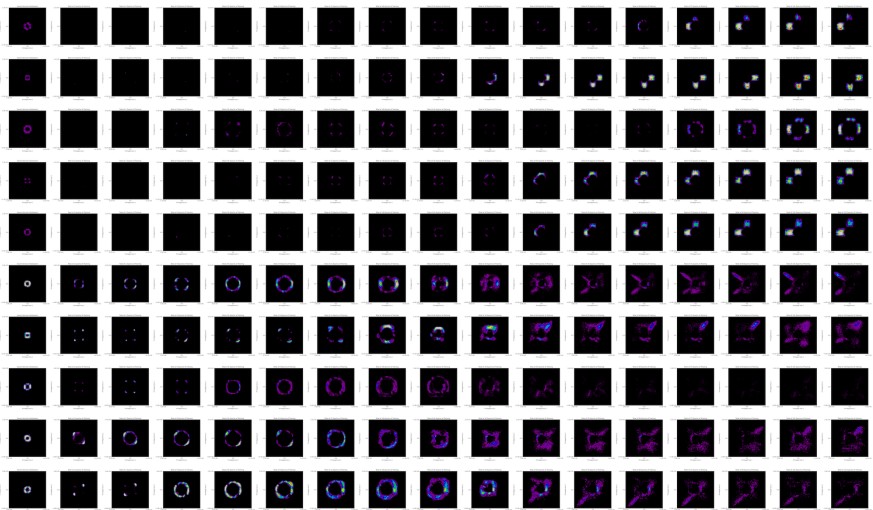

Figure 16: The results are identical to *Fig.* 14, yet demonstrate the PPP method on the 2nd latent layer of the three blocks. Each row corresponds to the exact same network as *Fig.* 14. In the anisotropic rows, even more distinct axis-aligned structures emerge, resulting in approximately discrete representational structures. An increasing number of representations appeared in these discrete directions as training progressed, and a greater amount compared to earlier latent layers in the model. Isotropic networks also tended to form more continuous representations, yet row six in particular demonstrated more quantised directions forming later in training. This indicates that isotropy can still occasionally produce such clusters without a function-driven bias, particularly emerging in deeper networks. The anti-aligned nature is also likely a result of *App.* A.2, which can reemerge when the distribution becomes uneven in deep latent layers, as inter-layer normalisation is not used. Nevertheless, there continues to be a stark difference between the structure present in the anisotropic rows compared to the isotropic rows.

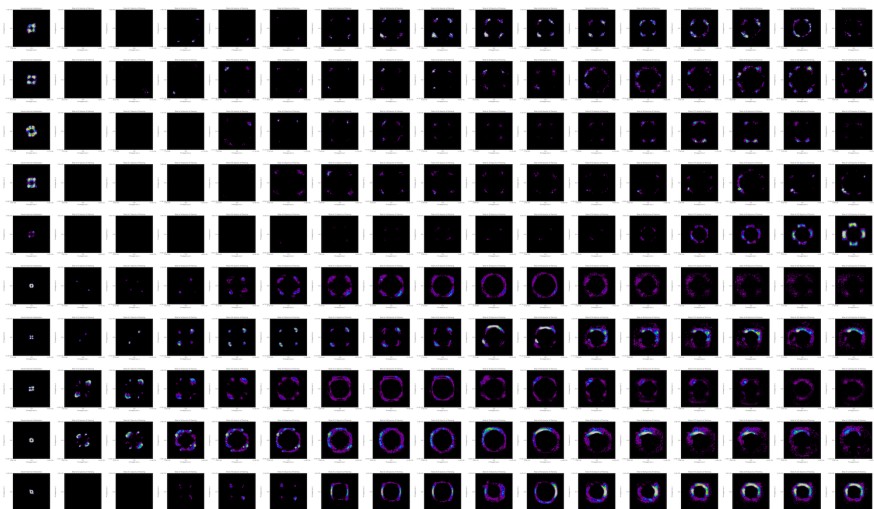

Figure 17: The results are identical to *Fig.* 14, yet demonstrate the PPP method on the 3$^{\rm rd}$ latent layer of the three blocks. Each row corresponds to the exact same network as *Fig.* 14. Anisotropic structure is again present in the top five rows, yet slightly more diffuse than the previous latent layer. This diverges from the previously discussed observed trend regarding the concentration of anisotropic representations over deeper layers. The isotropic examples appear more uniform with only slight clustering, continuing the trend that later latent layers tend to be smoother for isotropic examples.

## B.4 Results for Leaky-ReLU

Except for *Fig.* 3, all comparisons have been concerning Tanh-like functions. These have clearly demonstrated the tendency for task-agnostic structure to occur in anisotropic cases, whilst more continuous representations tend to form in isotropic variants.

Using $0^{\text{th}}$ latent layers, the experiments have also demonstrated that these effects are not trivially due to forward-pass saturation geometry, yet backwards pass vanishing-gradient geometry has not been ruled out, except for *Fig.* 3.

In this section, Leaky-ReLU-like functions will be compared to demonstrate that these quantisation phenomena are not unique to the Tanh-like cases. Moreover, except for an underlying symmetry connection, Leaky-ReLU-like and Tanh-like functions are substantially different analytically. For example, Leaky-ReLU is piecewise linear, so it does not saturate or produce gradients which tend to zero[10]; therefore, geometries due to these can be ruled out. Thus, an investigation using Leaky-ReLU can isolate and establish that the symmetry relations are the likely cause of such a structure, rather than the specifics of the chosen function.

In this section, the standard Leaky-ReLU, shown in *Eqn.* 6, is compared with its isotropic counterpart, shown in *Eqn.* 7 (Bird, 2025a).

$$\mathbf{f}\left(\vec{x}; \{\hat{e}_i\}_{\forall \hat{e}_i}\right) = \sum_{i=1}^{n} \max\left(\alpha \vec{x} \cdot \hat{e}_i, \vec{x} \cdot \hat{e}_i\right) \hat{e}_i \tag{6}$$

$$\mathbf{f}\left(\vec{x}\right) = \begin{cases} \alpha \vec{x} & : \quad \|\vec{x}\| < \varepsilon \\ \vec{x} - (1-\alpha)\,\varepsilon & : \quad \|\vec{x}\| > \varepsilon \end{cases} \tag{7}$$

Namely, standard Leaky-ReLU features an algebraic permutation $(S_n)$ equivariance, whilst isotropic-Leaky ReLU is defined from an algebraic orthogonal $(\mathrm{O}\,(n))$ equivariance.

The permutation symmetry of standard Leaky-ReLU subtly differs from that of standard tanh. Standard tanh is equivariant under both permutation and component sign-flips, termed a hyperoctahedral symmetry $(B_n)$, whereas standard Leaky-ReLU is only equivariant under permutation $(S_n)$. Differences in quantised structure may be indicative of such differences between symmetry groups obeyed by the functional forms.

In all the following results, the experiments continue to use normalisation, as in all appendix results, to mitigate statistical effects discussed in *App.* A. Whereas, an unnormalised case is displayed in *Fig.* 3.

In all cases, $\alpha = 10^{-2}$ is used, as is the default in PyTorch for standard Leaky-ReLU. The radius of the isotropic-ball threshold was set at $\varepsilon = 1$ as a completely arbitrary and non-optimised choice — refinement or parameterisation (especially in a smoother function) of this value may be beneficial in practice.

The following results demonstrate the effect these functions have on representation distribution, followed by commentary on how each representation may be influenced by the algebraic equivariant symmetry through which the function was defined.

To begin, all Leaky-ReLU results also show distinct quantisation in the $0^{\text{th}}$ layer, *Figs.* 18 and 19 are representative of this trend.

It is clear from both examples that the conclusions reached from tanh-like experiments are consistent with those in these Leaky-ReLU-like experiments. There is a clear emergence of structure in activations defined through a discrete permutation symmetry, whereas this task-agnostic structure is not present in the isotropically defined functions. Therefore, the symmetry classification seems to be the predominant cause of such quantisation behaviour within networks. Additionally, both of these figures are produced from PPP operating on the zeroth layer — so the effect is non-trivial.

---

[10]Leaky-ReLU was used as opposed to ReLU to also rule out the zero-gradient 'dead-neuron' geometry as the cause.

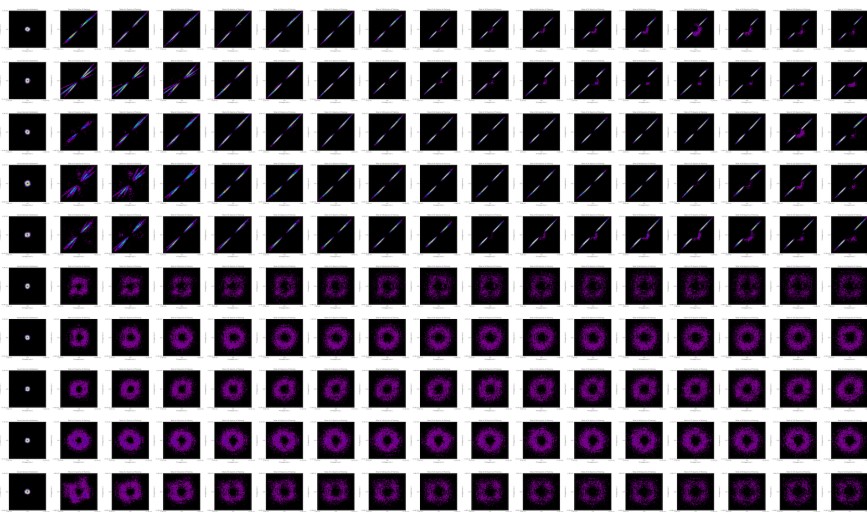

Figure 18: The results of the PPP method applied to the $0^{th}$ latent layer of an autoencoder with 3 hidden layer blocks and width of 18 neurons per block. The top five rows consist of the autoencoder using the standard Leaky-ReLU, while the bottom five rows consist of the same autoencoders using the isotropic Leaky-ReLU. All five of the top five rows demonstrate clear anti-aligned quantisation in representations, predominantly in a single anti-aligned axis. These emerge early in training and become progressively more (approximately) discrete. This contrasts with the isotropic Leaky-ReLU examples, which approach a more continuous, evenly distributed embedding distribution, consistent with previous observations. Additionally, at relatively smaller magnitudes, the Leaky-ReLU results consistently show some deviation from the discrete-like rays; this remains unexplained.

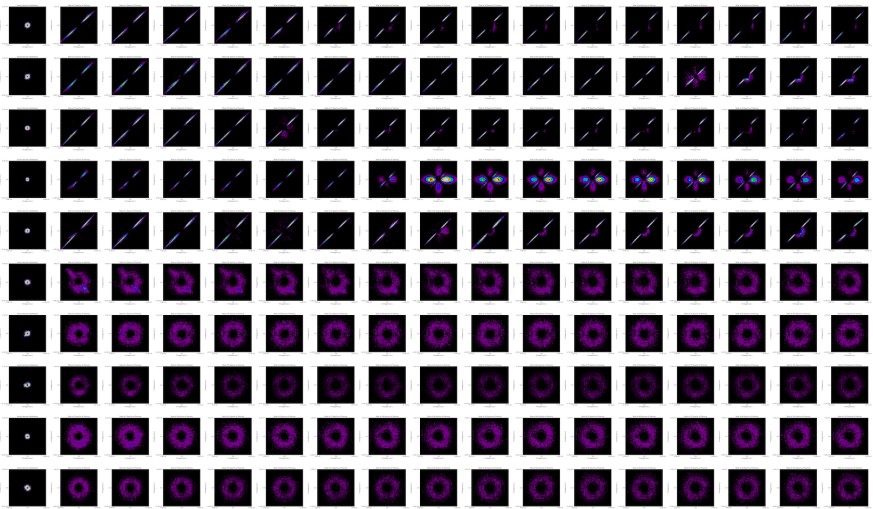

Figure 19: The results of the PPP method applied to the $0^{th}$ latent layer of an autoencoder with 2 hidden layer blocks and width of 18 neurons per block. The top five rows consist of the autoencoder using the standard Leaky-ReLU, while the bottom five rows consist of the same autoencoders using the isotropic Leaky-ReLU. These results are highly consistent with *Fig.* 18, yet the 3rd row also shows the addition of axis-aligned structure.

Additionally, the Leaky-ReLU graphs in all $0^{th}$ layers studied display a slightly differing structure from the structure present in Tanh-like networks. This seems to correspond well to the differing symmetry acting, but its emergence from other analytical differences between the functions is not ruled out.

In particular, the $0^{\text{th}}$ layer standard Leaky-ReLU results always produced two narrow beams arranged antipodally, along a single diagonal. This anti-aligned structure occurs in most cases, and the standard Leaky-ReLU only rarely displays an axis-aligned structure. This single-diagonal may be indicative of the $S_n$ group, which does not include sign flips, unlike tanh's $B_n$ group. Therefore, only a single diagonal may appear populated, as opposed to other diagonal directions, which could result from the various mirrors applicable in the $B_n$ group. This nuanced difference between demonstrations appears to closely follow from the differences between symmetry groups. These differing representations can be considered to support the influence of different symmetry groups on representations, with nuances following in a taxonomic fashion from the group structures. Particularly, the orthants containing $\pm\vec{1}$ are the only non-degenerate orthants produced by the $S_n$ symmetry, and the representations appear to be aligning with these, suggesting the remaining orthants appear not to produce computationally useful maps.

The exact causal mode through which the symmetry group acts can be speculated in the Leaky-ReLU case.

Particularly, Leaky-ReLU effectively uses the condition that the activation is scaled by $\alpha$ if any component of the vector, decomposed along the standard basis, is negative. This results in an orthant of the domain, for which the identity map is applied, whilst the remaining space is scaled by $\alpha$ along various combinations of components. The relative volume of the identity-orthant follows $2^{-n}$ for $n$-neurons in a layer, whilst the various scaled regions grow as $1 - 2^{-n}$. This is a consequence of Leaky-ReLU's $S_n$ form, whilst not exhibiting a sign-flip symmetry consistent with $B_n$. Similar argumentation applies to any $S_n$-only function. For the Leaky-ReLU case, with large widths, almost the entire domain is scaled, resulting in an exponentially vanishing volume fraction of the domain where the identity map acts. This differs from the univariate perspective, where the two domains occupy half the space.

Hence, in a multivariate view, an untrained network may be expected to have an exponentially vanishing number of representations which are not scaled. This may mean that the activation function approaches a state where every representation has been scaled in some combination of components by $\alpha$. This appears to be undesirable to the network; consequently, training diagonalises it over the only orthant containing an identity map, such that activations cross this boundary and get scaled significantly differently.

This introduces an explainable algebraic mode through which symmetries can affect representations, since the orthant-wise behaviour is characteristic of permutation symmetries. However, crucially, this is a specific form of orthant effect that is only applicable to Leaky-ReLU and fails to provide a mode through which one can explain the structure in Tanh-like networks, though it may be generalised. The generalised orthant explanation continues to hold. Hence, the overall maximal symmetry remains the summary and most effective primary causal mechanism, which may act through various modes dependent upon the function[11].

Structure is generally more complicated in later latent layers for Leaky-ReLU-like tests, as demonstrated in *Fig.* 20 and 21 — which are representative examples of the broader array of the results. Several results, particularly the deepest layers, also showed an absence of representations or a very dense single direction. The former is likely also indicative of a narrow, strongly discrete-like anti-aligned representation, which falls outside the $\epsilon = 0.75$ thresholding.

Therefore, it appears that many findings identified in Tanh-like networks similarly occur in Leaky-ReLU-like networks, with only slight differences appearing to correspond to the slightly different maximal permutation symmetries to which the functions abide. This significantly bolsters the certainty of conclusions, due to the maximal symmetry being the only comparable analytical quality between the Leaky-ReLU-like and Tanh-like networks. It also rules out trivial backwards-pass geometry as a cause. Additionally, a specific explanatory mode through which the symmetry group may affect representations is suggested for Leaky-ReLU. Overall, the comparisons between anisotropic and isotropic Leaky-ReLU results, in combination with the Tanh-like comparative results, establish symmetry as the primary cause of the observed representational structure. It

---

[11]An attempt was made to produce a Leaky-ReLU-like function which partitions the space into orthants with alternating scaling and identity maps. This was termed a global-parity Leaky-ReLU and obeyed a permutation with *even* sign flips (denoted $D_n$, not to be confused with dihedral groups). It was constructed by decomposing the vector along the standard basis, then taking the sign of the product of (the sign of) components to determine either a scaling or identity map. This equal partitioning was hoped to be beneficial; however, this function *failed to train in all cases* and produced no influence on the representations. Perhaps future work could improve this approach by creating a working activation function with such symmetry. One could then explore its effect on representations, and explore the resultant taxonomic chain: $S_n \subset D_n \subset B_n \subset \mathrm{O}\,(n)$.

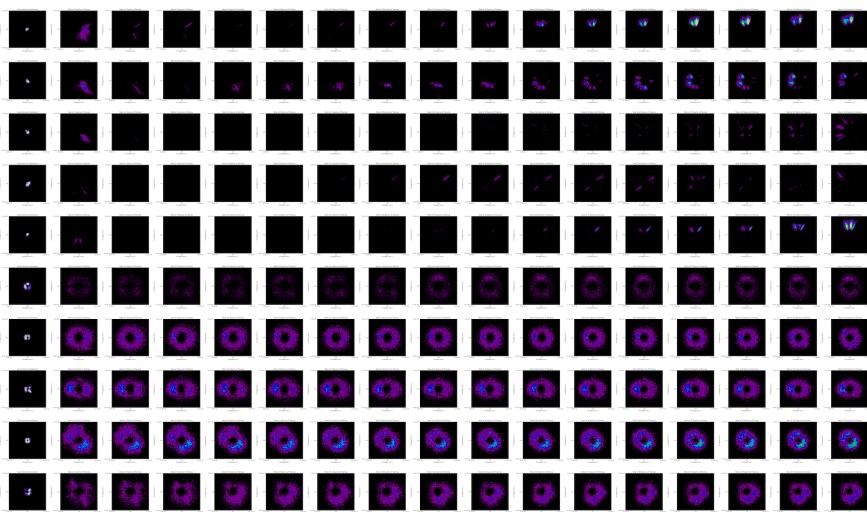

Figure 20: Shows the same networks and in the same order as *Fig.* 18, but results arise from the $1^{st}$ hidden layer. As is typical for isotropy, the bottom five rows demonstrate smooth, continuous-like representations filling the space. In anisotropic cases, the emergence of dense, ray-like clusters of representations can be observed again, although some are slightly more diffuse than those in the zeroth layer. Additionally, many of these occupy more general angles about the distinguished directions, possibly a further indication of discrete representational Superposition.

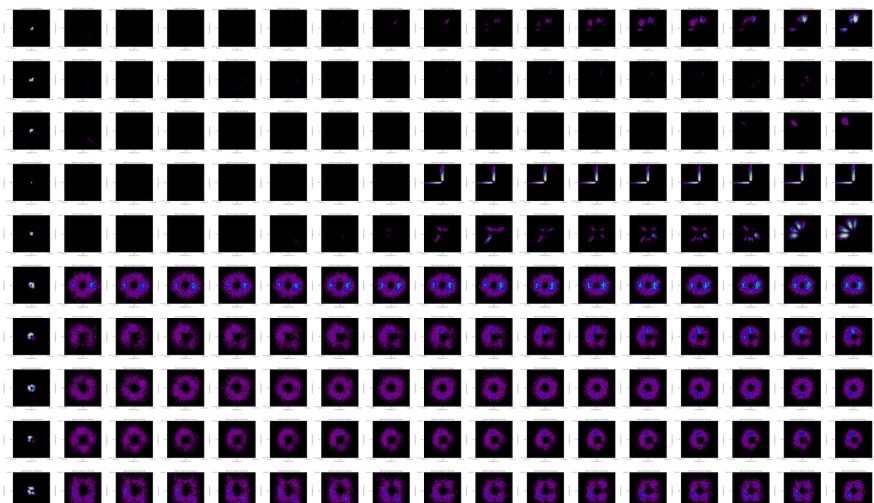

Figure 21: Shows the same networks and in the same order as *Fig.* 19, but results arise from the $1^{st}$ hidden layer. Again, as is typical for isotropy, the bottom five rows demonstrate smooth, continuous-like representations that fill the space. In comparison, anisotropic examples in the top five rows produce narrow, ray-like distributions. Notably, these are more axis-aligned than *Fig.* 21, but continue to frequently display discrete Superposition-like clusters, particularly in anisotropy's row 5. Unlike the prior plots, the anisotropic networks all align upwards and towards the left, potentially indicating a differing desirable orthant mapping in this case.

suggests that discrete groups tend to produce a quantising effect on representations, whilst more continuous groups remove such structure. Differences between Leaky-ReLU and Tanh experiments also indicate that nuances of the group structure may additionally affect the representations. This encourages a broader analysis

of the various symmetry groups through which primitives can be defined and their corresponding inductive biases on representations.

### B.4.1 Hyperoctahedral Leaky-ReLU

The prior results for Leaky-ReLU compared the contemporary standard permutation $S_n$ form of Leaky-ReLU against an isotropic form exhibiting $O(n)$ equivariance. These results are now extended by considering hyperoctahedral, $B_n$, forms of Leaky-ReLU. The hyperoctahedral group is situated between these groups: $S_n \subset B_n \subset O(n)$. Two approaches were considered to produce a $B_n$ form of Leaky ReLU. The first is an elementwise construction shown in *Eqn.* 8.

$$\mathbf{f}\left(\vec{x}; \{\hat{e}_i\}_{\forall \hat{e}_i}\right) = \sum_{i=1}^{n} f\left(\vec{x} \cdot \hat{e}_i\right) \hat{e}_i \tag{8}$$

The univariate map, $f$, can be given piecewise by *Eqn.* 9.

$$f(x) = \begin{cases} x - \epsilon(\alpha - 1) & : & x \leq -\epsilon \\ \alpha x & : & -\epsilon < x < \epsilon \\ x + \epsilon(\alpha - 1) & : & x \geq \epsilon \end{cases} \tag{9}$$

As an alternative, a max-norm formulation was constructed as shown in *Eqn.* 10.

$$\mathbf{f}\left(\vec{x}; \{\hat{e}_i\}_{\forall \hat{e}_i}\right) = \begin{cases} \alpha\vec{x} & : & \|\vec{x}\|_\infty < \epsilon \\ \vec{x} + \frac{\epsilon(\alpha-1)\vec{x}}{\|\vec{x}\|_\infty} & : & \|\vec{x}\|_\infty \geq \epsilon \end{cases} \tag{10}$$

Both formulations exhibit a hyperoctahedral symmetry about the standard basis. The results are shown below, including normalised and unnormalised cases, all performed on the CIFAR dataset.

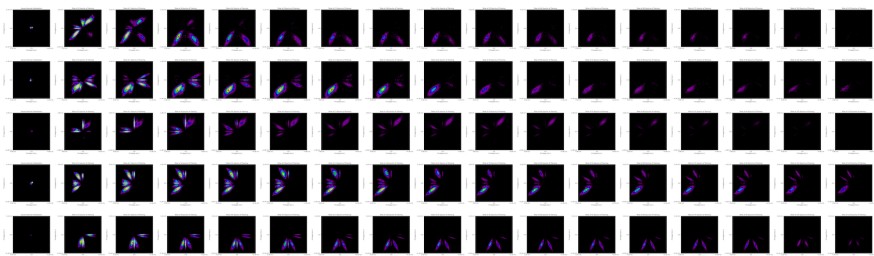

Figure 22: The results of the PPP method applied to the $0^{\text{th}}$ latent layer of an autoencoder with 1 hidden layer blocks and a width of 18 neurons per block, in this example, normalisation was not used. The five rows consist of the autoencoder using the Hyperoctahedral Leaky-ReLU from *Eqn.* 8. All five rows show dense beams of discretised-like clusters of representations. Many of these are situated along two orthogonal standard basis directions. Additionally, beams are observed at angles of approximately 25 degrees away from the standard basis directions in several of the plots. The beams also become fainter through training, as they are likely distributed outside of the PPP threshold.

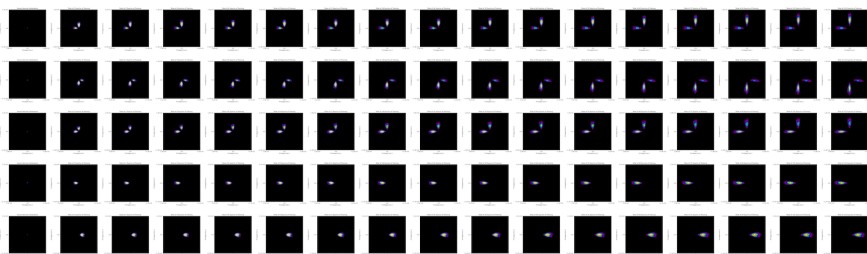

Figure 23: The results of the PPP method applied to the $0^{th}$ latent layer of an autoencoder with 1 hidden layer blocks and a width of 18 neurons per block, in this example, normalisation was not used. The five rows consist of the autoencoder using the Hyperoctahedral Leaky-ReLU from *Eqn.* 10. These are the same conditions as *Fig.* 22 but with the alternative implementation for hyperoctahedral Leaky-ReLU. In this plot, the top three rows show perpendicular beams aligning with the standard basis, whilst the bottom two rows show a single dense beam aligned with just one basis vector.

Both *Figs.* 22 and 23, demonstrate that the hyperoctahedral form of Leaky-ReLU also produces discrete-like distributions of activations. Additionally, these results are taken from the $0^{th}$ layer, indicating this is a non-trivial consequence. Normalisation was not utilised in these plots, and they are observationally similar to other permutation and hyperoctahedral plots without normalisation. This suggests that the algebraic symmetry of the functional form remains a useful predictor of biases that emerge in representations as a result of these functional form choices. The following plots of *Figs.* 24 and 25 demonstrate these hyperoctahedral variants for normalised cases.

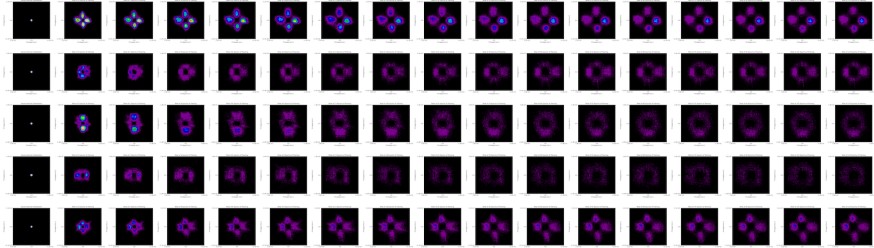

Figure 24: The results of the PPP method applied to the $0^{th}$ latent layer of an autoencoder with 2 hidden layer blocks and a width of 18 neurons per block, in this example, normalisation was used. The five rows consist of the autoencoder using the Hyperoctahedral Leaky-ReLU from *Eqn.* 8. Both rows 1 and 5 indicate a tendency for representation overdensities aligning with all standard basis vectors; in the remaining rows, a pattern is not so evident, but a faint alignment remains visible, particularly in rows 2 and 3.

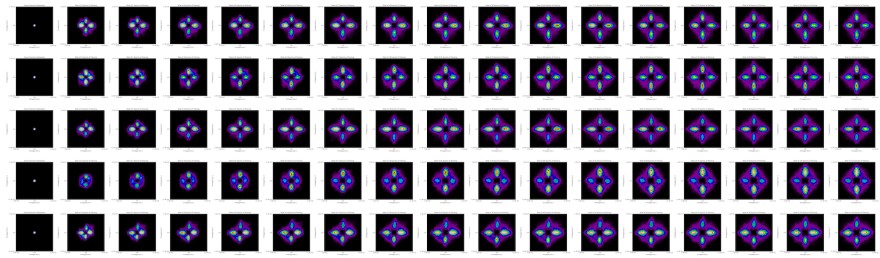

Figure 25: Demonstrates the same experiment as *Fig.* 24; however, with the hyperoctahedral formulation given in *Eqn.* 10. These results show a much stronger basis alignment in all cases, with a falloff around these more aligned beams. In all rows except row 1, there appear to be more dense representations in two antipodal directions, and then a slightly weaker density in the antipodal directions orthogonal to these.

Both *Figs.* 24 and 25 show that discretised representations continue to emerge in hyperoctahedral Leaky-ReLU networks with normalisation present too. Moreover, *Fig.* 25 shows qualitatively similar results to many hyperoctahedral Tanh networks of *Figs.* 9 and 10 as opposed to permutation Leaky-ReLU in *Fig.* 19. The latter is the same experiment as *Fig.* 25 but using permutation Leaky-ReLU. Overall, this strongly reinforces that maximal algebraic symmetries are a useful predictor, since the induced representational structure of *Fig.* 25, more closely aligns with other hyperoctahedral experiments as opposed to other Leaky-ReLU experiments, despite being a hyperoctahedral Leaky-ReLU analogue. This arrangement was observed in results for both *Eqns.* 8 and 10, but more often the latter.

A selection of further particularly indicative results is shown below, drawn from a variety of experiments.

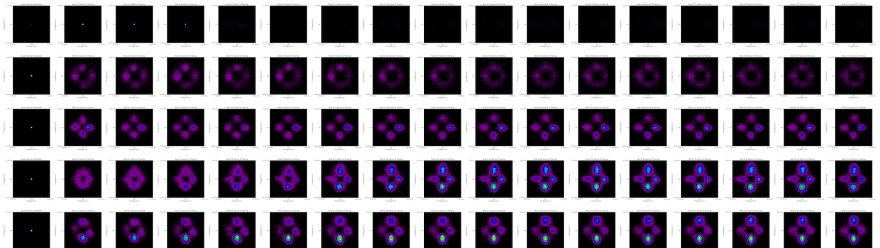

Figure 26: The results of the PPP method applied to the $3^{\text{rd}}$ latent layer of an autoencoder with 3 hidden layer blocks and a width of 18 neurons per block, in this example, normalisation was used. The five rows consist of the autoencoder using the Hyperoctahedral Leaky-ReLU from *Eqn.* 8. These results continue to show the pattern of representation overdensities situated over the standard basis directions.

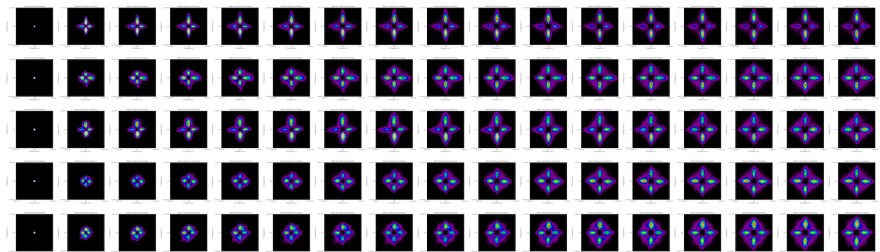

Figure 27: The results of the PPP method applied to the $0^{\text{th}}$ latent layer of an autoencoder with 3 hidden layer blocks and a width of 18 neurons per block, in this example, normalisation was used. The five rows consist of the autoencoder using the Hyperoctahedral Leaky-ReLU from *Eqn.* 10. These results appear similar to those of *Fig.* 25, with dense beams of the standard basis vectors with some falloff.

Overall, these results continue to confirm that activation functions can induce unintended structure into distributions. They additionally show that the maximal algebraic symmetries continue to be a good predictor of the presence of this structure, with all results showing a clear trend. Moreover, the hyperoctahedral Leaky-ReLU ($B_n$-LeakyReLU) results are largely different from those of the standard permutation Leaky-ReLU ($S_n$-LeakyReLU) over the standard basis; yet, some similarities emerge in comparison to the hyperoctahedral Tanh. This indicates that these maximal symmetries may be a helpful way to categorise biases emerging from these primitives. Together, these are considered to strengthen the paper's conclusions.

### B.5 MNIST Results

The MNIST results remain consistent with the overall conclusion and show little difference overall. Several representative examples are displayed in this section, which illustrate this, and continue to use a normalisation produced across all MNIST training samples and an/isotropic-tanh.

An initial example is displayed in *Fig.* 28 and 29 — the latter showing the same plot, with colour levels adjusted to make faint structure more observable for analysis. Both are included for comparability and transparency.

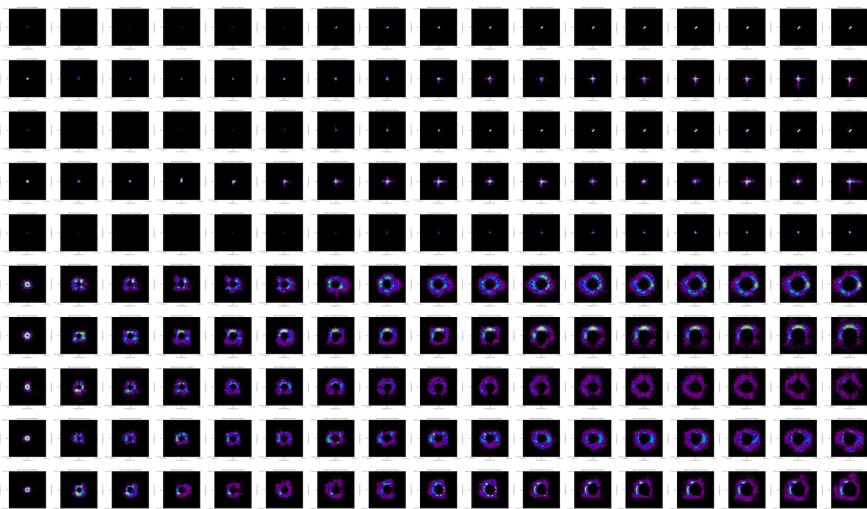

Figure 28: Displays the PPP method applied to the $0^{th}$ latent layer of an autoencoder with 3-hidden-layer blocks consisting of 18 neurons per block. This architecture is comparable to those in *App.* E in every way except for the dimensions of CIFAR, which is exchanged for MNIST shape on input-output layers. Overall, one can observe that isotropic networks in the bottom five rows continue to produce smooth distributions that fill the representation space. In comparison, anisotropic networks on the top five rows tend to have many representations centred around the origin, and some produce axis-aligned features.

Several of the $0^{th}$ layer plots show a closer grouping of representations about the origin. This is believed to be due to many of the MNIST examples' components reaching the extreme values for the space, 0 and 1, with fewer intermediate values. Often, this occurs repeatedly in specific patches due to the overlapping morphologies of MNIST numbers, which may produce clustered representations inherent to the dataset. This makes the MNIST dataset more anisotropic, featuring already dense, discrete clusters about the corners of an embedded hyper-cube — significantly different from the more intermediate RGB values arising from CIFAR. Slight band-groupings of representations are also sometimes observable, likely due to the same representation being projected in multiple planes, as well as the precision of the dataset, which produces slight clusterings. However, MNIST's greater inherent anisotropy appears not to significantly alter the results from CIFAR, likely due to the layer bottleneck embedding these higher-dimensional representations more evenly in the latent layer during initialisation.

Similar results continue to occur in later latent layers, where axis-aligned clustering becomes progressively more pronounced in deeper layers for anisotropy, whilst typically more diffuse for isotropy. This is demonstrated in *Fig.* 30.

Overall, this section corroborates that such a function-imposed structure generalises across other datasets and emphasises the task-agnostic production of such a structure through the inductive biases associated with functional forms.

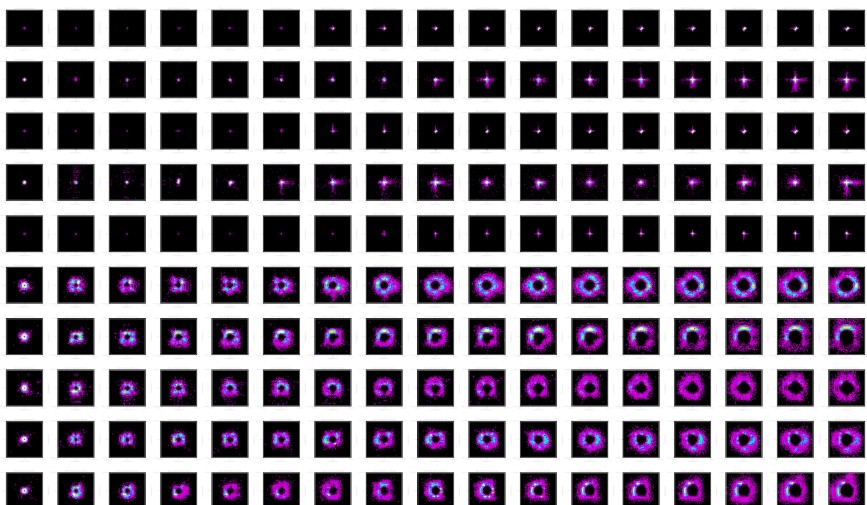

Figure 29: Displays the exact same plot as *Fig.* 28, with the colour map levels adjusted in post-processing to make the faint representations more observable. In this plot, the axis-aligned representations of anisotropic networks are more observable. This demonstrates how anisotropic tanh produces task-agnostic structure in MNIST reconstruction autoencoders as well.

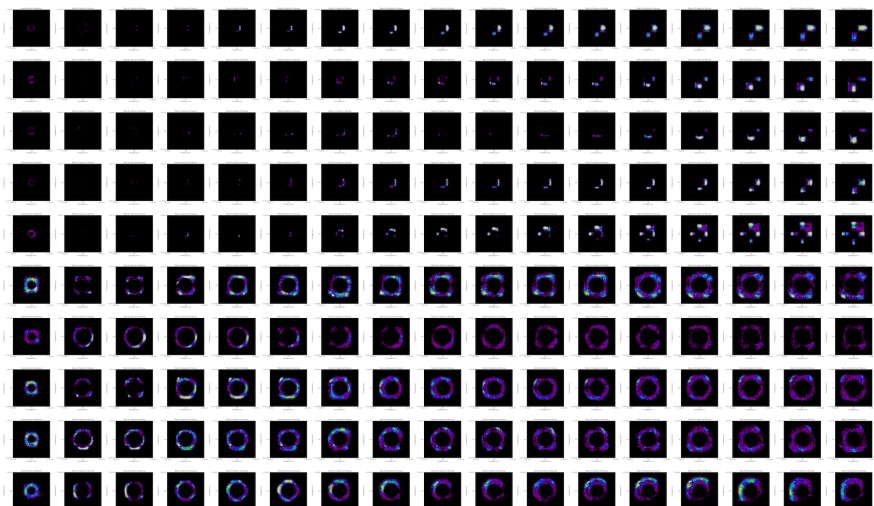

Figure 30: Displays the PPP method applied to the 1st latent layer of an autoencoder with 3-hidden-layer blocks consisting of 18 neurons per block. Row-wise, these are the exact same networks as displayed in *Fig.* 28 and 29, but for the next latent layer. One can observe an axis-aligned structure in the top five rows, corresponding to anisotropic standard-tanh, and this is consistent with all previous experiments. In contrast, a smoother, approximately continuous distribution is observed for the bottom five rows, which are isotropic, and is consistent with this trend.

## C    Continued Theoretical Development

This section extends the previous theoretical development to other architectures, low level description of gradient flow, alongside a discussion of the speculated pathological consequences of these function-driven biases.

The gradient flow work also highlights an "affine divergence", a structural mismatch inherent in affine maps, irrespective of anisotropic or isotropic activation functions. It shows that the parameter updates are the true

steepest descent, but induce a representational update which is *not* the steepest descent step in a sample-dependent manner. This forgrounds a discussion comparing theoretical-ideal and correction-propagated effective updates, which may also relate to the differences between isotropic and anisotropic performances.

### C.1 General Picture of Symmetry in General Networks

In this section, network symmetry breaking is briefly discussed, along with speculation on how general networks may respond to the foundational biases identified in this work. This clarifies how the consideration of such function-driven symmetry biases can act on networks with symmetry breaking, but may not contribute in specialist scenarios. This discussion is at a heuristic level.

For most general networks, their construction is not motivated or designed to vary predictably in response to group actions on representations. Therefore, if a network doesn't commute with actions or remain invariant to them, then a shift in representations corresponding to some group action will then impact and interact non-trivially with the rest of the network. This is because those representational shifts in the direction of group actions result in non-trivial changes to the resultant loss. Hence, they couple to optimisation through the production of a gradient. These non-zero gradients encourage the representations to take steps along paths between points related by the discrete group actions, thereby inducing exploration.

This is argued to produce a knock-on chain of interactions within the network. At a heuristic level, it is this influence which is expected to result in unintended biases. It also highlights why groups/symmetry, rather than just a general action, may contribute an important characterising property, since many analogous actions within a group may each contribute a similar effect in the sub/space they transform and the resultant effects. Similarly, these biases are expected to be broadly consistent across sets of primitives that share the same maximal symmetry, due to the actions from the group-defined forms relating differing computational maps. This is a more internalist model of how symmetries relate to deep learning: the preexisting symmetries of functional forms may already predispose the network to influenced dynamics, before considering data-specifics.

This is expected to differ from symmetries in end-to-end networks (Bronstein et al., 2021), where their representations are designed to transform with the symmetry, resulting in the entire model often being either equivariant or invariant to its effect. Hence, those representative degrees of freedom are connected through group actions and become semantically constrained by the group structure. As a result, they semantically represent the pose information derived from the group. This form of network-symmetry consideration relates to an externalist picture of symmetry, in which symmetry is designed into networks to mimic those expected in the data.

This is not the case in general networks, where a symmetry action does not pass cleanly through or identically cancel with layers — a general network is capable of both explicit and spontaneous symmetry breaking. Therefore, these group-connected degrees of freedom may represent more general semantics rather than solely pose.

More informally, one can imagine that such symmetry actions can result in interactions within the network due to its structure and consequently alter dynamics, rather than passing straight through unimpeded by design. In this informal heuristic, a layer may 'absorb' a symmetry action in a non-invariant/equivariant manner, or have an interaction with its differing symmetry. This may mean the symmetry does, in principle, passively 'act' on the network and hence produce a realisable and tangible effect. This suggests that maximal symmetry may be an important consideration in general networks, leading to similar function-driven biases. These are felt to be underappreciated in significance.

This may pertain to a range of architectural models: Convolutional Layers (LeCun et al., 2002), ResNets (He et al., 2016) and Transformers (Vaswani et al., 2017). Substantial theoretical development may be needed to produce a Popperian-style (Popper, 1935) predictive phenomenon to verify for each of these particular architectures; although the following speculations may be pertinent.

Convolutions exhibit two notable symmetries of somewhat different quality: local translational (ignoring edge effects) and kernel-permutational symmetry, which are innately built into their weight-sharing architecture. These contributions may couple non-trivially with the symmetries of activation functions, normalisers, initialisers, and more, producing complex, potentially hierarchical biases that influence trajectories. If all

parameters are adjusted accordingly, then the nature of the kernel-permutation symmetry means that channels can be exchanged without functional consequences — and such a permutation is induced by the activation function.

This is remarkably similar to the permutation-orthogonal promotion of the activation function in this study, which can be explored when applying over channels. This may mean that channels can continuously rotate into one another for orthogonal activation functions applied channel-wise. This may produce more vector-like semantics across channels, rather than the axis-aligned features currently observed (Bau et al., 2017). However, the implications of multiple convolutions may be challenging to predict, due to the repeated patchwise operations that introduce discreteness to incoming features. This may be a significant confounding factor.

If the Residual layer is of form $\mathbf{f}(\vec{x}) = (\mathrm{I})\vec{x} + \mathbf{g}(\vec{x})$, then since identity I commutes with all actions, then $\mathbf{f}$ will acquire the symmetry of $\mathbf{g}$. How such an identity contribution affects representation trajectories will require analysis similar to *App.* C.2. However, the preceding layer will provide a much more direct correction to the representations of the following layer due to the skip. This requires the first-order, single-layer analysis of *App.* C.2 to be further generalised. The result may be a more nuanced set of attractors for representations, those which preexisted due to $\mathbf{g}$ interacting with contributions by the prior layer. This could be an advantageous direction to explore further, and the predictive phenomenon may be more tractable to derive than convolutional or transformer-based models. It may itself indicate how (weighted) residual connections allow for a more complicated control over primitive-derived biases for networks, and greater control of how semantics organise. It could also be a preliminary avenue for exploring the decomposition of functions into symmetry-defined functional forms, as suggested previously.

Transformers are likely the least trivial to derive theoretical, compositional-bias predictions for. The architecture is composed of an array of distinct computational blocks with their own symmetries, many differing operations, and significant overall depth. This may require a thorough understanding of hierarchical-taxonomic biases produced by these differing individual functions before generalising them to transformer predictions. A stepwise approach of combining and studying each individual operation until reaching transformers may be a practical direction for understanding the foundational biases proposed. This may be a longer-term aspiration of the theory.

Finally, other classification codings may be explored within this hypothesis of function-induced representational biases. One-hot encoding is identical to a local coding scheme and is permutation-invariant as long as labels are exchanged, whereas binary representations correspond to dense coding. These may contribute significant unintended representational biases, which may explain phenomena such as Neural Collapse (Papyan et al., 2020). Such confounding influences were the direct reason for studying autoencoder reconstruction in these ablation trials. Determining the interaction between these *choices* of classification outputs and functional-form biases may be a promising direction for further exploration; finding well-matching functional-forms and classification outputs may be highly advantageous for model performance.

Overall, these considerations may be leveraged to gain a greater understanding and therefore control over the structure of representations induced by functional forms. Extending theoretical predictions to a broader array of architectures may enable improved representation organisation and yield better performance. In general, generative architectures may benefit more from smoother, non-quantised representations, which preserve more semantic nuance between representations, whilst classification networks, with their reductionist categories, may be less influenced by functional-form-induced quantisation, since it can be effective at grouping representations — although the risk of premature grouping may reduce semantic separability.

### C.2 Gradient Flow Mechanisms for Symmetry-Led Quantisation

At a macroscale, symmetry-led arguments provide a general route for how various symmetry-defined functional forms may yield structure in representations; yet this is a high-level argument. At a microscale, and for each instance of such functional forms, the influence must be explained by backpropagating gradients correcting parameters, which in turn influence representations. Hence, analysing gradient flow provides a low-level mechanistic understanding of the phenomena specific to each realised function of the form.

The connection between symmetry-defined functional forms and gradients is as follows: those defined functional forms result in particular contributions to their Jacobian. Hence, the function's Jacobian is directly determined by the symmetry, too. This Jacobian then modulates and deflects the gradient update direction, which may integrate to form stable attractors or continua relevant for the quantisation phenomenon.

Instead of studying gradient flow directly, this analysis considers a first-order correction to the representations of a single fully connected layer. This is because the phenomena observed in this paper occur in representations and emerge during optimisation, so such corrections are directly relevant and must be causal. All the respective gradients will be computed, and their implications on representations will be studied. The function is the map $\mathbf{f} : \mathbb{R}^n \to \mathbb{R}^n$. A single fully-connected layer is displayed in *Eqn: 11*.

$$
\begin{aligned}
\vec{a} &= \mathbf{f}\left(\vec{z}\right) \\
\vec{z} &= \mathbf{W}\vec{x} + \vec{b}
\end{aligned}
\tag{11}
$$

This single layer will also be expected to be part of a larger model, such that an eventual loss, $\mathcal{L}$, is determined and its gradient with respect to $\vec{a}$ similarly — notated as $\vec{g} \equiv \frac{\partial \mathcal{L}}{\partial \vec{a}}$.

Moving forward, these will be expressed in the Einstein Summation Convention to make the calculus more straightforward. The gradient terms are similarly evaluated at a particular $\vec{x}$, which is generally suppressed for notational simplicity.

Representations are the direct quantity of study, and, without regularisation, are typically the quantity that *more* directly contributes to the loss value than parameters. The gradient of loss with respect to $\vec{a}$ and $\vec{z}$ is given in *Eqns.* 12 and 13, respectivly. The term $\mathcal{J}_{pn}^{(\mathbf{f})}$ is the jacobian of the activation function, $\mathbf{f}$.

$$
\left.\frac{\partial \mathcal{L}}{\partial a_n}\right|_{\vec{x}} = g_n
\tag{12}
$$

$$
\left.\frac{\partial \mathcal{L}}{\partial z_n}\right|_{\vec{x}} = \frac{\partial \mathcal{L}}{\partial a_p}\frac{\partial a_p}{\partial z_n} = g_p \mathcal{J}_{pn}^{(\mathbf{f})}
\tag{13}
$$

However, these quantities cannot be modified directly because they are activations and dependent on the input (and parameters). Parameters do not depend on the variable input, so are amenable to update. Parameter updates are undertaken, which in turn modify these representations by proxy. Computing the gradients with respect to $\mathbf{W}$ and $\vec{b}$, given in *Eqns.* 14 and 15, respectivly.

$$
\left.\frac{\partial \mathcal{L}}{\partial W_{nm}}\right|_{\vec{x}} = \frac{\partial \mathcal{L}}{\partial a_p}\frac{\partial a_p}{\partial z_q}\frac{\partial z_q}{\partial W_{nm}} = g_p \mathcal{J}_{pn}^{(\mathbf{f})}\left(\vec{x}\right) x_m
\tag{14}
$$

$$
\left.\frac{\partial \mathcal{L}}{\partial b_n}\right|_{\vec{x}} = \frac{\partial \mathcal{L}}{\partial a_p}\frac{\partial a_p}{\partial z_q}\frac{\partial z_q}{\partial b_n} = g_p \mathcal{J}_{pn}^{(\mathbf{f})}
\tag{15}
$$

Using these updates, we can then determine the induced correction on representations. This 'update' is a first-order, single-layer approximation: it involves only one update and considers how this layer's parameter updates affect the representations, while assuming the preceding layer's parameters are held constant to avoid non-factorisable non-linear terms. A learning rate is notated as $\eta$. A 'propagation of corrections' is shown from parameters to representations in *Eqns.* 16 and 17.

$$
\begin{aligned}
z'_i &= W'_{ij} x_j + b'_i \\
&= \left( W_{ij} - \eta \frac{\partial \mathcal{L}}{\partial W_{ij}} \right) x_j + \left( b_i - \eta \frac{\partial \mathcal{L}}{\partial b_i} \right) \\
&= \left( W_{ij} - \eta g_p \mathcal{J}^{(\mathbf{f})}_{pi} (\vec{x}) x_j \right) x_j + \left( b_i - \eta g_p \mathcal{J}^{(\mathbf{f})}_{pi} \right) \\
&= \underbrace{W_{ij} x_j + b_i}_{z_i} - \eta \underbrace{\left( g_p \mathcal{J}^{(\mathbf{f})}_{pi} (\vec{x}) x_j x_j + g_p \mathcal{J}^{(\mathbf{f})}_{pi} \right)}_{\text{Effective } \frac{\partial \mathcal{L}}{\partial z_i}}
\end{aligned}
\tag{16}
$$

$$
z'_i = z_i - \eta g_p \mathcal{J}^{(\mathbf{f})}_{pi} \left( \|\vec{x}\|^2 + 1 \right)
\tag{17}
$$

The expression of *Eqn.* 17, must contain the low-level mechanistic mode through which representations are affected by the symmetry-defined functional forms. The only difference in the update arises from the $\mathcal{J}^{(\mathbf{f})}_{pi}$ term, depending on which functional form is implemented.

Similarly notable is that there is a distinct difference between the mathematical/theoretical ideal update for these affine layers, shown in *Eqn.* 13 and 18, and the effective/realised update to the representations, shown in *Eqn.* 17. These *do not coincide*, and the sample-wise scaling term emerges ($\|\vec{x}\|^2 + 1$) (upto single-layer order). In effect, although parameters take the optimum step of steepest descent, *representations do not move in the true steepest descent direction* across a batch, and are over-amplified for single samples, creating inconsistency between update-sizes per step. Overall, this gradient vector is deflected due to improper weighting of the contributions. This is further detailed in the dedicated *App.* C.2.1, which speculates that this is a fundamental phenomenon and hypothesises it is associated with the success of normalisation.

$$
z'_i = z_i - \eta \frac{\partial \mathcal{L}}{\partial z_i} = z_i - \eta g_p \mathcal{J}^{(\mathbf{f})}_{pi}
\tag{18}
$$

Proceeding with the anisotropic (*Eqn.* 19) versus isotropic (*Eqn.* 20) implications on gradients, the following equations must be considered for their distinct Jacobian contributions.

$$
\mathbf{f}(\vec{x}) = \sum_i \psi(\vec{x} \cdot \hat{e}_i) \hat{e}_i \Rightarrow \mathbf{f}(\vec{x})_k = \psi(x_j e_{ji}) e_{ki}
$$
$$
\Rightarrow \mathcal{J}^{(\mathbf{f})}_{nm} = \psi'(x_j e_{ji}) e_{ni} e_{mi}
\tag{19}
$$

$$
\mathbf{f}(\vec{x}) = \phi(\|\vec{x}\|) \hat{x} \Rightarrow \mathbf{f}(\vec{x})_k = \psi\left(\sqrt{x_j x_j}\right) \frac{x_k}{\sqrt{x_j x_j}}
$$
$$
\Rightarrow \mathcal{J}^{(\mathbf{f})}_{nm} = \left( \frac{\phi'(\|\vec{x}\|)}{\|\vec{x}\|^2} - \frac{\phi(\|\vec{x}\|)}{\|\vec{x}\|^3} \right) x_n x_m + \frac{\phi(\|\vec{x}\|)}{\|\vec{x}\|} \delta_{nm}
\tag{20}
$$

Using $e_{ij} = \mathrm{I}_{ij}$ as the standard basis and $r = \|\vec{x}\|$ for compactness. For familiarity when interpreting these Jacobians, conventional matrix notation will be used. The symbol $\odot$ denotes a Hadamard product.

$$
\begin{aligned}
\text{Anisotropic}: \quad & \mathcal{J}^{(\mathbf{f})}_{nm} = \mathrm{diag}\left(\psi'(\vec{x})\right) = \left(\psi'(\vec{x}) \vec{\mathrm{I}}^T\right) \odot \mathrm{I} \\
\text{Isotropic}: \quad & \mathcal{J}^{(\mathbf{f})}_{nm} = \left( \phi'(r) - \frac{\phi(r)}{r} \right) \hat{x}\hat{x}^T + \frac{\phi(r)}{r} \mathrm{I}
\end{aligned}
\tag{21}
$$

The difference between these equations determines how the representations are shaped mechanistically due to the symmetry-defined functional forms. Substituting the two Jacobians of *Eqn.* 21 into *Eqn.* 17 yields the two equations of *Eqn.* 22 respectivly. Where the top row is the anisotropic correction to representations and the bottom is the isotropic correction to representations. Notationally using $\eta_{\text{eff}} = \eta \left( \|\vec{x}\|^2 + 1 \right)$ to simplify interpretations. Similarly representing $\alpha(r) \equiv \frac{\phi(r)}{r}$ and $\beta(r) \equiv \left( \phi'(r) - \frac{\phi(r)}{r} \right)$ simplifies the contributions.

$$\text{Anisotropic}: \quad \vec{z}' = \vec{z} - \eta \left( \psi'\left(\vec{x}\right) \odot \vec{g} \right)$$
$$\text{Isotropic}: \quad \vec{z}' = \vec{z} - \eta \alpha\left(r\right) \vec{g} - \eta \beta\left(r\right) \left(\hat{x} \cdot \vec{g}\right) \hat{x} \tag{22}$$

The difference between these equations, to first-order, induces the effects observed in this paper — although higher-order effects may not be negligible either[12]. One may note that the expressions of *Eqn.* 22 are derived from the symmetry-defined *functional forms* not specific function instances, suggestive of them generalising over these taxonomies as observed. Moreover, it may support the decomposition principle, in which a function is represented as decomposed components of various symmetry-defined functional forms. Each component would then contribute foundational bias effects on representations and the broader network through their differing Jacobians.

One can see that the anisotropic form scales the decomposed components of the gradient by the diagonal Jacobian of the elementwise function. This results in an uneven, componentwise scaling and a systematic deflection of the gradient away from near-ideal $\vec{g}$ for the $\vec{a}$ correction (especially when propagating through the nonlinearity). Since this deflection acts componentwise, each component of the gradient is scaled, privileging or suppressing respective parts and bending the trajectory due to this symmetry-defined functional form. This may create degenerate attractors in each orthant, resulting in representations becoming denser over each attractor, potentially yielding quantisation in the extreme. One could also argue that when performing a first-order propagated correction to $\vec{a} = \mathbf{f}\left(\vec{z} + \vec{\delta z}\right) \approx \vec{a} + \vec{\delta a}$, this diverges substantially from the ideal correction on $\vec{a}$ which is in direction $\vec{g}$ as seen in *Eqn.* 12.

This differs considerably from the update to the isotropic representations. The adjustment to $\vec{z}$ has a correction directly in the direction of $\vec{g}$, hence a term colinear to $\vec{g}$ (close to ideal descent when propagated to $\vec{a}$). It has a second contribution which scales with the gradient $\vec{g}$ projection onto the $\hat{x}$: $\left(\vec{g} \cdot \hat{x}\right) \hat{x}$, a colinear term to $\vec{x}$ acting as a radial adjustment. Thus, representations move coherently as whole vector terms rather than independent decomposed elements.

Hence, the anisotropic gradient vector update is *deflected* from $\vec{g}$ predictably by the basis and the particular orthant it is in. These deflections will be similarly organised by symmetry, producing degenerate sets of deflections per orthant. Hence, the update is similarly influenced by the symmetry. Whilst the isotropic update is not dependent on the basis, nor the particular orthant, and has a component proportional to the desired $\vec{g}$[13]. This describes mechanistically the emergence of the representation organisation due to symmetry, with particular $\alpha, \beta, \psi'$ depending on the function, which can be further analysed as a future research direction.

Overall, the symmetry-led functional forms, at a high level, were predicted and shown to yield denser representations organised by symmetry, whilst, at a low level, the gradient and, particularly, propagated representation corrections describe how this phenomenon takes shape through a low-level mode. Further exploration of these functions, and others, and their consequences for update trajectories may be a worthwhile direction to probe for a better understanding of the ideal representations for deep learning. Particularly those which are less constrained by artificial structure injected by our functional form choices, or at least constrained desirably.

Additionally, this analysis shows that an ideal update to affine parameters induces discrepancies between idealised and effective updates when propagated through representations. This distinction between ideal and effective updates appears notably significant. This "affine divergence" is discussed to first-order, single-layer, but may be generalised further and generalises to other idealised-effective update discrepancies. For the

---

[12]These divergences could be pursued relentlessly. This would eventually become computationally intractable, requiring cycles of forward and backwards propagation of gradients until converging to the exact gradient of the loss with respect to outputs — perhaps producing an explicit/implicit-scheme-like split in the approach to optimisation. However, to first order, this remains tractable and arguably desirable, as empirical studies may validate. This could be generalised to ensure, layer-wise, that all representational corrections are ideal, and to accept some approximation by not pursuing corrections beyond non-linearities. Furthermore, if the proposed affine correction is undertaken, each backpropagated gradient will pick up an additional $n = (1 + \|\vec{x}\|^2)^{-0.5}$ per layer, which may allow a perturbative approach in $n$ to such representational corrections.

[13]Enforcing perfect alignment with the idealised $\vec{a}$ update would induce an identity-scaled Jacobian which integrates to a linear activation function and hence a linear system. So some discrepancy is necessitated to yield a non-linear network.

affine divergence, this may be a fundamental explanatory mode for the success of normalisation and requires substantive further consideration of which terms require ideal-effective correction alignment and which may be optimal to adapt/diverge to enable this. It is proposed that aligning representations may be preferable, as this is the primary quantity of interest, and this would be significantly different from current standard practice. Empirical testing will be needed to validate such a prediction, particularly whether it accounts for the success of normalisation by exploring if a perfect correction improves performance beyond current normalisers.

### C.2.1 An Affine Divergence in Updates

Continuing from the previous discussion regarding the difference between the ideal and effective representational corrections. One can write this effective update in compact notation provided in *Eqn.* 23, whilst this "affine divergence" is summarised in *Eqn.* 24. This equation shows that the effective and idealised updates *do not* coincide. The equality is shown in *Eqn.* 25 for completeness.

$$\frac{\Delta \mathcal{L}}{\Delta z_i} = -\frac{z_i' - z_i}{\eta} = g_p \mathcal{J}_{pi}^{(\mathbf{f})} \left( \|\vec{x}\|^2 + 1 \right) \tag{23}$$

$$\frac{\Delta \mathcal{L}}{\Delta z_i} \neq \frac{\partial \mathcal{L}}{\partial z_i} \tag{24}$$

$$\frac{\Delta \mathcal{L}}{\Delta z_i} = \frac{\partial \mathcal{L}}{\partial z_i} \left( 1 + \|\vec{x}\|^2 \right) \tag{25}$$

Since representations are the direct quantity of computational interest produced by the model and consequential to the loss, whilst parameters are only indirect, this suggests we must consider which is more important to align in terms of the theoretical and effective updates — it appears neither can be satisfied concurrently.

This shows a fundamental, unavoidable structural misalignment in the update step of affine maps, which can be generalised to other maps. This requires careful future consideration and treatment, determining which terms should be aligned and which should not.

The presence of such a sample-wise scaling, *strongly* weights updates by the magnitude of the incoming sample and may be undesirable compared to the desired representational correction, where this factor is absent.

A future direction is to thoroughly explore this. Several corrections may be empirically explored, such as through a learning-rate correction $\eta' = \frac{\eta}{\|\vec{x}\|^2 + 1}$, or through modifying the affine transform step to introduce such a scaling factor: $\vec{z} = s^{-1}(\mathbf{W}\vec{x} + \vec{b})$ with $s = \sqrt{\|\vec{x}\|^2 + 1}$. Moreover, this naturally induces/derives an *isotropic* nonlinearity into the system in the latter approach, which is somewhat analytically similar to the proposed isotropic-tanh.

However, this latter approach is not a classical normaliser or activation function; rather, it is more faithfully a first-order gradient-based correction to the affine map — although it may implicitly normalise the magnitudes of the output vector. It does not produce classical scale-invariance of normalisers, or project out representational degrees of freedom such as magnitude. Nor does it match the classical definition of an activation function as a nonlinearity precomposed with the affine map. Instead, it is best described as a new form of map, an alternative to the affine map.

Both approaches have consequences for gradient flow and implications for updates; e.g., by deweighting the parameter correction by its magnitude, the parameters are inverse-weighted by these terms. Different balances between these scalings should be empirically explored. Similarly, the latter approach propagates notably different gradients and representations to the preceding and subsequent layers, respectively; the consequences may be non-trivial.

The overall "affine divergence" term may also contribute to a fundamental mode in the success of normalisation. Normalisation may partially mitigate extreme over-weightings by ensuring that all samples' squared-term

factors approach a low-variance, approximately constant value. Thereby bringing all weightings into a more comparable range. This could be considered across the classical normalisations, determining which reduce the $\mathrm{Var}\left(\|\vec{x}\|^2 + 1\right)$ best and whether this correlates to performance. However, if this does contribute to the success of normalisation, the mitigation is only partial and can be fully corrected by the preceding suggestions for adapting the learning rate or the affine map. This is important because the representation correction is the actual direct adjustment desired, which is only indirectly materialised through the parameter update.

One can compare the two forward-pass cases, which resolve this discrepancy, a more classic normaliser on the input $\vec{x}$, $\vec{z} = \mathbf{W}\frac{\vec{x}}{s} + \vec{b}$ and this affine correction $\vec{z} = \frac{\mathbf{W}\vec{x} + \vec{b}}{s}$. Both approaches eliminate the sample-dependency in $\left(\|\vec{x}\|^2 + 1\right)$ through the first-order analysis. This is shown in Einstein notation in *Eqns.* 26 through 31 and *Eqns.* 32 through 37, respectively.

$$z_i = W_{ij}\frac{x_j}{s} + b_i \tag{26}$$

$$\frac{\partial \mathcal{L}}{\partial W_{nm}} = g_i \frac{\partial z_i}{\partial W_{nm}} = \frac{g_n x_m}{s} \tag{27}$$

$$\frac{\partial \mathcal{L}}{\partial b_n} = g_i \frac{\partial z_i}{\partial b_n} = g_n \tag{28}$$

$$\Rightarrow \quad z_i' = \left(W_{ij} - \eta \frac{g_i x_j}{s}\right)\frac{x_j}{s} + (b_i - \eta g_i) \tag{29}$$

$$z_i' = z_i - \eta g_i \left(\frac{\|\vec{x}\|^2}{s^2} + 1\right) \quad \Rightarrow \quad s = \|\vec{x}\| \tag{30}$$

$$z_i' = z_i - \eta' g_i \tag{31}$$

This correction may require a half-scaling of the learning rate, $\eta' = 2\eta$, to remain comparable. The other approach is the proposed affine correction term.

$$z_i = \frac{W_{ij}x_j + b_i}{s} \tag{32}$$

$$\frac{\partial \mathcal{L}}{\partial W_{nm}} = g_i \frac{\partial z_i}{\partial W_{nm}} = \frac{g_n x_m}{s} \tag{33}$$

$$\frac{\partial \mathcal{L}}{\partial b_n} = g_i \frac{\partial z_i}{\partial b_n} = \frac{g_n}{s} \tag{34}$$

$$\Rightarrow \quad z_i' = \left(W_{ij} - \eta \frac{g_i x_j}{s}\right)\frac{x_j}{s} + \left(b_i - \eta \frac{g_n}{s}\right) \tag{35}$$

$$z_i' = z_i - \eta g_i \left(\frac{\|\vec{x}\|^2 + 1}{s^2}\right) \quad \Rightarrow \quad s = \sqrt{1 + \|\vec{x}\|^2} \tag{36}$$

$$z_i' = z_i - \eta g_i \tag{37}$$

One can see that, up to a learning rate constant, these make the desired representational corrections constant and independent of the input samples. However, both induce consequences, not least in their effects on the backpropagating gradients.

To begin, the approach of $\vec{x}/\|\vec{x}\|$ is equivilant to $\hat{x}$ which is a map which *loses* a representational degree of freedom, which cannot be counterbalanced with any gain in parameter degrees of freedom as these are fundamentally different quantities. Hence, this form of map amounts to $\mathbb{R}^n \to S^{n-1} \hookrightarrow \mathbb{R}^n$, showing it embeds a lower-dimensional hypersphere object back into the same space — *the magnitude information is lost.* Losing such a representational degree of freedom generally seems undesirable without a strong justification. Whereas the affine correction of *Eqn.* 32, does not flatten this dimension, preserving all the original representational degrees of freedom.

Furthermore, this latter equation scales both the weight and bias updates by the same factor, preserving their original magnitude proportions in the update.

This is not the case for the $\vec{x}/\|\vec{x}\|$ approach, which selectively scales down the weight update. Whether this is detrimental remains to be seen. Still, it would appear, again, that strong motivation would be needed to justify these differing proportioned updates. This is because *Eqn.* 26 will emphasise corrections to the, arguably less nuanced, bias term, acting as global shifts, in its contribution to the representational correction instead of the direction-dependent weights.

Finally, when analysing the weight updates, two differing terms are yielded, shown in *Eqn.* 38 for the normaliser-like correction, and *Eqn.* 39 for the affine correction, now written in standard matrix notation.

$$\frac{\partial \mathcal{L}}{\partial \mathbf{W}} = \frac{\vec{g}\vec{x}^T}{\|\vec{x}\|} = \vec{g}\hat{x}^T \tag{38}$$

$$\frac{\partial \mathcal{L}}{\partial \mathbf{W}} = \frac{\vec{g}\vec{x}^T}{\sqrt{\|\vec{x}\|+1}} = \left(\frac{\|\vec{x}\|}{\sqrt{\|\vec{x}\|+1}}\right)\vec{g}\hat{x}^T \tag{39}$$

One may consider that the unavoidable $\hat{x}$ of *Eqn.* 38 as having no good limiting behaviour, unlike the smooth behaviour required of isotropic functions. This may make it unstable or less-representative of the direction as $\|\vec{x}\| \to 0$. This is unlike *Eqn.* 39, which does possess smooth limiting behaviour. Moreover, the *parameter* updates close to zero contribute less to weight corrections, asymptotically the same as a typical affine update, and this may be desired in a magnitude-direction coding picture where small magnitude implies uncertain or absent semantics. Their influence grows initially approximately linearly until asymptotically limiting to a maximum magnitude dependence on $\|\vec{x}\|$. Hence, this likely also stabilises weight-gradient updates from large samples, while identically correcting for the affine divergence.

Overall, across all three reasons, this suggests that, at least in theory, the affine correction proposal is the preferred forward-pass map to account for the affine divergence terms. Both should be explored empirically as implementations for new maps, and could offer improved insight into the nature of normalisations.

In general, this distinction between theoretical ideal and effective updates may require much further consideration of where each is most appropriate and the assumptions that support each. To the best of the author's knowledge, explicitly highlighting this 'affine divergence' term as problematic appears novel, and its speculated explanatory mode for normalisation — it seems not to be in common discourse or a standard explanation related to such results. The implications of this are conceptually and theoretically significant and pertain to affine maps in general networks, not just to anisotropic and isotropic differences. However, this consideration between ideal and effective may similarly underlie the divergences between anisotropic and isotropic representations. Overall, this appears a fruitful direction for future study, and so far shows that the optimal update for parameters does not result in the optimal correction for the desired representations.

### C.3 Potential Pathological Consequences

There may be several pathological consequences to these foundational biases. This section will speculate on several.

At a high level, anisotropic forms may lead to a detectable tendency for representations to overly cluster around distinguished directions. This is due to the anisotropic nature of the functional form, where some angular arrangements may become overrepresented during optimisation because of their irregularity.

This task-agnostic over-density bias may be detrimental, as representations may tend to collapse towards single directions — aligned, anti-aligned, or others — and consequently lose informative representational degrees of freedom. This hypothesised loss in representational degrees of freedom may be damaging a network's internal expressiveness. This *quantising effect* of anisotropy on representations is hypothesised to also affect the semantics being represented, resulting in a limited representational capacity and a reduction to a simplistic neural coding scheme, such as approximately local coding.

This may be exacerbated when the inputs and outputs are better represented continuously, and transformations between them should preserve an approximately continuous structure. Consequently, isotropic primitives are expected to remove this form-induced task-agnostic discretising bias on representations and may be better suited where continuous semantics are desirable to retain — such as in generative architectures. However, more complex architectures require greater study of how foundational biases may emerge.

One heuristic is that the underlying distributions might be better represented continuously internally within the network. Avoiding function-induced collapse may enable greater concept-separability and, hence, manipulation, interpolation, and interactions of semantics. This may also align better with the frequently continuous real-world semantics being represented, such as broad object morphologies and arrangements, multivariate signals, lighting and shading, continuous expressions, and modelling distributions, etc. Therefore, enabling latent variables and representations to be continuous and potentially more interpolatable across various levels of a network, rather than imposing a task-agnostic structure, may be beneficial for many tasks. This may increase representational capacity and organisation of the representations.

Overall, this strongly relates to the optimal coding schemes for networks. Axis-aligned representations, previously observed to frequently emerge (Bau et al., 2017; Elhage et al., 2022; Bird, 2025b), approximate a local coding scheme (Sherrington, 1940; Konorski, 1968; Barlow, 1953; Gross, 2002; Connor, 2005) for the network. This is where each neuron corresponds to a single semantic. Yet, from this study, it is highly suggestive that the functional form choices induce these observations and are thus not fundamentally 'desirable' to optimisation. This coding scheme has reduced concept interference but has a much lower representational capacity than other coding schemes. Models have been shown to produce other interference-capacity balances depending on input sparsities (Elhage et al., 2022), suggesting that axis-alignment is not an appropriate *universal* inductive bias — further motivating the study of alternative functional-form inductive biases and their effects on representations.

A functional form that does not distinguish particular directions may therefore not deter such continuous, interpolatable structures present in the dataset, and this may be advantageous to some applications. Yet, sometimes discretisation may be appropriate, such as in classification networks as discussed. In these scenarios, abstracting away redundant detail can be beneficial — though in autoencoding models this would not be an expectation. In any case, rigidly imposing this through a task-agnostic, function-driven, inductive bias about a specific arrangement might be detrimental.

For example, when reducing the dimensionality of a dataset inherently containing discrete representations via a linear layer, to some arbitrary lower dimension, one would not expect it to be always appropriate for those discrete directions to be best represented through an alignment to an orthonormal basis or other imposed geometry for that space — as these are as arbitrary to the bottleneck chosen.

Other speculative pathological modes are attributed to hypothesised functional-form quantisation. For example, discrete representations may be particularly susceptible to perturbations that move activations outside the cluster into an in-between region of representation space. Maps from these unpopulated regions may be poorly trained and fragile due to their training-time rarity and, therefore, highly unpredictable in their resultant maps. This may suggest that such a quantising inductive bias decreases adversarial robustness (Goodfellow et al., 2015; Moosavi-Dezfooli et al., 2017; Ilyas et al., 2019; Tsipras et al., 2019), whereas removing this bias through isotropy may be advantageous. However, this speculation is not explored within this work.

Overall, there may be several implications of the demonstrated symmetry-led foundational biases in representations, not least the review of functional forms, symmetry-led primitive design, and symmetry-taxonimisation of effects, but also the potential for broader impacts on network robustness, adversarial fragility, coding schemes, representational capacity, and optimisation, among others. These speculative connections are considered worthwhile to explore to better understand the behaviours exhibited by networks and how they might be leveraged. It already suggests that the induced quantised representations of discrete-group defined primitives, such as existing permutations, may have pathological consequences, and a preliminary exploration of this is conducted in the subsequent section.

# D  Analysis of Performance and Pathological Indications

This section displays the reconstruction errors for several of the networks tested. Normalised and isotropic networks tend to outperform all other networks, though several exceptions do occur. The results should be taken as preliminary, since there were fewer CIFAR reconstructions computed using unnormalised datasets, due to limited compute resources and performance not being the primary study of this paper — particularly since Isotropic-Tanh is considered a placeholder activation function which is likely suboptimal and only made to appear superficially similar to standard-Tanh (Bird, 2025a). Nevertheless, they indicate a general trend in which isotropic networks tend to outperform anisotropic networks on this task.

These results are of particular interest as isotropic-tanh regularly outperforms conventional tanh. This is despite the preexisting ecosystem of support implicitly developed over the prior eighty years for anisotropic functions, covering function selection, architectural developments, and hyperparameter tuning and optimisation processes. These isotropic functions appear to outperform on these tasks, despite being entirely unoptimised placeholders, not undergoing any form of search, merely superficially resembling tanh for the purpose of this ablation, and lacking the broader ecosystem to support them. Although preliminary, it is felt that this is both promising and intriguing, and supports the development of better-suited isotropic functions which may yield further improved performance.

Within normalised networks, isotropic activation functions *always* outperform otherwise identical anisotropic networks through training. In unnormalised networks, they frequently tie or vary slightly, with instances where anisotropic networks do perform better. However, from the results gathered, unnormalised networks typically perform poorer overall — an exception to this is demonstrated.

The unnormalised and normalised instances can be compared by correcting unnormalised reconstruction errors, with the same normalisation steps to make the two forms of errors comparable. When this occurs, normalised-isotropic networks tend to always outperform the three other combinations: "unnormalised-isotropic", "normalised-anisotropic" and "unnormalised-anisotropic". Though when exceptions do occur, it tends to be "unnormalised-isotropic" slightly outperforming "normalised-isotropic", and sometimes "unnormalised-anisotropic" outperforming all others, but this is relatively infrequent.

Each plot shown represents a specific architecture, with isotropic and anisotropic activations plotted in different colours. Sometimes with their normalised and unnormalised counterparts overlaid where relevant. There are five samples per variant (each plotted faintly), along with the mean and standard deviation. These are all for the unseen testing-set reconstruction error against the number of epochs trained.

Figure 31 displays all the normalised results obtained for CIFAR reconstruction. The isotropic networks show significantly lower reconstruction error in all cases for *Fig.* 31.

In *Fig.* 32, the results between normalised and unnormalised networks for CIFAR reconstruction are overlayed.

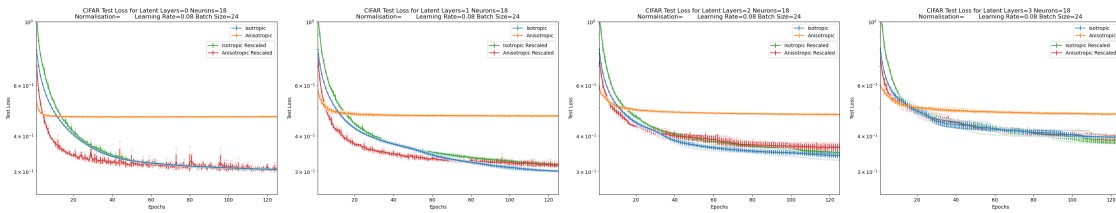

Figure 32: These plots show the tanh-like normalised anisotropic (orange) and normalised isotropic (blue) overlayed over the unnormalised anisotropic (red) and unnormalised isotropic (green) reconstruction errors on test-set CIFAR against the number of training epochs. The networks have a width of 18 neurons, and their depth increases to the right. In the leftmost plot, unnormalised isotropy has the lowest error by a very slight margin. The central two plots show that normalised isotropic networks have the lowest error by a large margin. The rightmost plot again shows unnormalised isotropy with the lowest error and a larger margin. Overall, a trend is not apparent, but isotropy tends to outperform by a greater margin.

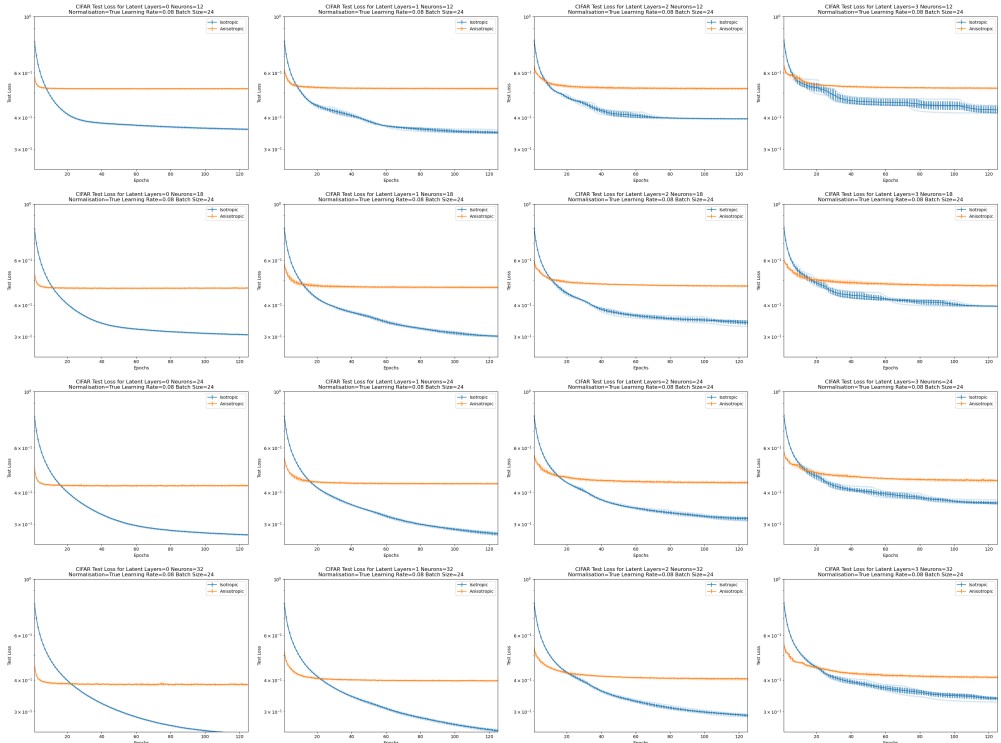

Figure 31: Displays the tanh-like test-set reconstruction errors across all normalised-CIFAR experiments using an autoencoder reconstruction task. Network depth increases along rightward columns, whilst width increases along downwards rows. In every case, isotropic networks (in blue) exhibit the lowest reconstruction error; therefore, they are the best-performing activation functions. Per graph, the axes show reconstruction error against the number of epochs trained for. These graphs are scaled identically, and one can see that width tends to lower the reconstruction error, while depth often increases it. Repeats are very consistent, showing very small deviation between repeats.

Notably, in every example, the plots for unnormalised anisotropic networks featured significantly greater noise in reconstruction error on the testing set than the unnormalised isotropic networks. This may be indicative of smoother training dynamics, indicated for isotropic functions (Bird, 2025a). However, this observation is insufficiently rigorous to draw a conclusive statement regarding smoother optimisation dynamics in isotropic networks.

The tendency for unnormalised anisotropic networks to outperform normalised anisotropic networks may be due to the unnormalised dataset displaying a greater degree of initial anisotropy. This is because the unnormalised samples are bound between $[0,1]^{32\times32\times3}$ and rather densely cover this space. Hence, the unnormalised dataset is strongly anisotropic, only populating these directions from the origin. This may be better leveraged by the anisotropic primitives, and hence demonstrate improved reconstruction error. This is in contrast to the normalised dataset, which is more uniformly distributed about the origin, as shown in *App.* A.2. This may also explain why isotropic networks perform better on these normalised cases. However, this does not appear to explain the later Leaky-ReLU results of *Fig.* 36, where anisotropic Leaky-ReLU performed more poorly on the unnormalised case compared to the normalised case — though this may be because the normalised inputs resulted in initialised activation which better straddle the non-linear primitive's piecewise boundary.

An instance of 32 neurons was also computed, and this example did show unnormalised-anisotropic functions performing better as shown in *Fig.* 33. However, these were not the best-performing models overall.

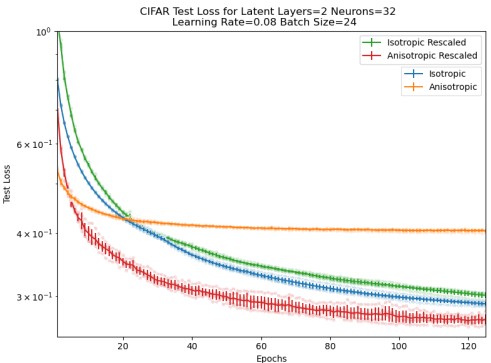

Figure 33: This plot shows the tanh-like normalised anisotropic (orange) and normalised isotropic (blue) overlayed over the unnormalised anisotropic (red) and unnormalised isotropic (green), reconstruction error on test-set CIFAR against the number of training epochs. The networks have a width of 32 neurons, and 2 hidden layer blocks. In this counterexample, the unnormalised-anisotropic network had the lowest reconstruction error.

Similar findings were observed for MNIST examples, as shown in *Fig.* 34, which also features Tanh-based networks.

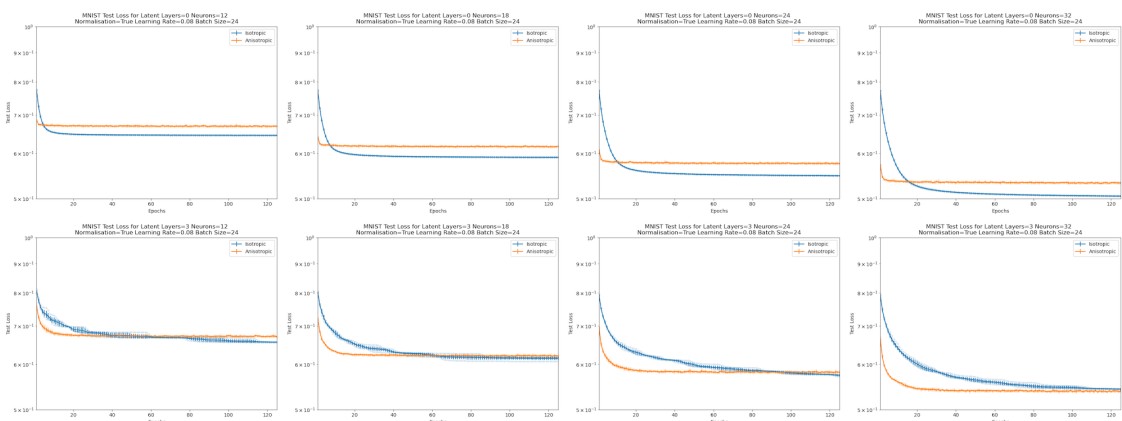

Figure 34: This plot shows the tanh-like normalised anisotropic (orange) and normalised isotropic (blue) reconstruction errors for test-set MNIST against the number of training epochs, network width increases rightward over columns, and depth increases downward over rows (flipped ordering from *Fig.* 31). It can be observed that reconstruction error is lower in wider and less-deep networks. Isotropic networks outperform anisotropic networks in every case, except for the 3-hidden-layer block and 32 neurons, where anisotropic networks exhibit a slight marginal success. The best performing network is the isotropic network with 0 hidden layer blocks and 32 neurons — this network is also thought to have the greatest representation capacity.

For the unnormalised against normalised case tested in MNIST networks, isotropic networks continue to outperform with a significant margin, as seen in *Fig.* 35.

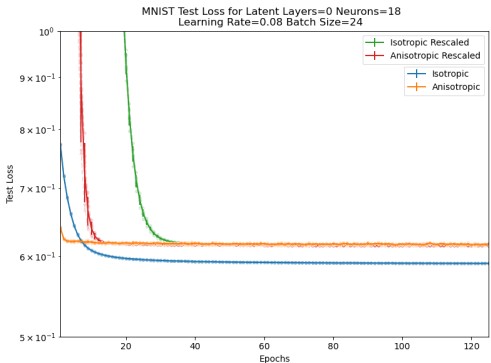

Figure 35: These plots show the tanh-like normalised anisotropic (orange) and normalised isotropic (blue) overlayed over the unnormalised anisotropic (red) and unnormalised isotropic (green), reconstruction errors on test-set MNIST against the number of training epochs. The networks have a width of 18 neurons, and a depth of 0 hidden layer blocks. It can be observed that all networks tend to a similar reconstruction error, except for the normalised-isotropy graph, which reaches a significantly lower reconstruction error.

Additionally, in all Leaky-ReLU cases, the isotropic networks demonstrated significant improvements. This is shown in *Fig.* 36.

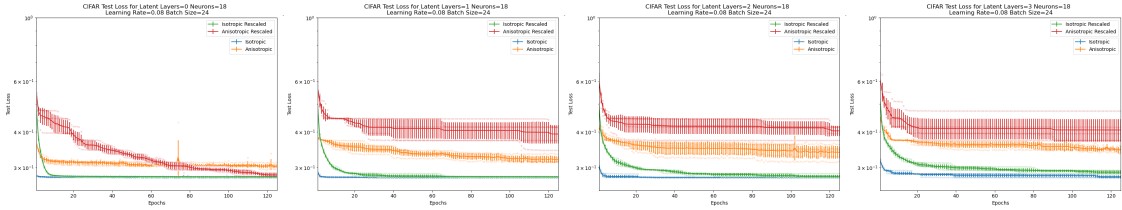

Figure 36: These plots show the Leaky-ReLU-like normalised anisotropic (orange) and normalised isotropic (blue) overlayed over the unnormalised anisotropic (red) and unnormalised isotropic (green) reconstruction errors on test-set CIFAR against the number of training epochs. The networks have a width of 18 neurons, and their depth increases to the right. In all instances, normalised or unnormalised isotropic-Leaky-ReLU reaches the lowest reconstruction error by a large margin. Anisotropic networks also appear to exhibit significantly greater noise and sample-to-sample variation in test-set accuracies across the differing training epochs.

Overall, this shows a clear tendency for isotropic-defined functions to outperform anisotropic counterparts across most tests, with only limited exceptions. This is greatly interesting, as the isotropic functions are considered only a placeholder, intended to imitate the appearance of the anisotropic counterparts superficially. Thus, they have not undergone a selection process to optimise their functionality, leveraging their distinctly different isotropic properties, so they can likely be improved substantially. Moreover, they are not reliant on an ecosystem of support, which has long developed implicitly around their anisotropic counterparts. These results are a promising indication, but remain preliminary, and are so far only applicable to autoencoding reconstruction, where one may expect isotropy to be particularly favourable in preserving information for reconstruction.

These improvements may not extend to larger models such as transformers. For example, the structure of these models may also be partially a product of selection over anisotropic primitives, and may constitute a circular reinforcement of such forms (Bird, 2025a). Therefore, these improvements for isotropic primitives may not be applicable. A careful approach, and perhaps symmetry-principled redesign/rediscovery, of larger models may be necessary for determining isotropy improvements. Nevertheless, these preliminary results offer

some support for their adoption in autoencoding reconstruction tasks, even for these illustrative placeholder activation functions, which were not intended for actual usage (Bird, 2025a).

# E    Network Architectures and Training

This section discusses the various details required to reproduce these results, including architectures, hyper-parameter details and how the plots were generated. Besides the unusual isotropic activation function, all training details are felt to be conventional choices.

Autoencoder networks are studied in these results, due to their lack of one-hot classification head, which could impose anisotropy on results[14]. The autoencoders are comprised of an encoding, decoding, and hidden section. Separately categorising the 'hidden' layer is perhaps unorthodox but important for the study. All networks tested are derived from the one shown in *Fig.* 37. This diagrammatic depiction is consistent with open-sourced standardisation available at [REDACTED FOR ANONYMITY].

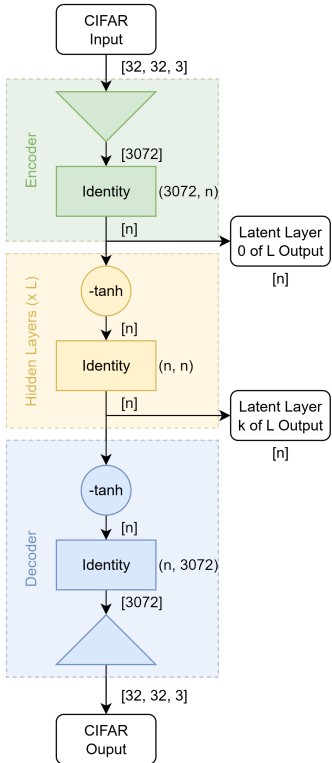

Figure 37: Shows an autoencoding architecture for reconstruction of CIFAR data, in a standardised diagrammatic key system. It is comprised of three blocks: encoder, hidden layers and decoder. The encoder flattens the tensor representations into a vector of dimension 3072 and then performs an affine transformation down to n dimensions. Following this, each hidden layer consists of the application of an activation function, which is either standard or isotropic-tanh, followed by another affine transformation from n to n dimensions. This hidden layer process is repeated $L$ times. Finally, the decoder applies the chosen activation function again, followed by an affine map from n to 3072 dimensions, before reshaping the vector into its classical representation. Various latent layers are drawn from the model, particularly after the encoder, denoted as layer 0, or after a hidden-layer block, denoted as $k$. Therefore, all latent layers of the study are $n$-dimensional. The parameters of all affine maps are optimisable through gradient descent.

This depicts the architectures used across all results, where $n$ varies in width and $L$ varies in depth of the autoencoder model. The width $n$ was kept relatively small, as the hypervolume grows exponentially with the

---

[14]Some anisotropy may remain due to the spatialised nature of the image data; however, this is inherent to the data and the nature of visual datasets. Residual anisotropy through data is not the remit of the hypothesis being tested nor the overarching algebraic symmetry framework being studied. Though its effect was considered and discussed in *App.* A.

width. This resulted in the relative density of isotropic activations, which appear to more fully occupy this hypervolume, falling exponentially with width. Therefore, this limited the experiments to the arbitrarily chosen $n \leq 32$ to strike a suitable balance between ensuring good visualisations of isotropic activations and exploring varied widths.

The latent layer of study is indexed by integers $k \in [0, L]$. For MNIST (LeCun et al., 2010) models, the tensor representations are changed from $[32, 32, 1] \to [28, 28, 1]$ and $[3072] \to [784]$ in all instances, with corresponding changes in affine maps. This ensures these results follow from a similar and comparable model.

Finally, following from *App.* A, the MNIST and CIFAR datasets are sometimes normalised over their training sets to reduce statistical geometry artefacts. This involves an element-wise calculation of the mean tensor over all training samples, as well as for the standard deviation statistic. Where the standard deviation is found to be 0, those elements are replaced with the value 1. When normalisation is used, every ingoing sample per batch is standardised by subtracting the mean and dividing by the standard deviation, where the non-zero standard deviation is made equal to one.

Throughout all experiments, a batch size of 24 was maintained, alongside a learning rate for PyTorch's (Paszke et al., 2019) momentum gradient descent, with momentum 0.9, a learning rate 0.08, and using a mean-squared error loss for reconstruction. Parameter initialisation was the standard Xavier-Normal (Glorot and Bengio, 2010) for weights and zero-initialised for biases, this is due to their left-, right- probabilistic invariances (Bird, 2025a). However, standard multivariate normal or orthogonal initialisation would have been equally valid choices. Still, Xavier-Normal was deemed more conventional, though its gain is only optimised for anisotropic activation functions (Bird, 2025a). All values were kept in float64.

Networks were trained for a total of 125 epochs; however, intermediate measurements were gathered after training for each of the incremental following amount of epochs: $[1, 1, 2, 2, 2, 5, 5, 6, 6, 10, 10, 10, 15, 15, 15, 20]$ corresponding to a cumulative epoch total of $[1, 2, 4, 6, 8, 13, 18, 24, 30, 40, 50, 60, 75, 90, 105, 125]$. This was chosen arbitrarily, but was found to give meaningful insight into the formation of the representation structures.

Data was gathered using combination-PPP, with $\epsilon = 0.75$, which was found to be a good overall value. Plots were compiled using Gaussians centred at each representation's projected coordinates, and the colour map indicates representation density per region, with a maximum value of the mean plus five standard deviations and a minimum of zero. Additionally, in the anisotropic cases, so many representations became aligned with the axes that computing the Gaussian maps became intractable due to the number of computations. Therefore, a random subset of the samples generated by PPP was plotted. The size of this subset was chosen to be the closest integer of the larger set raised to the power of 0.9. Hence, the number of representations plotted monotonically increased with the number of PPP samples, yet remained computationally tractable for very large numbers of representations. This approach was used in *all* cases.

## F    Mathematical Notation Glossary

This section will outline the notations used across equations.

- Various groups, $S_n$, $D_n$, $B_n$, $\mathrm{O}(n)$, denote the permutation, even-sign-flip permutation, hyperoctahedral and orthogonal groups respectivly.

- **P** would indicate a matrix representation of a permutation group, similarly **R** would indicate a matrix representation for an orthogonal group. These are in standard representations. Mapping a group element to a matrix representation can be denoted by a function $\rho : \mathcal{G} \to \mathrm{GL}_n(\mathbb{R})$, which indicates a map from the set of group elements $\mathcal{G}$ to a $n \times n$ invertible matrix that is a member of the real General-Linear, GL, group.

- Members of sets are denoted $g \in G$, and usual set-notation is assumed, such as subsets $A \subset B$.

- Multivariate functions such as $\mathbf{f}$, provide a map as follows $\mathbf{f} : \mathbb{R}^n \to \mathbb{R}^m$, with dimensionality $n$ and $m$ defined.

- Standard basis vectors are denoted $\hat{e}_i$, where this is the $i^{\text{th}}$ vector of the basis set $\{\hat{e}_i | \forall i\}$ or sometimes denoted $\{\hat{e}_i\}_{\forall i}$ in shorthand.

- Vector hats, $\hat{x}$, denote unit-normalised vectors, compared to unnormalised general vectors given by $\vec{x}$. Vectors tangent to others may be denoted $\vec{x}_\perp$, whilst parallel vectors denoted by $\vec{x}_\parallel$.

- The vector norm is given by $\|\vec{x}\|_2 \equiv \|\vec{x}\|$. This generalises to $l$-norms when explicitly given as $\|\vec{x}\|_l$

- In general vector spaces may be denoted $\mathbb{R}^n$ or with a particular bound $[0,1]^n$, and may be generalised with multiple indices to $\mathbb{R}^{n \times m \times i \times \cdots}$. This extends to spaces such as $\mathcal{S}^n$, which is the unit-norm hyper-sphere, where vectors are all unit-normalised.

- Jacobians are denoted $\mathcal{J}^{\mathbf{f}}$ for their respective function $\mathbf{f}$.

- The Einstein summation convention is used for sums and indices.

- The identity matrix is denoted $\mathrm{I}$.

- Hadamard products are denoted as $\odot$.

- The operation diag produces a diagonal matrix, with entries specified by the input to the diag operation, generally this is given by $\mathrm{diag}\left(\vec{x}\right) = \left(\vec{x}\vec{1}^T\right) \odot \mathrm{I} = \left(\vec{1}\vec{x}^T\right) \odot \mathrm{I}$ or symmetrised to $\frac{1}{2}\left(\vec{1}\vec{x}^T + \vec{x}\vec{1}^T\right) \odot \mathrm{I}$, where $\vec{1}$ is the vector of 1s.

