# OpenReview forum: "Emergence of Quantised Representations Isolated to Anisotropic Functions"
_TMLR — Rejected by TMLR_

### Review · Reviewer_bF6V · 2025-09-13

**Summary Of Contributions:**

### Summary Of Contributions

This paper introduces and empirically validates the compelling hypothesis that the algebraic symmetries of network primitives, specifically activation functions, act as a powerful, task-agnostic inductive bias that shapes the geometry of learned representations.

The primary contribution is a rigorous comparative study between conventional "anisotropic" activation functions (defined by discrete, permutation-based symmetries like $S_n$ or $B_n$) and novel "isotropic" counterparts (defined by continuous, orthogonal $O(n)$ symmetry). Using a new visualization tool, the Privileged-Plane Projective (PPP) method, the authors demonstrate that anisotropic functions consistently induce a "quantization" effect, causing representations in an autoencoder to collapse into discrete, axis-aligned clusters. In stark contrast, isotropic functions preserve a more continuous and distributed representational structure.

The authors argue that this function-driven quantization provides a fundamental causal mechanism for several widely-observed interpretability phenomena, including grandmother neurons, discrete linear features, and representational superposition, suggesting they may be artifacts of current design choices rather than fundamental properties of deep learning.

**Key Strengths:**

* **Novel and Insightful Hypothesis:** The core idea of linking the maximal algebraic symmetry of a function directly to emergent representational structures is highly original and provides a powerful new lens for understanding deep learning models from first principles.
* **Strong and Convincing Empirical Evidence:** The paper's claims are supported by extensive and visually clear experiments across various model widths, depths, and datasets. The controlled ablation study effectively isolates the influence of the activation function, making a strong case for the causal link.
* **Valuable Methodological Contribution:** The proposed PPP method is a useful tool for the community, offering an intuitive and high-fidelity way to visualize and analyze the structure of high-dimensional embeddings.
* **Significant Implications:** The findings have broad relevance to the fields of interpretability, AI safety, and representation learning, offering a potential explanation and a new mode of control over complex emergent phenomena.

**Key Weaknesses:**

* **Limited Scope of Experiments:** The study is confined to autoencoder models on image reconstruction tasks. While excellent for isolating the phenomenon, it leaves open the question of how these inductive biases manifest in more complex architectures (like Transformers) and other tasks (e.g., classification, language modeling).
* **Preliminary Performance Claims:** The evidence that isotropic functions lead to better performance (lower reconstruction error) is presented as preliminary. While promising, a more comprehensive analysis would be needed to make a definitive claim about their practical benefits.
* **High-Level Causal Mechanism:** While the paper establishes a strong causal *link*, the precise dynamics *through which* optimization exploits these symmetries to shape the representation geometry (e.g., via gradient flow) remains at a more heuristic level of explanation.

**Additional Comments:**

This is a fascinating and well-executed study that addresses a fundamental and surprisingly underexplored question: what is the origin of discrete, interpretable structures in neural representations? The central hypothesis—that the algebraic symmetry of activation functions acts as a primary inductive bias—is both elegant and powerful. The paper provides a compelling causal mechanism for phenomena that are often observed but not well understood, and the connection drawn to a wide range of interpretability research is a significant contribution.

The empirical work is exemplary. The use of a carefully controlled ablation study is the correct methodology for this type of investigation, and it successfully isolates the effect of the activation function. The evidence provided through the novel PPP method is visually striking and unambiguous. The authors’ diligence in testing the hypothesis across different architectures, datasets, and function families, as well as their transparent handling of potential methodological artifacts, makes the core claims highly credible.

My recommendation for acceptance is contingent on one minor but critical clarification regarding the use of unnormalized data in the main figures, as detailed in the "Requested Changes" section. Addressing this will further bolster the paper's rigor. My other suggestions—regarding a discussion on generalization, the underlying optimization dynamics, and the nuance of the performance claims—are intended to strengthen what is already a very strong submission.

Overall, this is a high-quality, insightful, and impactful paper that I believe will be of great interest to the TMLR audience. It has the potential to shift how the community thinks about the role of the most basic components of our models. Congratulations to the authors on this excellent work.

**Audience:**

Yes

**Audience Explanation:**

The findings of this paper would be of significant interest to a broad cross-section of the TMLR audience, as they touch upon fundamental questions in several key areas of machine learning research. The work's appeal is not limited to a niche subfield; rather, it provides a foundational perspective that is relevant to both theoretical and applied researchers.

Specifically, the paper would be of high interest to the following communities:

* **Interpretability and Explainable AI (XAI):** This is perhaps the most directly impacted audience. The paper offers a novel, first-principles causal mechanism for the emergence of widely studied phenomena like grandmother neurons, axis-alignment, and discrete representational superposition. Researchers in this area are constantly seeking to understand *why* these structures form, and this paper provides a compelling, testable hypothesis that they are consequences of the algebraic symmetries of common network components.
* **Representation Learning:** The work provides fundamental insights into how latent representations are shaped by architectural choices. It challenges the idea of representational uniformity and demonstrates that strong, predictable geometric structures can be induced simply by the choice of activation function. This is core to the goals of understanding and controlling learned representations.
* **AI Safety and Robustness:** The paper speculates on how function-driven quantization might impact model safety. The creation of sparsely populated regions between discrete clusters could make models vulnerable to adversarial attacks, as these "untrained" areas may have unpredictable behavior. The idea that this vulnerability can be mitigated by using isotropic primitives would be of great interest to this community.
* **Geometric Deep Learning and Symmetry:** The paper's core framing around the algebraic symmetries of functions (e.g., permutation vs. orthogonal equivariance) aligns directly with the principles of geometric deep learning. It explores how these symmetries, even at the level of individual primitives rather than the entire network, act as powerful inductive biases, a concept central to this field.
* **Theory and Foundations of Deep Learning:** The work probes the fundamental "why" of deep learning. It posits that the symmetry of functions propagates into the geometry of representations, a candidate for a general principle of representation learning. This type of foundational inquiry is valuable for theorists aiming to build a more complete understanding of neural networks.
* **Model and Architecture Design:** The findings have direct practical implications. They suggest that the choice of activation function is not a minor detail but a critical design decision for imparting specific inductive biases. This could lead to more principled approaches for designing architectures tailored to specific data types or desired representational properties.

Because it connects a fundamental and often overlooked design choice to a wide range of high-impact research areas, this paper is likely to stimulate considerable discussion and follow-up work within the TMLR community.

**Broader Impact Concerns:**

This paper presents fundamental research on the internal mechanisms of neural networks and does not pose immediate, direct broader impact concerns. The work does not introduce a new application or dataset that could be readily misused.

However, the findings have a significant dual-use potential related to **model interpretability and obfuscation**, which warrants a discussion in a Broader Impact Statement.

The core result of the paper is that conventional anisotropic primitives tend to create discrete, structured, and thus more easily interpretable representations, while the proposed isotropic primitives create continuous, distributed, and potentially less interpretable representations (at least with current tools). The authors themselves note that isotropic representations "act in opposition to the assumptions of many interpretability approaches."

While this can be seen as a path towards more "natural" or higher-capacity representations, it also provides a clear recipe for creating intentionally opaque systems. In sensitive domains like finance, medicine, or justice, where model transparency and accountability are critical, the choice to use primitives that inherently make a model harder to audit is an ethical concern. An actor could use isotropic functions to build a high-performing model that is functionally a "black box," making it difficult to diagnose biases or unfair decision-making processes.

Therefore, the authors should include a Broader Impact Statement to address this implication. This statement should discuss:
1.  The potential for these findings to be used to intentionally design less interpretable models.
2.  The importance of considering the trade-off between potential performance/robustness gains from isotropy and the critical need for transparency in sensitive applications.
3.  A call for parallel research into new interpretability methods specifically designed for the continuous, distributed representations generated by isotropic models, to ensure that progress in model capability does not come at the cost of accountability.

**Claims And Evidence:**

Yes

**Claims Explanation:**

The claims made in this submission are supported by evidence that is exceptionally **accurate**, **convincing**, and **clear**. The authors have built a robust case for their central hypothesis through a well-designed and thoroughly executed empirical investigation.

Clarity of Evidence

The evidence presented is remarkably clear, primarily due to the effectiveness of the proposed **Privileged-Plane Projective (PPP) method**.

* **Intuitive Visualization**: The PPP plots provide a direct and visually intuitive representation of high-dimensional data geometry. The primary finding—the stark contrast between the sharp, axis-aligned "beams" from anisotropic functions and the continuous, ring-like distributions from isotropic functions—is immediately apparent without needing complex statistical analysis.
* **Unambiguous Results**: The difference between the experimental conditions is not subtle; it is a qualitative shift in the entire representational structure. This removes ambiguity and makes the central claim easy to verify visually from the provided figures.
* **Temporal Evolution**: By showing the PPP plots at various stages of training, the paper clearly illustrates *how* these discrete structures emerge and refine over time in the anisotropic case, while the isotropic representations remain distributed.

---
Convincing Nature of Evidence

The evidence is highly convincing due to the rigor of the experimental methodology and the robustness of the findings.

* **Controlled Ablation Study**: The paper's core strength is its use of a controlled ablation study. By isolating the activation function as the *only* variable altered between experimental setups, the authors make a compelling case for a causal link between the function's symmetry and the emergent representation structure.
* **Robustness Across Conditions**: The central phenomenon is shown to be highly robust. It is not an isolated artifact but is consistently reproduced across:
    * **Different function families**: The effect is demonstrated for both tanh-like and Leaky-ReLU-like functions. The consistency in the Leaky-ReLU case is particularly powerful, as it rules out alternative explanations related to saturation or vanishing gradients, strengthening the claim that algebraic symmetry is the primary causal factor.
    * **Varying Architectures**: The results hold for autoencoders of different widths and depths.
    * **Multiple Datasets**: The findings are consistent on both CIFAR and MNIST datasets, suggesting the effect is not dataset-specific.
* **Ruling Out Trivial Explanations**: The authors shrewdly conduct measurements on latent layers that *precede* any activation function in the forward pass. This demonstrates that the effect is a non-trivial result of the optimization process influencing the entire network, not just a simple geometric bounding of the output space.
* **Methodological Transparency**: The authors dedicate an entire appendix to discussing and demonstrating potential statistical artifacts of the PPP method itself. This transparency and rigor in addressing potential confounders significantly increases confidence in the main results.

---
Accuracy of Evidence

The evidence appears to be accurate and faithfully reported.

* **Reproducibility**: The paper provides detailed descriptions of the network architectures, training procedures, and hyperparameters, which should be sufficient for other researchers to reproduce the experiments.
* **Code Availability**: The authors state that a code repository is available (redacted for anonymity), signaling a commitment to transparency and allowing for independent verification of the results.
* **Direct Interpretation**: The conclusions drawn in the text are a direct and accurate interpretation of the data presented in the figures and graphs. The authors do not appear to overstate their claims and are careful to label secondary findings (like performance improvements) as "preliminary".

**Requested Changes:**

Below are the requested changes for the submission. They are separated into two categories: critical changes that are necessary for my recommendation of acceptance, and suggestions that I believe would further strengthen the work.

Critical for Acceptance

1.  **Clarify the Role of Data Normalization in the Main Text.** The main experimental results (e.g., Figure 1) are presented using unnormalized data. However, Appendix A provides a thorough and important discussion on how data normalization helps mitigate statistical artifacts in the PPP method. While the core findings remain convincing even with unnormalized data, the paper's rigor would be significantly enhanced by explicitly addressing this in the main text. **Please add a brief statement in Section 3 (Experiments and Interpretation) acknowledging that the main figures use unnormalized data for demonstration, and direct the reader to Appendix A for results with normalized data and a detailed analysis of methodological artifacts.** This small addition would proactively address potential questions about the methodology and strengthen the reader's confidence in the conclusions.

---
Suggestions to Strengthen the Work

1.  **Expand Discussion on Generalizability.** The paper's focus on autoencoders is a key strength, as it provides a clean, controlled environment to establish the core phenomenon. To further enrich the discussion, I suggest adding a paragraph in the Conclusion or a new Future Work section that speculates on how these findings might translate to other domains. For example, how might these function-driven biases interact with the strong permutation symmetry of a one-hot classification output layer? Or how might they manifest in the attention layers of a Transformer? Discussing these open questions would broaden the paper's impact.
2.  **Elaborate on the Potential Optimization Dynamics.** The paper convincingly establishes a causal link between symmetry and representational geometry but is more heuristic about the precise mechanism. The work could be strengthened by adding a brief discussion on the potential role of **gradient flow**. For instance, the authors could hypothesize about how the gradients of anisotropic, elementwise functions (which are often axis-aligned or sparse) might create preferential optimization pathways toward the standard basis, a dynamic that would be absent in the radially symmetric gradient fields of the proposed isotropic functions. This would add a deeper layer of theoretical intuition to the empirical results.
3.  **Add Nuance to the Performance Discussion.** The preliminary results showing isotropic functions outperform their anisotropic counterparts are intriguing. The discussion could be made even stronger by briefly acknowledging that decades of research into architectures and hyperparameter tuning have been implicitly optimized for standard anisotropic functions. Mentioning that the observed performance gains for the novel isotropic functions occur *despite* this ecosystem-level bias would add valuable context and further highlight the potential of this research direction.

---

> ### Author Response · Authors · 2025-11-12
> **Response to Reviewer bF6V**
>
> Dear Reviewer bF6V,
>
> Thank you for taking the time to provide this thorough review and helpful suggestions to improve the quality of the presented manuscript. These suggestions have been particularly insightful, and this level of engagement with the work is appreciated. In this response, it is outlined how all suggestions will be implemented into the updated version of the manuscript.
>
> Regarding the requested changes, and then a brief discussion of the mentioned weaknesses.
>
> Thanks for suggesting that a more thorough discussion of the role of data normalisation should be included in the main text, and that correcting this would be critical for paper acceptance. This has now been undertaken in the updated manuscript; it appears in the second paragraph of section 4: "Experiments and Interpretation". These considerations are now explicitly discussed in the main text to inform the reader preemptively.
>
> The further suggestions of: 1) expanded generalisability discussion, 2) optimisation dynamics, and 3) performance nuances, have been similarly included, in Appendix C.1, C.2 and D, respectively. The addition of these proposed sections substantively enriches the paper. The addition of a discussion of gradient flow now moves the exact mechanism from more heuristic to mechanistic for each function. This complements a newly created section 2: "Theoretical Framework and Considerations", which goes into further depth on the heuristic mechanisms which may induce the observed phenomenon. The nuance regarding performance has been a great addition to the paper, clarifying the existing ecosystems that provide a supporting bias toward anisotropy, thus making it more explicit that this is a surprising result. This will undoubtedly improve the quality of the discussion in the revised manuscript.
>
> Future work regarding the mentioned weaknesses: it is agreed that the experiments focused on reconstruction tasks, as was necessary for these particular ablation trials. Now that the association has been established on this example, it would be beneficial to broaden the search into other, more complex architectures. This is aimed to be a follow-up to the current paper, exploring the interplay between one-hot classification encoding and the potential link to Neural collapse through the same mechanisms, as well as architectures such as transformers and convolutions --- which are discussed in the new Appendix C.1. The latter may require careful discernment between kernel-wise architectural permutation symmetries and those stemming from the functional forms. This will be a beneficial direction and broadly impactful to explore, as suggested. It is agreed that a more extensive study would be required to thoroughly assess the performance of isotropic implementations as suggested, where gradient flow may help establish exactly how this and other phenomena are induced in each network and support the search for optimal isotropic functions.
>
> In hindsight, the indirect potential for this symmetry-led approach to obfuscate model interpretability does warrant a broader impact statement, and how its representations oppose most current interpretability methods. Not considering this possibility was an oversight in the original manuscript, and the broader impact statement has now been included in section 2.2 to clarify and suggest research directions to mitigate this possibility. All three proposed questions have been considered and discussed in the revision. Thank you for bringing this to my attention; it's a possibility I did not foresee.
>
> Overall, it is pleasing that the idea that algebraic symmetries of activation functions have repercussions on representations was of interest and offered a new insight into the organisation of deep learning models. It seems the methodological nuances and ablation studies across the various settings were sufficiently robust to support the conclusions, with the use of autoencoders noted as a key strength. It is also reassuring that you agree with the broad implications this may yield for the variety of subdisciplines mentioned.
>
> It is encouraging that you feel this is a fundamental and overlooked choice with the potential to have a high impact across numerous research areas, likely to stimulate considerable discussion within the TMLR community and to prompt follow-up research into this emerging avenue.
>
> I thank you for the positive and thorough feedback on the paper, and I appreciate the productive suggestions surfaced in your review. It is hoped that, with the corrections noted in the revised manuscript, this paper will be a worthwhile publication for the TMLR community.

---

> > ### Comment · Reviewer_bF6V · 2025-12-07
> >
> > Thank you for your detailed and constructive response. I appreciate your thorough engagement with the review comments and your willingness to incorporate the suggested changes.
> >
> > I am pleased to confirm that the inclusion of the discussion on data normalization in the main text (Section 4) addresses my critical request for acceptance. Furthermore, the additions regarding generalizability, optimization dynamics (gradient flow), and performance nuances in the Appendices, along with the new Broader Impact Statement in Section 2.2, significantly strengthen the manuscript.
> >
> > These revisions add important transparency, ethical consideration, and theoretical depth to what was already a compelling empirical study. I remain confident that this work will be a valuable contribution to the TMLR community and look forward to seeing the final version published.

---

### Review · Reviewer_qrKh · 2025-10-03

**Summary Of Contributions:**

### Summary
> This paper investigates the causal influence of algebraic symmetries on deep neural networks’ latent representations using a Privileged-Plane Projective (PPP) method. PPP improves upon the Spotlight Resonance Method (SRM) by preserving both angle and magnitude and by visualizing the structure of distinguished directions in representation space. Using PPP, the paper shows that discrete linear features and axis-alignment are not fundamental or spontaneous properties of deep learning models. Rather, they are induced by the choice of activation function: anisotropic activations drive anisotropic representations that appear discrete, whereas isotropic activations yield isotropic representations that vary more continuously. Through CIFAR reconstruction experiments, authors demonstrate that isotropic activation functions—by lacking intrinsic “distinguished directions”—mitigate the representation bias induced by anisotropic activations toward direction-centered structure. As a result, models with isotropic activations may better reflect the data’s natural structure in these settings.

---

### Strength
> 1. The PPP method simultaneously preserves both angle and magnitude.
> 2. Mapping vectors onto an orthonormal basis aligned with the distinguished direction provides a more principled dimensionality reduction for analyzing representative orientations than using the full direction vector.


---
### Weakness
> #### **Major Concerns**
>
> **1. Readability** : The introduction, experiments, and conclusion sections appear longer than necessary, which makes the core message harder to follow. It would help to streamline repeated material and to split longer paragraphs into shorter, focused units. At minimum, it would be helpful to make the following points explicit:
> - Section 1.1. Prior work (the Spotlight Resonance Method) established the alignment phenomenon, but it did not clarify whether this bias itself causes the discretization of representations.
>
> - Section 1.2. From an Isotropic Deep Learning perspective, we compare primitives that obey different maximal symmetries. This comparative study shows that their functional forms induce distinct inductive biases, (1) enabling control over the emergence and removal of representational structure, and (2) testing whether downstream alignment is genuinely driven by anisotropic choices—thereby revealing hidden biases embedded in modern deep-learning practice.
>
> - Section 1.3. We directly compare cases with no distinguished directions (isotropic forms) and with distinguished directions (anisotropic forms) to assess whether the presence of such directions is the causal mechanism behind the formation of approximately discrete representations.
>
> - Section 1.4. We advance a detailed hypothesis explaining why and how anisotropic functional forms induce representational quantization, why this can be problematic, and how the present study contributes to identifying the underlying causal mechanism.
>
> **2. Lack of References**: The introduction lacks sufficient citations to support several claims. Assertions that are not grounded in prior work should be demonstrated experimentally, but no validating experiments are provided. The items noted below are only a subset of the gaps; in particular, statements qualified with “may” are presented without references or empirical results. I kindly suggest either (i) supply appropriate citations, (ii) add supporting experiments (e.g., controlled ablations or benchmarks), or (iii) revise the prose to offer a clearer, more intuitive and testable explanation.
>
> - 4$^{th}$ paragraph in section 1.3: what is the “Isotropic paper”?.
>
> - 4$^{th}$ paragraph in page 5: “This may result …”, need reference or experimental results.
>
> **3. Insufficient experimental evidence, 4. Lack of statistical quantitative analysis, 5. Question**
> To strengthen the paper, I kindly ask that authors respond thoroughly to the issues raised in the **box below**, which I consider particularly sensitive areas of the work.


> #### **Minor Concern**
>
> **Notation**: The notation is under-specified: several symbols are not introduced when they first appear, which makes the equations hard to follow. Since the number of symbols is small, a concise notation summary (e.g., a short table or paragraph) should be feasible.

**Audience:**

Yes

**Audience Explanation:**

This paper is likely to be of interest to TMLR readers, as it introduces a tool for explaining model mechanisms.

**Broader Impact Concerns:**

I don't see critical concerns.

**Claims And Evidence:**

No

**Claims Explanation:**

I conclude that the paper does not adequately substantiate the authors' claims in three key respects.
#### Insufficient experimental evidence
>
> - On section 1.3 (first paragraph) claims about downstream transfer and **“more general models.”**:
The claim that “establishing this base case of a causal … can then be expected to carry downstream through **more general models** …” appears under-supported by the current evidence. The experiments are limited to small datasets and effectively a single model configuration, which seems too narrow to justify claims about general models. In addition, the autoencoder architecture is under-specified (e.g., encoder/decoder details and training settings). For more general models, it would be helpful to **1) validate the results on canonical backbones (e.g., CNNs/ResNets)**. While the reported reconstruction errors indicate successful AE training, MNIST and CIFAR are often reconstructable with MLPs; therefore, **2) the architectural choices should be spelled out**. This is particularly important given the unusually small latent dimension for CIFAR (18), which would benefit from a clear rationale. More broadly, the conclusions are not demonstrated beyond CIFAR and a relatively narrow setup; thus, **3) the claim would be strengthened by broader evidence (e.g., multiple datasets and architectures)**.

> - On page 10 (fourth paragraph, final sentence), the claim that the observed effect may be “potentially crucial … in terms of **interpretability**” seems ahead of the current evidence. The results primarily demonstrate isotropy-related effects, but they do not yet establish interpretable structure in a causal or interpretability sense.

#### Lack of statistical quantitative analysis
> - **A need for statistical, quantitative evidence**: To substantiate the paper’s central claims, statistical and quantitative metrics would be helpful. Although the manuscript notes—on page 11 (last paragraph), page 12 (first and second paragraphs), and in the appendix—that certain effects may be statistical artefacts, these statements are not accompanied by quantitative analyses, which makes the conclusions difficult to evaluate.
>
>- **Ambiguity allowing interpretations opposite to the current claims**: In the absence of quantitative metrics, Figures 1, 2, and 14 appear open to alternative interpretations that differ from the manuscript’s reading. For example, Figure 1 suggests alignment to a single direction, whereas Figure 2 (third layer, last column, anisotropic activation) appears to show two or more directions; one could therefore view the multi-layer case as more isotropic than the single-layer case, at least superficially—seemingly at odds with the claim that anisotropic activations induce a task-agnostic discretization bias. Likewise, in Figure 14, comparing the fifth and last rows, the anisotropic case appears more isotropic than the isotropic case. Because such opposite readings are possible from qualitative panels alone, **additional validation and statistical assessment would be very helpful**.

#### Question
> - **Impact of proposed Isotropic Activation Function (Eq. 4)**: In Eq. 4, $\tanh(||x||)$ (a scalar of the vector norm) is multiplied by the normalized vector $\hat{x}=\vec{x}/||x||$, resulting in a vector-level radial rescaling rather than an element-wise nonlinearity.
**(1)** What concrete expressive advantage does this provide over a learnable scalar gate ($\alpha \vec{x}$, where $\alpha$ is a learnable scalar)?
**(2)** Does this nonlinearity (isotropy activation function) work beyond CIFAR reconstruction if the claim targets “general models”?

> - **Isotropic Activation Function vs. Normalization**: Viewed as "normalization + scaling", the isotropic activation resembles the effect of batch normalization (or $L_2$ normalization with a learned again). Therefore, the key comparison should include existing methods such as Batch normalization (and possibly Layer or Group normalization), not only anisotropic activations, because batch normalization is usually utilized in current deep neural networks rather than the proposed isotropic activation function. To demonstrate that isotropic activation is not merely a normalization effect, **please include a comparison experiment between the isotropic activation function and normalization**.
>
> - **Odd Function**: Any odd element-wise activation (e.g., $\text{Tanh}(\cdot)$ or identity matrix) is equivariant to the hyperoctahedral group (signed permutations). Do the phenomenon authors report also appear without activations (as the activation function is an identity matrix)? **Please include ablations 1) without activation function, 2) without normalized vector in Eq. 4 ($\text{Tanh} (||x||) \vec{x}$), and 3) the proposed isotropic activation comparison experiment to isolate which component drives the effect**. I remain unconvinced: the isotropic activation used in the experiments appears to improve reconstruction primarily **due to its normalization effect, rather than its isotropic property**.

**Requested Changes:**

Please address the 1) "Weakness" and the 2) “Are the claims supported by accurate, convincing, and clear evidence?” sections I mentioned above.

---

> ### Author Response · Authors · 2025-11-12
> **Response to Reviewer qrKh (Part 1/3)**
>
> Dear Reviewer qrKh,
>
> Thank you for taking the time to review this manuscript and share thoughtful feedback. Please find the responses below offering clarifications and discussion of several points and questions raised, along with indications of how the updated manuscript has taken your suggestions into account.
>
> To satisfactorily respond to every point, this rebuttal required three parts due to space limitations.
>
> The first concern was readability. It was felt that the introduction and experimental sections were overly long and obscured the paper's significant findings.
>
> To address this, as suggested, several longer paragraphs have now been split, and explicit, concise summary statements have been added for Sections 1.1, 1.2, and 1.3. These can be seen in the final paragraph of 1.1, the 2nd and 4th paragraphs of 1.2 and the 3rd paragraph of 1.3, respectively. These locations improved approachability, and their positioning aided logical flow. Following your concerns about the length and readability of the introduction, section 1.4 has been removed, and any remaining key details are now included in a "Section 2: Theoretical Framework and Considerations", including a more detailed discussion of the phenomena. Redundancy across these is now mitigated. This should improve the approachability of the manuscript's introduction to prospective readers while retaining the technical nuances essential for fully understanding the paper.
>
> The second-mentioned weakness was that claims lacked citations. Particularly, in section 1.3 and paragraph 4 of page 5, although other incidences were noted.
>
> 1) Regarding the first stated comment, this is referring to (Bird, 2025a), discussed explicitly as "Isotropic Deep Learning (Bird, 2025a)" in paragraph 1 of the preceding section 1.2. Therefore, a repeated citation to this paper was not initially given in section 1.3. The citation has now been added in the corrected manuscript following this comment, to make clear that it continues to discuss (Bird, 2025a) and avoid reader confusion. Thank you for highlighting this.
>
> 2) The second example was regarding "*The network’s optimisation may then shape representations about these orthants of differing and equal mapping, to achieve the desired computation. [This may result] in representations that organise differently over these maps related by the symmetry of the functional form.*" This phrase has now been moved to section 2.1 and rephrased for greater precision. Although the overall statement is felt to constitute the central hypothesis being tested and is consistent with the paper's primary finding: the representations appear to be shaped by the function's symmetry through training, as observed throughout the paper's results. These symmetries relate the various computation maps, organised by orthant in permutations, which align with the stated excerpt.
>
> Regarding notation, all equations in the revised manuscript now have terms clearly stated, and a notation table has been provided in the appendices (see Appendix F).
>
> The excerpt, including "base case can [...] general models", is now softened to reflect the speculative nature of the impact on larger models; it appears in the 2nd paragraph of section 1.3. Your suggestion on generalisability discussion similarly echoes that of reviewer bF6V. To address these joint generalisability comments, a further clarifying appendix section has been added to discuss this more thoroughly (see Appendix C.1). This section explicitly discusses these hypotheses, how they might translate to other models, and the difficulties therein. This correction addresses these joint concerns by clarifying the speculated generalisation of these insights and providing a more robust discussion. In addition to rephrasing in the main section, an effort has been made to redirect readers to the more substantive discussion to be added.
>
> In the manuscript, the phenomenon is demonstrated across various widths (18/24/32), depths (0, 1, 2, 3 latent blocks), various layerwise analyses, differing activation functions (B_n/O(n)-Tanh, S_n/B_n/O(n)-Leaky ReLU), standard datasets (CIFAR and MNIST) and multiple initialisation-noise controls. This is not felt to be a single model configuration and is supported beyond CIFAR; instead, the hypothesis is supported across a range of autoencoder models in terms of reconstruction. This is felt to strengthen the claims.
>
> Additionally, it is felt that the models utilised were not underspecified; a dedicated appendix (originally Appendix D, now see Appendix E) provides full model details and training choices for independent replication of all the results. A GitHub link is also provided within the text where it states "[Redacted for Anonymity]".

---

> ### Author Response · Authors · 2025-11-12
> **Response to Reviewer qrKh (Part 2/3)**
>
> Unfortunately, the suggested canonical backbone experiments exhibit significant confounding factors that interfere with the ablation trials at this stage. This necessitates the use of autoencoders to demonstrate that the hypothesised effect exists to some degree. To address this, the new discussion in Appendix C.2 addresses these confounding factors in other architectures, providing readers with greater clarity on why autoencoder reconstruction was necessary in this study and offering a direction for future studies of these alternative model types. This pertains to CNNs exhibiting an inherent architectural anisotropy, and similarly, classification reintroduces permutation-based local coding through the one-hot representations --- so neither can be used for a clean ablation test. Currently, this makes conclusions about functional-form symmetries inseparable from the architecture. Until the framework is further developed, such tests would yield misleading conclusions about the relationship between functional forms and representations.
>
> The weakness regarding "potentially crucial … in terms of interpretability", does not express the crucial intention of the statement, including the term "to investigate", which is present in the full excerpt: "Furthermore, it suggests that symmetry taxonomies are [potentially crucial] *to investigate* [in terms of interpretability], capacity, and resultant performance.". Hence, the excerpt as a whole does not assert a result or conclusion, as suggested (e.g., "Crucial to interpretability"); this is not the intended interpretation of the passage. Instead, the excerpt flags the need for further study to determine the phenomenon's importance to interpretability, now that an effect has been established.
>
> Regarding differing interpretations of figures. A new theoretical section (see section 2) has been introduced to clarify the predictions resulting from the hypothesis, including the concerns you raised in greater detail. Particularly, the hypothesis isn't expecting single, discrete-like patches due to anisotropy, nor isotropic functions that induce uniformly continuous distributions every time; these would be a conflation of the algebraic and representational meanings of the terms anisotropic/isotropic. Nor do multiple clusters per orthant imply that a distribution is more 'isotropic', since the notion of isotropy concerns only the functional form. The hypothesis is instead that an association exists in the *organisation* through a symmetry-led bias. A synopsis is provided below.
>
> > The expectation is only that anisotropic networks *tend* to organise representations into a variable number of clusters which then organise according to the representation of the symmetry, and in extreme cases may approach discrete-like distributions for various reasons. This does not infer the number of clusters; the organisation is the key signature of the study and is successfully amplified by the PPP method.
> >
> > This makes clear that the findings in Figures 1 and 2 are entirely consistent with the theory. Despite showing multiple patches, the theory doesn't infer perfect anisotropic/isotropic distributions, only symmetry-led organisation. Similarly, Figure 14, 5th row, shows four distinct clusters aligned along the standard axes, which is not seen in the isotropic rows. The much greater density of anisotropic representations in PPP also supports this conclusion.
> >
> > Several plots happen to show the extreme anisotropic/isotropic distributions, which was not necessary but did strengthen the conclusion further, stated explicitly in the phrase “*in the datasets studied, the isotropic networks did tend to form representations which smoothly filled the representation space; consequently, enabling a stronger conviction for the conclusions reached. Effectively, this was an incidentally ideal scenario for determining the validity of the hypotheses*”. Hence, figures 1, 2 and 14 are still sufficient for determining the hypothesis, even though they did not display the more extreme phenomenon of distributional an/isotropy.
>
> The PPP method was introduced as a visualisation tool with a primary focus on preserving the magnitude-based distributions and high-angular fidelity that an angular density measurement would otherwise project out. The predecessor tool, SRM, undertakes a more statistically favourable process and remains best suited for such quantitative measurements, particularly angular density measurements. In the response to reviewer MAQR, a discussion of statistical measurements is provided, concluding that the development of a method that unifies SRM and PPP, providing visualisable and statistical results, would be a beneficial future avenue to explore.
>
> The remaining questions raised are answered in the Part 3 response.

---

> ### Author Response · Authors · 2025-11-12
> **Response to Reviewer qrKh's Questions (Part 3/3)**
>
> ### Question 1:
> The learnable scalar-gate proposed for comparison constitutes a linear map, and can be absorbed into the learnable affine map of existing weight and bias parameters $\alpha \left(\mathbf{W} \vec{x}+\vec{b}\right)=\mathbf{W}'\vec{x}+\vec{b}$ --- all layers could be expressed as a single affine transform. Thus, the suggested function adds no additional expressiveness beyond a simple multivariate linear fit. Whereas the isotropic functions are manifestly non-linear due to maximally orthogonal equivariance --- a network equivalence with an affine map is generally not possible, as it is inherently nonlinear.
>
> Thus, isotropic functions produce nonlinear fits. They also remain locally linear about the origin due to the identity Jacobian, hence it can accommodate both nonlinear and linear data. The proposed $\alpha \vec{x}$ function remains solely linear.
>
> (2) The functions also show good reconstruction on the MNIST dataset, as shown in Appendix B.5 and on a wide variety of reconstruction models. Alternatives to reconstruction, such as classification, cannot be used for such ablation tests for the reasons stated above.
>
> ### Question 2:
> I feel this inference does not represent the function's mathematical properties; therefore, I respectfully must disagree with the premise and suggestions. The isotropic activation function does not resemble normalisation as it does not use any statistic that would draw such an equivalence to batch norm or L2 normalisation. For example, it doesn't eliminate any representational degrees of freedom that produce scale invariances characteristic of normalisers. Moreover, batchnorm is permutation-equivariant, not orthogonal. This impression may be due to the vector norm; however, I'd like to reassure that this is a notational convention, and when considering the function as a whole, it does not exhibit any normalisation-like behaviour.
>
> Equivilantly to this notation is $f\left(\left\|\vec{x}\right\|\right)\hat{x}=g\left(\left\|\vec{x}\right\|\right)\vec{x}$, where such apparant normalisation is absent. The Leaky-ReLU results in Appendix B.4 do not include the $\hat{x}$ term in their isotropic notation, and their results remain consistent with the primary findings, since this is a notational artefact. Isotropic-tanh does bound the L2-norm, just as standard tanh bounds the $L_{\infty}$ norm. Hence, the functions remain strongly mathematically analogous except for the symmetry.
>
> Furthermore, along the standard bases, the anisotropic and isotropic forms are functionally *identical*, emphasising that isotropy does not include any notion of a normaliser in its action. Functionally, it is not possible to make the isotropic and anisotropic implementations any more similar than they already are whilst retaining their respective maximal symmetries.
>
> Therefore, it is not felt that batch-norm+standard-tanh is a fair or appropriate comparison, nor batch-norm+standard-tanh, nor batch-norm+isotropic tanh, which would break orthogonal equivariance, nor implementing an L2-norm+isotropic tanh, since these result in fundamental inequivalences or break the desired symmetries. In the manuscript, the phrasing now further emphasises that standard-tanh & isotropic-tanh are perfectly equivalent in function along basis directions, as noted in the paragraph below equations 3 and 4.
>
> ### Question 3:
> (1) The study of this work is maximal symmetries, where an identity mapping would not constitute a hyperoctahedral group, but a general linear group. This is quite distinct, primarily due to the discrete versus continuous nature of such equivariances. It does remain equivariant to hyperoctahedral symmetries, since general linear is a supergroup (same for both isotropic and odd-anisotropic functions, which also satisfy hyperoctahedral, but the former is maximally orthogonal). This is why maximal equivariance is essential for characterising symmetry, as in related DL fields such as Geometric Deep Learning. Therefore, the identity map would be in a category of its own and would not contribute to this anisotropy-versus-isotropy distinction.
>
> (2 & 3) As mentioned, the apparent normalisation is only notational. The functions are already strongly comparable besides symmetry, and identical for standard-tanh along the basis directions --- changing this would destroy comparability between them. Similarly, removing the $\hat{x}$ term in favour of $\vec{x}$ would remove comparability. Removing the activation function (the identity function) would yield the linear fits discussed in Q1 and would not typically be considered a type of deep learning. Hence, it would not relate to the an/isotropic study.
> ### --------
> I thank the reviewer for taking the time to read and provide constructive feedback on this contribution. With the revisions to the manuscript discussed, intended to address all raised concerns, and the clarifications provided, it is hoped that you find this a worthwhile contribution to the TMLR community.

---

### Review · Reviewer_MAQR · 2025-10-30

**Summary Of Contributions:**

The paper studies how the symmetries of activation functions shape hidden representations.
A new visualization tool is introduced: the Privileged-Plane Projective (PPP) method, a successor of a precedently published Spotlight Resonance Method (SRM).
The author perform controlled ablations targeting the activation nonlinearity. Elementwise/anisotropic activations (e.g., tanh, leaky-ReLU variants) privilege coordinate axes (permutation/sign-flip symmetry), and are shown to produce quantized, axis-aligned clusters.

On the contrary, an "isotropic analogue" (orthogonal-equivariant) yields smooth, continuous latent distributions.
The study reports consistent effects across architectures/datasets and notes a preliminary correlation between quantization and reconstruction error in autoencoders.

The paper’s main thesis is that activation-form symmetries induce unintended inductive biases that strongly influence representations; PPP is proposed as a practical probe of such effects.

**Audience:**

Yes

**Audience Explanation:**

The question addressed by the author: "do primitive choices (activations) impose task-agnostic structure on learned representations?" is an important and timely question. The study isolates that effect in a quite convincing way, although a bit preliminar and limited to autoencoder architectures for reconstruction.

The methodology is extremely simple, but original (at least to my knowledge). The PPP method extends and improves the Spotlight Resonance Method, furthermore a lot of care is given to isolate the role of the activation functions symmetries.

The empirical findings are quite consistent and can be summarized as follows:
- anisotropic forms imply quantization and/or axis-aligned clusters
- isotropic forms imply smooth, diffused latent representations
- these results are found across several non-linearities with different symmetries (tanh vs leaky-ReLU)

Furthermore the author establishes a clear conceptual link from algebraic symmetry of the activation functions to representational geometry.

Due to the originality of the approach and vision, and the impact of these choices on the structure of hidden representations this is a work that, if strenghtened by more quantitative metrics, deserve to be discussed by the scientific community.

**Broader Impact Concerns:**

No impact concerns forseeable.

**Claims And Evidence:**

No

**Claims Explanation:**

The experiments are carefully crafted to isolate the hypotesized phenomenon.

However,  while PPP provides visual evidence, the paper needs explicit metrics (angular concentration, clustering, alignment measurements) to be really convincing.

Finally, the sections of the text regarding superposition, robustness, and global “principles”, although interesting, are too speculative without dedicated testing. Consider moving to discussion and clearly labeling them as hypothetical.

**Requested Changes:**

There are two major weakness that affect the paper: firstly, the exposition is too verbose and repetitive. Key ideas recur with near-identical phrasing; several sections could be tightened substantially. As a rule of thumb I believe that the main text could be reduced by at least 1/4 of its lenght or more.

Secondly, as anticipated above, metrics are badly needed to make it a convincing case, beyond visual evidence.

Furthermore, the figures are overcrowded and hard to read. Many PPP heatmaps are tiny; labels/legends are often illegible, obscuring otherwise interesting results.


If the author will make the effort to improve readibility by a substantial re-organization of the text and adds suitable metrics that reinforce the visual impression of the PPT heatmaps I think that this work can be an interesting contribution.

---

> ### Author Response · Authors · 2025-11-12
> **Response to Reviewer MAQR**
>
> Dear Reviewer MAQR,
>
> Thank you for taking the time to read and review this manuscript. In the following rebuttal, each point raised will be discussed in terms of corrections which will improve the manuscript.
>
> Thank you for raising the point on superposition, robustness and general principles. I am pleased that these were an interesting and valuable addition to the paper, offering potential explanations and follow-on considerations from the results.
>
> To make clear in the manuscript that not all of these are conclusions from the data, the phrasing has been revised to more explicitly state that these are speculative explanatory hypotheses, as suggested. Particularly, the robustness claims have now been moved to Appendix C.3 and reaffirm that this is a speculation. Additionally, the manuscript's exposition has been substantially tightened to ensure conciseness and avoid redundancy in its statements. This has involved removing the original section 1.4, where most redundant statements were found, and creating a new section 2: "Theoretical Framework and Considerations", that rephrases the unique discussion points from the prior section 1.4 whilst adding further details. This new section goes into greater depth on the theory for prospective readers who may be interested in the additional nuances, whilst the core, concise understanding remains in the introduction. Thank you for the suggestion to revise the introduction. These suggestions have already improved the clarity of the manuscript.
>
> Concerning representational superposition, it is felt that these statements do follow as conclusions of the data, and the manuscript has now been clarified to support this finding. As seen in Figures 1, 2, 3, among several others, the characteristic arrangements of discrete linear features of superposed phenomena can be repeatedly observed. These are observed exclusively in anisotropic networks, not in their isotropic counterparts, associating the representation phenomena with the discrete symmetry. Additionally, this novel distinction between representation superposition and parameterised superposition is introduced to clarify that two separate but closely related phenomena may coexist in a network. It is the former which appears affected by symmetries of functional forms.
>
> Thank you for highlighting the need for quantitative metrics. I agree that statistical measures can strengthen the interpretability of PPP-based results. While PPP is designed primarily as a high-fidelity visual diagnostic for complex geometry rather than a statistical estimator, I recognise that the current manuscript does not sufficiently articulate this distinction. In this revision, the differences in the use of SRM and PPP are now explicitly noted on page 5, paragraphs 4 through 6, and on page 18, paragraph 5, to inform readers. The former now explicitly explains upfront why SRM is the appropriate tool for angular-density statistics and why PPP is used as the primary tool for studying the complex geometry present. The latter clarifies how the present PPP results should be interpreted in the absence of such metrics. To avoid ambiguity, we will also include a brief discussion of possible future directions for integrating statistical measures (e.g., hybrid PPP–SRM metrics or distribution-estimation approaches). These changes will ensure that the manuscript makes a clear and accurate methodological claim: that PPP provides high-resolution geometric insight complementary to, but distinct from, purely statistical approaches.
>
> Not all figures were felt to be legible, with tiny axes titles present on several. This was undertaken due to .pdf space constraints; the first three plots of the data are full resolution, while subsequent plots are strongly compressed. The titles and axis labels remain consistent across all the PPP plots, as discussed, so that they can be inferred from the first three plots. These figures, due to including a very large number of individual images, were inherently memory-intensive. An attempt was made to strike a compromise between a sufficient number of plots to support all relevant observations and conclusions of the paper, necessitating the compression of the latter plots to make this feasible, whilst sacrificing some fidelity in the text, whilst preserving individual image interpretability. The existing PDF with compression was already very large at 24MB, and unfortunately, this compression of the plots was deemed necessary but unfortunate to further save space.
>
> Thank you also for your concluding comments, noting that the question and ablation study overall yield an important and timely result, consistently establishing a clear conceptual link between activation function symmetries and task-agnostic representational biases in a convincing and straightforward manner.
>
> With the amendments discussed and implemented, it is hoped that you will consider this a beneficial paper to share with the TMLR audience.

---

### Decision · Action_Editor_VwX7 · 2026-01-16

**Recommendation:** Reject

**Additional Comments:**

The reviews for this submission spanned a range from one that is enthusiastic about the work and its significance, to one that is largely critical of the level of evidence and readability, and a third in the middle seeing both clear positive and negative aspects. While the middle reviewer did ultimately decide that the positives outweigh the negatives, this was done with reservations regarding the presentation issues, which I believe still have to be addressed. Overall, given the number of serious concerns that were brought up and the substantial revisions made to address them (as described in the "claims and evidence" section above), I think that the manuscript needs to undergo another full review.

I note that Reviewer qrKh also had several questions and comments about the proposed isotropic activation function. The rebuttal discussed why the suggested comparisons would not be appropriate. It would be good to incorporate this discussion into the paper as well.

**Audience:**

Yes

**Audience Explanation:**

I think that the criterion of "interest" is a major strength of the submission, as both Reviewers bF6V and MAQR were enthusiastic about the positive aspects:
- Both reviewers praised the originality of the research hypothesis, i.e., that algebraic symmetry in network primitives such as activation functions could lead to corresponding structure in the network's representations.
- Reviewer bF6V commented on the potentially significant implications for multiple areas, notably (mechanistic) interpretability, representation learning, and safety and adversarial robustness. For example, the question of why discrete linear features appear in representations is a major question in mechanistic interpretability.

**Claims And Evidence:**

No

**Claims Explanation:**

The main claim of this submission is that algebraic symmetry in the activation functions of a neural network leads to discretization/quantization of the representations within the network. Below I summarize and elaborate upon the reviewers' opinions of the evidence for this hypothesis.

In terms of **strengths**, reviewers commented on the following:
- The experiments were carefully constructed to test the hypothesis, isolating the activation function as the only variable.
- The experimental results are consistent across different activation functions, autoencoder model sizes, the CIFAR and MNIST datasets, and other factors. Reviewer bF6V thought that the consistency for the Leaky-ReLU function was particularly notable, and also appreciated the demonstration that quantization emerges during training.

Two weaknesses were brought up by Reviewers MAQR and/or qrKh, which I group together because I see them as related:
- **Lack of quantitative metrics** for the degree of quantization or axis alignment. The lack of metrics can leave the figures open to alternative, conflicting interpretations.
- Many **figures are too small and too low-quality** to be read

The author rebuttal had the following responses to these points:
- Regarding quantitative metrics, the rebuttal argued that the new Privileged-Plane Projection (PPP) visualization, which builds on the existing Spotlight Resonance Method (SRM), is primarily intended for visualization while SRM is intended for angular-density statistics. Integrating the two was left as future work.
- In response to the alternative interpretations, the new theoretical Section 2 clarifies what one should expect to see in the figures (essentially, the presence/absence of quantization and not more specific predictions about the number of clusters, etc.).
- Regarding figures, the rebuttal explained that because of the large number of images needed to support all findings, the images had to be compressed (substantially so after Figures 1-3) to yield a reasonably-sized PDF file.
- I also note Reviewer bF6V's counterargument that the clear visual contrast between the images for anisotropic and isotropic activation functions reduces the need for statistical analysis.

**My take regarding quantitative metrics and figures:** I mostly side with Reviewers qrKh and MAQR and think it would be good to have statistics that summarize the images, focusing on the degree of quantization. I think these do not have to be perfect in terms of statistical properties, and they may not require a full integration of the PPP and SRM methods as proposed. Rather, I think the value would lie mainly in having a way to summarize images. This would also reduce the need to include such a large number of images, at least in the main paper PDF. I would still encourage the inclusion of the images (with larger size and higher quality), and note that TMLR allows the submission of up to 100 MB of supplementary material, which could be used for this purpose.

Reviewers identified several **claims that are not well supported or are speculative** in nature:
- Generalization to other models is not well supported because the experiments are limited to a single autoencoder architecture.
- The parts on interpretability, superposition, and robustness are speculative and should be more clearly labelled as such.

The rebuttal responded to these points as follows:
- The claim about generalization is now softened and a clarifying appendix has been added. (However, one of the reviewers commented to me that the discussion in this appendix may indicate a limitation of the approach to model architectures without inherent anisotropies.)
- The claim about interpretability was clarified as being about the need for further study to determine the importance to interpretability, not that the importance has been established.
- The rebuttal asserts that the claim about superposition is indeed supported by the figures (but I think this should be described more concretely).
- The discussion of robustness has been moved to the appendix.

**My take regarding these unsupported/speculative claims** is that this was a concern raised by all reviewers to varying degrees and affects several claims. While the rebuttal and revised manuscript appear to go a long way in addressing the issues, I think that the initial magnitude of the concern calls for a fuller review of the revisions.

Finally, Reviewers MAQR and qrKh both noted that the **writing is verbose and repetitive** and paragraphs are long. As the rebuttal describes, the revised manuscript has taken steps to address these comments, by removing Section 1.4 from the introduction and replacing it with a new Section 2 on theory, adding summary sentences to subsections of the introduction, etc. Nevertheless, the reviewers' concern remains. While this is not about the accuracy or convincingness of evidence for claims, it does relate to clarity, as the writing style may make it difficult for readers to appreciate the content of the paper. I therefore encourage the authors to undertake a deeper, top-to-bottom revision in this regard.

**Resubmission Of Major Revision:**

The authors may consider submitting a major revision at a later time.